# One for All: Simultaneous Metric and Preference Learning over Multiple Users

**Gregory Canal**
University of Wisconsin-Madison
Madison, WI
gcanal@wisc.edu

**Blake Mason**
Rice University
Houston, TX
bm63@rice.edu

**Ramya Korlakai Vinayak**
University of Wisconsin-Madison
Madison, WI
ramya@ece.wisc.edu

**Robert Nowak**
University of Wisconsin-Madison
Madison, WI
rdnowak@wisc.edu

## Abstract

This paper investigates simultaneous preference and metric learning from a crowd of respondents. A set of items represented by $d$-dimensional feature vectors and paired comparisons of the form "item $i$ is preferable to item $j$" made by each user is given. Our model jointly learns a distance metric that characterizes the crowd's general measure of item similarities along with a latent ideal point for each user reflecting their individual preferences. This model has the flexibility to capture individual preferences, while enjoying a metric learning sample cost that is amortized over the crowd. We first study this problem in a noiseless, continuous response setting (i.e., responses equal to differences of item distances) to understand the fundamental limits of learning. Next, we establish prediction error guarantees for noisy, binary measurements such as may be collected from human respondents, and show how the sample complexity improves when the underlying metric is low-rank. Finally, we establish recovery guarantees under assumptions on the response distribution. We demonstrate the performance of our model on both simulated data and on a dataset of color preference judgments across a large number of users.

## 1 Introduction

In many data-driven recommender systems (e.g., streaming services, online retail), multiple users interact with a set of items (e.g., movies, products) that are common to all users. While each user has their individual preferences over these items, there may exist shared structure in how users *perceive* items when making preference judgments. This is a reasonable assumption, since collections of users typically have shared perceptions of similarity between items regardless of their individual item preferences [1–3]. In this work we develop and analyze models and algorithms for simultaneously learning individual preferences and the common metric by which users make preference judgments.

Specifically, suppose there exists a known, fixed set $\mathcal{X}$ of $n$ items, where each item $i \in 1, \ldots, n$ is parameterized by a feature vector $x_i \in \mathbb{R}^d$. We model the crowd's preference judgments between items as corresponding to a common Mahalanobis distance metric $d_M(x, y) = \|x - y\|_M$, where $\|x\|_M := \sqrt{x^T M x}$ and $M$ is a $d \times d$ positive semidefinite matrix to be learned. Measuring distances with $d_M$ has the effect of reweighting individual features as well as capturing pairwise interactions between features. To capture individual preferences amongst the items, we associate with each of $K$ users an *ideal point* $u_k \in \mathbb{R}^d$ for $k \in 1, \ldots, K$ such that user $k$ prefers items that are closer to $u_k$ than those items that are farther away, as measured by the common metric $d_M$. The ideal point

model is attractive since it can capture nonlinear notions of preference, and preference rankings are determined simply by sorting item distances to each user point and can therefore be easily generalized to items outside of $\mathcal{X}$ with known embedding features [4–7] . Furthermore, once a user's point $\boldsymbol{u}_k$ is estimated, in some generative modeling applications it can then be used to synthesize an "ideal" item for the user located exactly at $\boldsymbol{u}_k$, which by definition would be their most preferred item if it existed.

In order to learn the metric and ideal points, we issue a series of *paired comparison* queries to each user in the form "do you prefer item $i$ or item $j$?" Since such preferences directly correspond to distance rankings in $\mathbb{R}^d$, these comparisons provide a signal from which the user points $\{\boldsymbol{u}_k\}_{k=1}^K$ and common metric $\boldsymbol{M}$ can be estimated. *The main contribution of this work is a series of identifiability, prediction, and recovery guarantees to establish the first theoretical analysis of simultaneous preference and metric learning from paired comparisons over multiple users.* Our key observation is that by modeling a shared metric between all users rather than learning separate metrics for each user, the sample complexity is reduced from $O(d^2)$ paired comparisons per user to only $O(d)$, which is the sample cost otherwise required to learn each ideal point; in essence, when amortizing metric and preference learning over multiple users, the metric comes for free. Our specific contributions include:

- Necessary and sufficient conditions on the number of items and paired comparisons required for exact preference and metric estimation over generic items, when noiseless differences of item distances are known exactly. These results characterize the fundamental limits of our problem in an idealized setting, and demonstrate the benefit of amortized learning over multiple users. Furthermore, when specialized to $K = 1$ our results significantly advance the existing theory of identifiability for single-user simultaneous metric and preference learning [7].

- Prediction guarantees when learning from noisy, one-bit paired comparisons (rather than exact distance comparisons). We present prediction error bounds for two convex algorithms that learn full-rank and low-rank metrics respectively, and again illustrate the sample cost benefits of amortization.

- Recovery guarantees on the metric and ideal points when learning from noisy, binary labels under assumptions on the response distribution. Furthermore, we empirically validate the recovery performance of our multi-user learning algorithms on both synthetic datasets as well as on real psychometrics data studying individual and collective color preferences and perception.

**Summary of related work:** Metric and preference learning are both extensively studied problems (see [8] and [9] for surveys of each). A common paradigm in metric learning is that by observing distance comparisons, one can learn a linear [10–12], kernelized [13, 14], or deep metric [15, 16] and use it for downstream tasks such as classification. Similarly, it is common in preference learning to use comparisons to learn a ranking or to identify a most preferred item [5, 6, 17–19]. An important family of these algorithms reduces preference learning to identifying an ideal point for a fixed metric [5, 20]. The closest work to ours is [7], who perform metric and preference learning simultaneously from paired comparisons in the single-user case and propose an alternating minimization algorithm that achieves empirical success. However, that work leaves open the question of theoretical guarantees for the simultaneous learning problem, which we address here. A core challenge when establishing such guarantees is that the data are a function of multiple latent parameters (i.e., unknown metric and ideal point(s)) that interact with each other in a nonlinear manner, which complicates standard generalization and identifiability arguments. To this end, we introduce new theoretical tools and advance the techniques of [12] who showed theoretical guarantees for learning a Mahalanobis metric from triplet queries. We survey additional related work more extensively in Appendix B.

**Notation:** Let $[K] := 1 \ldots K$. Unless specified otherwise, $\|\cdot\|$ denotes the $\ell_2$ norm when acting on a vector, and the operator norm induced by the $\ell_2$ norm when acting on a matrix. Let $\boldsymbol{e}_i$ denote the $i$th standard basis vector, $\mathbf{1}$ the vector of all ones, $\mathbf{0}_{a,b}$ the $a \times b$ matrix of all zeros (or $\mathbf{0}$ if the dimensions are clear), and $\boldsymbol{I}$ the identity matrix, where the dimensionality is inferred from context. For a symmetric $d \times d$ matrix $\boldsymbol{A}$, let $\mathrm{vec}^*(\boldsymbol{A}) := [\boldsymbol{A}_{1,1}, \boldsymbol{A}_{1,2}, \ldots, \boldsymbol{A}_{1,d}, \boldsymbol{A}_{2,2}, \boldsymbol{A}_{2,3}, \ldots, \boldsymbol{A}_{2,d}, \ldots \boldsymbol{A}_{d,d}]^T$ denote the vectorized upper triangular portion of $\boldsymbol{A}$, which is a $D$-length vector where $D := d(d+1)/2$. Let $\boldsymbol{u} \otimes_S \boldsymbol{v} := \mathrm{vec}^*(\boldsymbol{u}\boldsymbol{v}^T)$ denote the unique entries of the Kronecker product between vectors $\boldsymbol{u}, \boldsymbol{v} \in \mathbb{R}^d$, and let $\odot$ denote the Hadamard (or element-wise) product between two matrices.

## 2 Identifiability from unquantized measurements

In this section, we characterize the fundamental limits on the number of items and paired comparisons per user required to identify $\boldsymbol{M}$ and $\{\boldsymbol{u}_k\}_{k=1}^K$ exactly. In order to understand the fundamental

hardness of this problem, we begin by presenting identifiability guarantees under the idealized case where we receive *exact, noiseless* difference of distance measurements[1], before deriving similar results in the case of *noisy* realizations of the *sign* of these differences in the following sections.

We formally define our model as follows: if user $k$ responds that they prefer item $i$ to item $j$, then $\|\boldsymbol{x}_i - \boldsymbol{u}_k\|_{\boldsymbol{M}} < \|\boldsymbol{x}_j - \boldsymbol{u}_k\|_{\boldsymbol{M}}$. Equivalently, by defining

$$\delta_{i,j}^{(k)} := \|\boldsymbol{x}_i - \boldsymbol{u}_k\|_{\boldsymbol{M}}^2 - \|\boldsymbol{x}_j - \boldsymbol{u}_k\|_{\boldsymbol{M}}^2 = \boldsymbol{x}_i^T \boldsymbol{M} \boldsymbol{x}_i - \boldsymbol{x}_j^T \boldsymbol{M} \boldsymbol{x}_j - 2\boldsymbol{u}_k^T \boldsymbol{M}(\boldsymbol{x}_i - \boldsymbol{x}_j), \qquad (1)$$

user $k$ prefers item $i$ over item $j$ if $\delta_{i,j}^{(k)} < 0$ (otherwise $j$ is preferred). In this section, we assume that $\delta_{i,j}^{(k)}$ is measured exactly, and refer to this measurement type as an *unquantized* paired comparison. Let $m_k$ denote the number of unquantized paired comparisons answered by user $k$ and let $m_T := \sum_{k=1}^{K} m_k$ denote the total number of comparisons made across all users.

It is not immediately clear if recovery of both $\boldsymbol{M}$ and $\{\boldsymbol{u}_k\}_{k=1}^{K}$ is possible from such measurements, which depend quadratically on the item vectors. In particular, one can conceive of pathological examples where these parameters are not identifiable (i.e., there exists no unique solution). For instance, suppose $d = n$, $\boldsymbol{M} = \alpha \boldsymbol{I}$ for a scalar $\alpha > 0$, $\boldsymbol{x}_i = \boldsymbol{e}_i$ for $i \in [n]$, and for each user $\boldsymbol{u}_k = \beta_k \mathbf{1}$ for a scalar $\beta_k$. Then one can show that $\delta_{i,j}^{(k)} = 0$ for all $i, j, k$, and therefore $\alpha, \beta_1, \ldots, \beta_K$ are unidentifiable from any set of paired comparisons over $\mathcal{X}$. In what follows, we derive necessary and sufficient conditions on the number and geometry of items, number of measurements per user, and interactions between measurements and users in order for the latent parameters to be identifiable.

Note that eq. (1) includes a nonlinear interaction between $\boldsymbol{M}$ and $\boldsymbol{u}_k$; however, by defining $\boldsymbol{v}_k := -2\boldsymbol{M}\boldsymbol{u}_k$ (which we refer to as user $k$'s "pseudo-ideal point," reflecting the component of $\boldsymbol{u}_k$ in the column space of $\boldsymbol{M}$) eq. (1) becomes linear in $\boldsymbol{M}$ and $\boldsymbol{v}_k$:

$$\delta_{i,j}^{(k)} = \boldsymbol{x}_i^T \boldsymbol{M} \boldsymbol{x}_i - \boldsymbol{x}_j^T \boldsymbol{M} \boldsymbol{x}_j + (\boldsymbol{x}_i - \boldsymbol{x}_j)^T \boldsymbol{v}_k. \qquad (2)$$

If $\boldsymbol{M}$ is full-rank and the system of equations admits unique solutions for $\boldsymbol{M}$ and $\{\boldsymbol{v}_k\}_{k=1}^{K}$, then $\boldsymbol{u}_k$ can be recovered exactly from $\boldsymbol{v}_k$.[2] In other words, the non-convex eq. (1) can be solved in two stages by first solving a linear relaxation (2) in terms of $\boldsymbol{M}$ and $\boldsymbol{v}_k$, and then solving for $\boldsymbol{u}_k$. Note that since $\boldsymbol{M}$ is symmetric, we may write $\boldsymbol{x}_i^T \boldsymbol{M} \boldsymbol{x}_i = \langle \text{vec}^*(2\boldsymbol{M} - \boldsymbol{I} \odot \boldsymbol{M}), \boldsymbol{x}_i \otimes_S \boldsymbol{x}_i \rangle$. Defining $\boldsymbol{\nu}(\boldsymbol{M}) := \text{vec}^*(2\boldsymbol{M} - \boldsymbol{I} \odot \boldsymbol{M})$, from which $\boldsymbol{M}$ can be determined, we have

$$\delta_{i,j}^{(k)} = \begin{bmatrix} (\boldsymbol{x}_i \otimes_S \boldsymbol{x}_i - \boldsymbol{x}_j \otimes_S \boldsymbol{x}_j)^T & (\boldsymbol{x}_i - \boldsymbol{x}_j)^T \end{bmatrix} \begin{bmatrix} \boldsymbol{\nu}(\boldsymbol{M}) \\ \boldsymbol{v}_k \end{bmatrix}.$$

By concatenating all user measurements in a single linear system, we can directly show conditions for identifiability of $\boldsymbol{M}$ and $\{\boldsymbol{v}_k\}_{k=1}^{K}$ by characterizing when the system admits a unique solution. To do so, we define a class of matrices that will encode the item indices in each pair queried to each user:

**Definition 2.1.** A $a \times b$ matrix $\boldsymbol{S}$ is a *selection matrix* if for every $i \in [a]$, there exist distinct indices $p_i, q_i \in [b]$ such that $\boldsymbol{S}[i, p_i] = 1$, $\boldsymbol{S}[i, q_i] = -1$, and $\boldsymbol{S}[i, j] = 0$ for $j \in [b] \setminus \{p_i, q_i\}$.

In Appendix C, we characterize several theoretical properties of selection matrices, which will be useful in proving the results that follow.

For each user $k$, we represent their queried pairs by a $m_k \times n$ selection matrix denoted $\boldsymbol{S}_k$, where each row selects a pair of items corresponding to its nonzero entries. Letting $\boldsymbol{X} := [\boldsymbol{x}_1, \ldots, \boldsymbol{x}_n] \in \mathbb{R}^{d \times n}$, $\boldsymbol{X}_\otimes := [\boldsymbol{x}_1 \otimes_S \boldsymbol{x}_1, \ldots, \boldsymbol{x}_n \otimes_S \boldsymbol{x}_n] \in \mathbb{R}^{D \times n}$, and $\boldsymbol{\delta}_k \in \mathbb{R}^{m_k}$ denote the vector of unquantized measurement values for user $k$, we can write the entire linear system over all users as a set of $m_T$ equations with $D + dK$ variables to be recovered:

$$\boldsymbol{\Gamma} \begin{bmatrix} \boldsymbol{\nu}(\boldsymbol{M}) \\ \boldsymbol{v}_1 \\ \vdots \\ \boldsymbol{v}_K \end{bmatrix} = \begin{bmatrix} \boldsymbol{\delta}_1 \\ \vdots \\ \boldsymbol{\delta}_K \end{bmatrix} \quad \text{where } \boldsymbol{\Gamma} := \begin{bmatrix} \boldsymbol{S}_1 \boldsymbol{X}_\otimes^T & \boldsymbol{S}_1 \boldsymbol{X}^T & \mathbf{0}_{m_1,d} & \cdots & \mathbf{0}_{m_1,d} \\ \boldsymbol{S}_2 \boldsymbol{X}_\otimes^T & \mathbf{0}_{m_2,d} & \boldsymbol{S}_2 \boldsymbol{X}^T & \cdots & \mathbf{0}_{m_2,d} \\ \vdots & \vdots & \vdots & \vdots & \vdots \\ \boldsymbol{S}_K \boldsymbol{X}_\otimes^T & \mathbf{0}_{m_K,d} & \mathbf{0}_{m_K,d} & \cdots & \boldsymbol{S}_K \boldsymbol{X}^T \end{bmatrix}. \qquad (3)$$

---

[1]We use the term "measurement" interchangeably with "paired comparison."

[2]If $\boldsymbol{M}$ were rank deficient, only the component of $\boldsymbol{u}_k$ in the row space of $\boldsymbol{M}$ affects $\delta_{i,j}^{(k)}$. In this case, there is an equivalence class of user points that accurately model their responses. We then take $\boldsymbol{u}_k$ to be the minimum norm solution, i.e., $\boldsymbol{u}_k = -\frac{1}{2}\boldsymbol{M}^\dagger \boldsymbol{v}_k$. This generalizes Proposition 1 of [7] for the multiple user case.

From this linear system, it is clear that $\boldsymbol{\nu}(\boldsymbol{M})$ (and hence $\boldsymbol{M}$) and $\{\boldsymbol{v}_k\}_{k=1}^K$ (and hence $\{\boldsymbol{u}_k\}_{k=1}^K$, if $\boldsymbol{M}$ is full-rank) can be recovered exactly if and only if $\boldsymbol{\Gamma}$ has full column rank. In the following sections, we present necessary and sufficient conditions for this to occur.

## 2.1 Necessary conditions for identifiability

To build intuition, note that the metric $\boldsymbol{M}$ has $D$ degrees of freedom and each of the $K$ pseudo-ideal points $\boldsymbol{v}_k$ has $d$ degrees of freedom. Hence, there must be at least $m_T \geq D + Kd$ measurements in total (i.e., rows of $\boldsymbol{\Gamma}$) to have any hope of identifying $\boldsymbol{M}$ and $\{\boldsymbol{v}_k\}_{k=1}^K$. When amortized over the $K$ users, this corresponds to each user providing at least $d + D/K$ measurements on average. In general, $d$ of these measurements are responsible for identifying each user's own pseudo-ideal point (since $\boldsymbol{v}_k$ is purely a function of user $k$'s responses), while the remaining $D/K$ contribute towards a *collective* set of $D$ measurements needed to identify the common metric. While these $D$ measurements must be linearly independent from each other and from those used to learn the ideal points, a degree of overlap is acceptable in the additional $d$ measurements each user provides, as the $\boldsymbol{v}_k$'s are independent of one another. We formalize this intuition in the following proposition, where we let $\boldsymbol{S}_T := [\boldsymbol{S}_1^T, \ldots, \boldsymbol{S}_K^T]^T$ denote the concatenation of all user selection matrices.

**Proposition 2.1.** *If $\boldsymbol{\Gamma}$ has full column rank, then $\sum_{k=1}^K m_k \geq D + dK$ and the following must hold:*

(a) *for all $k \in [K]$, $\mathrm{rank}(\boldsymbol{S}_k \boldsymbol{X}^T) = d$, and therefore $\mathrm{rank}(\boldsymbol{S}_k) \geq d$ and $m_k \geq d$*

(b) *$\sum_{k=1}^K \mathrm{rank}(\boldsymbol{S}_k \begin{bmatrix} \boldsymbol{X}_\otimes^T & \boldsymbol{X}^T \end{bmatrix}) \geq D + dK$, and therefore $\sum_{k=1}^K \mathrm{rank}(\boldsymbol{S}_k) \geq D + dK$*

(c) *$\mathrm{rank}(\boldsymbol{S}_T \begin{bmatrix} \boldsymbol{X}_\otimes^T & \boldsymbol{X}^T \end{bmatrix}) = D + d$, and therefore $\mathrm{rank}(\boldsymbol{S}_T) \geq D + d$, $\mathrm{rank}(\begin{bmatrix} \boldsymbol{X}_\otimes^T & \boldsymbol{X}^T \end{bmatrix}) = D + d$, and $n \geq D + d + 1$*

If $\sum_{k=1}^K m_k = D + dK$ exactly, then (a) and (b) are equivalent to $m_k \geq d \; \forall \, k$ and each user's selection matrix having full row rank. (c) implies that the number of required items $n$ scales as $\Omega(d^2)$; in higher dimensional feature spaces, this scaling could present a challenge since it might be difficult in practice to collect such a large number of items for querying. Finally, note that the conditions in Proposition 2.1 are *not* sufficient for identifiability: in Appendix C.6, we present a counterexample where these necessary properties are fulfilled, yet the system is not invertible.

## 2.2 Sufficient condition for identifiability

Next, we present a class of pair selection schemes that are sufficient for parameter identifiability and match the item and measurement count lower bounds in Proposition 2.1. This result leverages the idea that as long the the $d$ measurements each user provides to learn their ideal point do not "overlap" with the $D$ measurements collectively provided to learn the metric, then the set of $m_T$ total measurements is sufficiently rich to ensure a unique solution. First, we define a property of certain selection matrices where each pair introduces at least one new item that has not yet been selected:

**Definition 2.2.** An $m \times n$ selection matrix $\boldsymbol{S}$ is *incremental* if for all $i \in [m]$, at least one of the following is true, where $p_i$ and $q_i$ are as defined in Definition 2.1: (a) for all $j < i$, $\boldsymbol{S}[j, p_i] = 0$; (b) for all $j < i$, $\boldsymbol{S}[j, q_i] = 0$.

We now present a class of invertible measurement schemes that builds on the definition of incrementality. For simplicity assume that $m_T = D + dK$ exactly, which is the lower bound from Proposition 2.1. Additionally, assume without loss of generality that each $m_k > d$; if instead there existed a user $k^*$ such that $m_{k^*} = d$ exactly, one can show under the necessary conditions in Proposition 2.1 that the system would separate into two subproblems where first the metric would need to be learned from the other $K - 1$ users, and then $\boldsymbol{v}_{k^*}$ is solved for directly from user $k^*$'s measurements.

**Proposition 2.2.** *Let $K \geq 1$, and suppose $m_k > d \; \forall \, k \in [K]$, $m_T = D + dK$, and $n \geq D + d + 1$. Suppose that for each $k \in [K]$, there exists a $d \times n$ selection matrix $\boldsymbol{S}_k^{(1)}$ and $m_k - d \times n$ selection matrix $\boldsymbol{S}_k^{(2)}$ such that $\boldsymbol{S}_k = \begin{bmatrix} (\boldsymbol{S}_k^{(1)})^T & (\boldsymbol{S}_k^{(2)})^T \end{bmatrix}^T$, and that the following are true:*

(a) *For all $k \in [K]$, $\mathrm{rank}(\boldsymbol{S}_k^{(1)}) = d$*

*(b) Defining the $D \times n$ selection matrix $\boldsymbol{S}^{(2)}$ as $\boldsymbol{S}^{(2)} := \left[\, (\boldsymbol{S}_1^{(2)})^T \; \cdots \; (\boldsymbol{S}_K^{(2)})^T \,\right]^T$, there exists a $D \times D$ permutation $\boldsymbol{P}$ such that for each $k \in [K]$, $\begin{bmatrix} \boldsymbol{S}_k^{(1)} \\ \boldsymbol{P}\boldsymbol{S}^{(2)} \end{bmatrix}$ is incremental*

*Additionally, suppose each item $\boldsymbol{x}_i$ is sampled i.i.d. from a distribution $p_X$ that is absolutely continuous with respect to the Lebesgue measure. Then with probability 1, $\boldsymbol{\Gamma}$ has full column rank.*

**Remark** 2.3. In Appendix C.6 we construct a pair selection scheme that satisfies the conditions[3] in Proposition 2.2 while only using the minimum number of measurements and items, with $m_k = d + D/K$ (and therefore $m_T = D + dK$) and $n = D + d + 1$. Importantly, this construction confirms that the lower bounds on the number of measurements and items in Proposition 2.1 are in fact tight. Since $D = O(d^2)$, if $K = \Omega(d)$ then only $m_k = O(d)$ measurements are required per user. This scaling demonstrates the benefit of amortizing metric learning across multiple users, since in the single user case $D + d = \Omega(d^2)$ measurements would be required.

## 2.3 Single user case

In the case of a single user ($K = 1$), it is straightforward to show that the necessary and sufficient selection conditions in Proposition 2.1 and Proposition 2.2 respectively are *equivalent*, and simplify to the condition that $\mathrm{rank}(\boldsymbol{S}) \geq D + d$ (where we drop the subscript on $\boldsymbol{S}_1$). In a typical use case, a practitioner is unlikely to explicitly select pair indices that result in $\boldsymbol{S}$ being full-rank, and instead would select pairs uniformly at random from the set of $\binom{n}{2}$ unique item pairs. By proving a tail bound on the number of random comparisons required for $\boldsymbol{S}$ to be full-rank, we have with high probability that randomly selected pairs are sufficient for metric and preference identifiability in the single user case. We summarize these results in the following corollary:

**Corollary 2.3.1.** *When $K = 1$, if $\boldsymbol{\Gamma}$ is full column rank then $\mathrm{rank}(\boldsymbol{S}) \geq D + d$. Conversely, for a fixed $\boldsymbol{S}$ satisfying $\mathrm{rank}(\boldsymbol{S}) \geq D + d$, if each $\boldsymbol{x}_i$ is sampled i.i.d. according to a distribution $p_X$ that is absolutely continuous with respect to the Lebesgue measure then $\boldsymbol{\Gamma}$ is full column rank with probability 1. If each pair is selected independently and uniformly at random with $n = \Omega(D + d)$ and $m_T = \Omega(D + d)$, then if $\boldsymbol{x}_i$ is drawn i.i.d. from $p_X$, $\boldsymbol{\Gamma}$ has full column rank with high probability.*

Importantly, the required item and sample complexity for randomly selected pairs matches the lower bounds in Proposition 2.1 up to a constant. As we describe in Appendix C.7, we conjecture that a similar result holds for the multiuser case ($K > 1$), which is left to future work.

# 3 Prediction and generalization from binary labels

In practice, we do not have access to exact difference of distance measurements. Instead, paired comparisons are one-bit measurements (given by the user preferring one item over the other) that are sometimes noisy due to inconsistent user behavior or from model deviations. In this case, rather than simply solving a linear system, we must optimize a loss function that penalizes incorrect response predictions while enforcing the structure of our model. In this section, we apply a different set of tools from statistical learning theory to characterize the sample complexity of randomly selected paired comparisons under a general noise model, optimized under a general class of loss functions.

We assume that each pair $p$ is sampled uniformly with replacement from the set of $\binom{n}{2}$ pairs, and the user $k$ queried at each iteration is independently and uniformly sampled from the set of $K$ users. For a pair $p = (i, j)$ given to user $k$, we observe a (possibly noisy) binary response $y_p^{(k)}$ where $y_p^{(k)} = -1$ indicates that user $k$ prefers item $i$ to $j$, and $y_p^{(k)} = 1$ indicates that $j$ is preferred. Let $\mathcal{S} := \{(p, k, y_p^{(k)})\}_{p=(i,j)}$ be an i.i.d. joint dataset over pairs $p$, selected users $k$, and responses $y_p^{(k)}$, where $|\mathcal{S}|$ denotes the number of such data points. We wish to learn $\boldsymbol{M}$ and vectors $\{\boldsymbol{u}_k\}_{k=1}^K$ that

---

[3]We note that these conditions are not exhaustive: in Appendix C.6 we construct an example where $\boldsymbol{\Gamma}$ is full column rank, yet the conditions in Proposition 2.2 are not met. A general set of matching necessary and sufficient identifiability conditions on $\{\boldsymbol{S}_k\}_{k=1}^K$ has remained elusive; towards this end, in Appendix C.7 we describe a more comprehensive set of conditions that we conjecture are sufficient for identifiability.

predict the responses in $\mathcal{S}$: given a convex, $L$-Lipschitz loss $\ell \colon \mathbb{R} \to \mathbb{R}_{\geq 0}$,[4] we wish to solve

$$\min_{\boldsymbol{M}, \{\boldsymbol{u}_k\}_{k=1}^K} \frac{1}{|\mathcal{S}|} \sum_{\mathcal{S}} \ell\left(y_p^{(k)}\left(\|\boldsymbol{u}_k - \boldsymbol{x}_i\|_M^2 - \|\boldsymbol{u}_k - \boldsymbol{x}_j\|_M^2\right)\right)$$

$$\text{s.t. } \boldsymbol{M} \succeq 0, \|\boldsymbol{M}\|_F \leq \lambda_F, \|\boldsymbol{u}_k\|_2 \leq \lambda_u \ \forall k \in [K], |\delta_{i,j}^{(k)}| \leq \gamma \ \forall i, j, k \tag{4}$$

where $\lambda_F, \lambda_u, \gamma > 0$ are hyperparameters and $\delta_p^{(k)}$ is defined as in eq. (1). The constraint $\boldsymbol{M} \succeq 0$ ensures that $\boldsymbol{M}$ defines a metric, the Frobenius and $\ell_2$ norm constraints prevent overfitting, and the constraint on $\delta_p^{(k)}$ is a technical point to avoid pathological cases stemming from coherent $\boldsymbol{x}$ vectors.

The above optimization is non-convex due to the interaction between the $\boldsymbol{M}$ and $\boldsymbol{u}$ terms. Instead, as in Section 2 we define $\boldsymbol{v}_k := -2\boldsymbol{M}\boldsymbol{u}_k$ and solve the relaxation[5]

$$\min_{\boldsymbol{M}, \{\boldsymbol{v}_k\}_{k=1}^K} \widehat{R}(\boldsymbol{M}, \{\boldsymbol{v}_k\}_{k=1}^K) \text{ s.t. } \boldsymbol{M} \succeq 0, \|\boldsymbol{M}\|_F \leq \lambda_F, \|\boldsymbol{v}_k\|_2 \leq \lambda_v \ \forall k \in [K], |\delta_{i,j}^{(k)}| \leq \gamma \ \forall i, j, k$$

$$\text{where } \widehat{R}(\boldsymbol{M}, \{\boldsymbol{v}_k\}_{k=1}^K) := \frac{1}{|\mathcal{S}|} \sum_{\mathcal{S}} \ell\left(y_p^{(k)}\left(\boldsymbol{x}_i^T \boldsymbol{M} \boldsymbol{x}_i - \boldsymbol{x}_j^T \boldsymbol{M} \boldsymbol{x}_j + \boldsymbol{v}_k^T(\boldsymbol{x}_i - \boldsymbol{x}_j)\right)\right). \tag{5}$$

The quantity $\widehat{R}(\boldsymbol{M}, \{\boldsymbol{v}_k\}_{k=1}^K)$ is the *empirical risk*, given dataset $\mathcal{S}$. The empirical risk is an unbiased estimate of the true risk given by

$$R(\boldsymbol{M}, \{\boldsymbol{v}_k\}_{k=1}^K) := \mathbb{E}\left[\ell\left(y_p^{(k)}\left(\boldsymbol{x}_i^T \boldsymbol{M} \boldsymbol{x}_i - \boldsymbol{x}_j^T \boldsymbol{M} \boldsymbol{x}_j + \boldsymbol{v}_k^T(\boldsymbol{x}_i - \boldsymbol{x}_j)\right)\right)\right],$$

where the expectation is with respect to a random draw of $p = (i, j)$, $k$, and $y_p^{(k)}$ conditioned on the choice of $p$ and $k$. Let $\widehat{\boldsymbol{M}}$ and $\{\widehat{\boldsymbol{v}}_k\}_{k=1}^K$ denote the minimizers of the empirical risk optimization in eq. (5), and let $\boldsymbol{M}^*$ and $\{\boldsymbol{v}_k^*\}_{k=1}^K$ minimize the *true* risk, subject to the same constraints. The following theorem bounds the excess risk of the empirical optimum $R(\widehat{\boldsymbol{M}}, \{\widehat{\boldsymbol{v}}_k\}_{k=1}^K)$ relative to the optimal true risk $R(\boldsymbol{M}^*, \{\boldsymbol{v}_k^*\}_{k=1}^K)$.

**Theorem 3.1.** *Suppose $\|\boldsymbol{x}_i\|_2 \leq 1$ for all $i \in [n]$. With probability at least $1 - \delta$,*

$$R(\widehat{\boldsymbol{M}}, \{\widehat{\boldsymbol{v}}_k\}_{k=1}^K) - R(\boldsymbol{M}^*, \{\boldsymbol{v}_k^*\}_{k=1}^K) \leq \sqrt{\frac{256 L^2(\lambda_F^2 + K\lambda_v^2)}{|\mathcal{S}|}\log(d^2 + d + 1)}$$

$$+ \frac{\sqrt{128 L^2(\lambda_F^2 + K\lambda_v^2)}}{3|\mathcal{S}|}\log(d^2 + d + 1) + \sqrt{\frac{8L^2\gamma^2\log(\frac{2}{\delta})}{|\mathcal{S}|}}. \tag{6}$$

*Remark* 3.2. To put this result in context, suppose $\|\boldsymbol{M}^*\|_F = d$ so that the average squared magnitude of each entry is a constant, in which case we can set $\lambda_F = d$. Similarly, if each entry of $\boldsymbol{v}_k$ is dimensionless, then $\|\boldsymbol{v}_k\|_2 \propto \sqrt{d}$ and so we can set $\lambda_v = \sqrt{d}$. We then have that the excess risk in eq. (6) is $\widetilde{O}\left(\sqrt{\frac{d^2 + Kd}{|\mathcal{S}|}}\right)$ where $\widetilde{O}$ suppresses logarithmic factors, implying a sample complexity of $d^2 + Kd$ measurements across all users, and therefore an average of $d + d^2/K$ measurements per user. If $K = \Omega(d)$, this is equivalent to $\widetilde{O}(d)$ measurements per user, which corresponds to the parametric rate required per user in order to estimate their pseudo-ideal point $\boldsymbol{v}_k$. Similar to the case of unquantized measurements, the $O(d^2)$ sample cost of estimating the metric from noisy one-bit comparisons has been amortized across all users, demonstrating the benefit of learning multiple user preferences simultaneously when the users share a common metric.

### 3.1 Low-rank modeling

In many settings, the metric $\boldsymbol{M}$ may be low-rank with rank $r < d$ [8, 12]. In this case, $\boldsymbol{M}$ only has $dr$ degrees of freedom rather than $d^2$ degrees as in the full-rank case. Therefore if $K = \Omega(d)$, we intuitively expect the sample cost of learning the metric to be amortized to a cost of $O(r)$

---

[4]We restrict ourselves to the case where the loss is a function of $y_p^{(k)}\left(\|\boldsymbol{u}_k - \boldsymbol{x}_i\|_M^2 - \|\boldsymbol{u}_k - \boldsymbol{x}_j\|_M^2\right)$.

[5]Specifically we may choose $\lambda_v = 2\lambda_F\lambda_u$, resulting in a constraint set containing the solution to (4). To see this, let $\boldsymbol{M}^*, \{\boldsymbol{u}_k^*\}_{k=1}^K$ be the solution to (4) and define $\boldsymbol{v}_k^* = -2\boldsymbol{M}^*\boldsymbol{u}_k^*$. Then $\|\boldsymbol{v}_k^*\|_2 = 2\|\boldsymbol{M}^*\boldsymbol{u}_k^*\|_2 \leq 2\|\boldsymbol{M}^*\|\|\boldsymbol{u}_k^*\|_2 \leq 2\|\boldsymbol{M}^*\|_F\|\boldsymbol{u}_k^*\|_2 \leq 2\lambda_F\lambda_u$.

measurements per user. Furthermore, as each $\boldsymbol{v}_k$ is contained in the $r$-dimensional column space of $\boldsymbol{M}$, we also expect a sample complexity of $O(r)$ to learn each user's pseudo-ideal point. Hence, we expect the amortized sample cost per user to be $O(r)$ in the low-rank setting, which can be a significant improvement over $O(d)$ in the full-rank setting when $r \ll d$.

Algorithmically, ideally one would constrain the $\widehat{\boldsymbol{M}}$ and $\{\widehat{\boldsymbol{v}}_k\}_{k=1}^K$ that minimize the empirical risk such that $\mathrm{rank}(\boldsymbol{M}) = r$ and $\boldsymbol{v}_k \in \mathrm{colsp}(\boldsymbol{M})$; unfortunately, such constraints are not convex. Towards a convex algorithm, note that since $\boldsymbol{v}_k \in \mathrm{colsp}(\boldsymbol{M})$, $\mathrm{rank}([\,\boldsymbol{M}, \boldsymbol{v}_1, \cdots, \boldsymbol{v}_K\,]) = \mathrm{rank}(\boldsymbol{M}) = r$. Thus, it is sufficient to constrain the rank of $[\,\boldsymbol{M}, \boldsymbol{v}_1, \cdots, \boldsymbol{v}_K\,]$. We relax this constraint to a convex constraint on the nuclear norm $\|[\,\boldsymbol{M}\ \boldsymbol{v}_1\ \cdots\ \boldsymbol{v}_K\,]\|_*$, and solve a similar optimization problem to eq. (5):

$$\min_{\boldsymbol{M}, \{\boldsymbol{v}_k\}_{k=1}^K} \widehat{R}(\boldsymbol{M}, \{\boldsymbol{v}_k\}_{k=1}^K) \ \text{s.t.}\ \boldsymbol{M} \succeq 0, \|[\,\boldsymbol{M}\quad \boldsymbol{v}_1\quad \cdots\quad \boldsymbol{v}_K\,]\|_* \le \lambda_*, |\delta_{i,j}^{(k)}| \le \gamma\ \forall\, i, j, k. \quad (7)$$

We again let $\boldsymbol{M}^*$ and $\{\boldsymbol{v}_k^*\}_{k=1}^K$ minimize the true risk $R(\boldsymbol{M}, \{\boldsymbol{v}_k\}_{k=1}^K)$, subject to the same constraints. The following theorem bounds the excess risk over this constraint set:

**Theorem 3.3.** *Suppose* $\|\boldsymbol{x}_i\|_2 \le 1$ *for all* $i \in [n]$*. With probability at least* $1 - \delta$,

$$R(\widehat{\boldsymbol{M}}, \{\widehat{\boldsymbol{v}}_k\}_{k=1}^K) - R(\boldsymbol{M}^*, \{\boldsymbol{v}_k^*\}_{k=1}^K) \le 2L \sqrt{\frac{2\lambda_*^2 \log(2d+K)}{|\mathcal{S}|} \left[\left(8 + \frac{4\min(d,n)}{K}\right)\frac{\|\boldsymbol{X}\|^2}{n} + \frac{16}{\sqrt{K}}\right]}$$

$$+ \frac{8L\lambda_*}{3|\mathcal{S}|}\log(2d+K) + \sqrt{\frac{8L^2\gamma^2 \log(2/\delta)}{|\mathcal{S}|}}.$$

To put this result in context, suppose that the items $\boldsymbol{x}_i$ and ideal points $\boldsymbol{u}_k$ are sampled i.i.d. from $\mathcal{N}(\boldsymbol{0}, \frac{1}{d}\boldsymbol{I})$. With this item distribution it is straightforward to show that with high probability, $\|\boldsymbol{X}\|^2 = O(\frac{n}{d})$ (see [21]). For a given $r < d$ let $\boldsymbol{M} = \frac{d}{\sqrt{r}}\boldsymbol{L}\boldsymbol{L}^T$, where $\boldsymbol{L}$ is a $d \times r$ matrix with orthonormal columns sampled uniformly from the Grassmanian. With this choice of scaling we have $\|\boldsymbol{M}\|_F = d$, so that each element of $\boldsymbol{M}$ is dimensionless on average. Furthermore, recalling that $\boldsymbol{v}_k = -2\boldsymbol{M}\boldsymbol{u}_k$, with this choice of scaling $\mathbb{E}[\|\boldsymbol{v}_k\|_2^2] \propto d$ and so each entry of $\boldsymbol{v}_k$ on average is dimensionless. To choose a setting for $\lambda^*$ recall that $[\boldsymbol{M}, \boldsymbol{v}_1, \ldots \boldsymbol{v}_K]$ has rank $r$ and therefore

$$\|[\,\boldsymbol{M}\quad \boldsymbol{v}_1\quad \cdots\quad \boldsymbol{v}_K\,]\|_* \le \sqrt{r}\|[\,\boldsymbol{M}\quad \boldsymbol{v}_1\quad \cdots\quad \boldsymbol{v}_K\,]\|_F \le \sqrt{r(d^2 + K\max_{k\in[K]}\|\boldsymbol{v}_k\|_2^2)},$$

which one can show is $O(\sqrt{r(d^2 + dK\log K)})$ with high probability and so we set $\lambda_* = O\left(\sqrt{r(d^2 + dK\log K)}\right)$. With these term scalings, we have the following corollary:

**Corollary 3.3.1.** *Let* $n \ge d$, $\boldsymbol{x}_i, \boldsymbol{u}_k \sim \mathcal{N}(\boldsymbol{0}, \frac{1}{d}\boldsymbol{I})$ *and* $\boldsymbol{M} = \frac{d}{\sqrt{r}}\boldsymbol{L}\boldsymbol{L}^T$*, where* $\boldsymbol{L}$ *is a* $d \times r$ *matrix with orthonormal columns. If* $K = \Omega(d^2)$*, then in the same setting as Theorem 3.3 with high probability*

$$R(\widehat{\boldsymbol{M}}, \{\widehat{\boldsymbol{v}}_k\}_{k=1}^K) - R(\boldsymbol{M}^*, \{\boldsymbol{v}_k^*\}_{k=1}^K) = \widetilde{O}\left(\sqrt{\frac{dr + Kr}{|\mathcal{S}|}}\right).$$

*Remark* 3.4. The scaling $|\mathcal{S}| = O(dr + Kr)$ matches our intuition that $O(dr)$ collective measurements should be made across all users to account for the $dr$ degrees of freedom in $\boldsymbol{M}$, in addition to $O(r)$ measurements per user to resolve their own pseudo-ideal point's $r$ degrees of freedom. If $K = \Omega(d)$, then each user answering $O(r)$ queries is sufficient to amortize the cost of learning the metric with the same order of measurements per user as is required for their ideal point. Although Corollary 3.3.1 requires the even stronger condition that $K = \Omega(d^2)$, we believe this is an artifact of our analysis and that $K = \Omega(d)$ should suffice.[6] Even so, a $\Omega(d^2)$ user count scaling might be reasonable in practice since recommender systems typically operate over large populations of users.

---

[6]The required user scaling of $K = \Omega(d^2)$ implies that Corollary 3.3.1 only recovers an amortized scaling of $O(r)$ measurements per user if the user count is very large. Nevertheless, the original statement of Theorem 3.3 *does* apply for any user count, and in this case the only drawback to not invoking $K = \Omega(d^2)$ is that the amortized scaling is larger than $O(r + dr/K)$ measurements per user. We believe this scaling can in fact be tightened to $O(r + dr/K)$ measurements per user for *all* user counts $K$ (not just $K = \Omega(d^2)$), which we leave to future work. See the proof of Corollary 3.3.1 for additional discussion.

# 4 Recovery guarantees

The results in the previous section give guarantees on the generalization error of a learned metric and ideal points when predicting pair responses over $\mathcal{X}$, but do not bound the recovery error of the learned parameters $\widehat{M}, \{\widehat{v}_k\}_{k=1}^K$ with respect to $M^*$ and $\{v_k^*\}_{k=1}^K$. Yet, in some settings such as data generated from human responses [22,23] it may be reasonable to assume that a true $M^*$ and $\{v_k^*\}_{k=1}^K$ do exist that generate the observed data (rather than serving only as a model) and that practitioners may wish to estimate and interpret these latent variables, in which case accurate recovery is critical. Unfortunately, for an arbitrary noise model and loss function, recovering $M^*$ and $\{v_k^*\}_{k=1}^K$ exactly is generally impossible if the model is not identifiable. However, we now show that with a small amount of additional structure, one can ensure that $\widehat{M}$ and $\{\widehat{v}_k\}_{k=1}^K$ accurately approximate $M^*$ and $\{v_k^*\}_{k=1}^K$ if a sufficient number of one-bit comparisons are collected.

We assume a model akin to that of [12] for the case of triplet metric learning. Let $f: \mathbb{R} \to [0,1]$ be a strictly monotonically increasing *link function* satisfying $f(x) = 1 - f(-x)$; for example, $f(x) = (1 + e^{-x})^{-1}$ is the logistic link and $f(x) = \Phi(x)$ is the probit link where $\Phi(\cdot)$ denotes the CDF of a standard normal distribution. Defining $\delta_p(M, v) := x_i^T M x_i - x_j^T M x_j + v^T(x_i - x_j)$ for $p = (i,j)$, we assume that $\mathbb{P}(y_p^{(k)} = -1) = f\left(-\delta_p(M^*, v_k^*)\right)$ for some $M^* \succeq 0$ and $v_k^* \in \text{colsp}(M^*)$. This naturally reflects the idea that some queries are easier to answer (and thus less noisy) than others. For instance, if $\delta_{ij}^{(k)} \ll 0$ such as may occur when $x_i$ very nearly equals user $k$'s ideal point, we may assume that user $k$ almost always prefers item $i$ to $j$ and so $f(-\delta_{ij}^{(k)}) \to 1$ (since $f$ is monotonic). Furthermore, we assume that eq. (5) is optimized with the *negative log-likelihood* loss $\ell_f$ induced by $f$: $\ell_f(y_p, p; M, v) := -\log(f(y_p \delta_p(M, v_k)))$. In Appendix E, we show that we may lower bound the excess risk of $\widehat{M}, \{\widehat{v}_k\}_{k=1}^K$ by the squared error between the *unquantized* measurements corresponding to $\widehat{M}, \{\widehat{v}_k\}_{k=1}^K$ and $M^*, \{v_k^*\}_{k=1}^K$. We then utilize tools from Section 2 combined with the results in Section 3 to arrive at the following recovery guarantee.

**Theorem 4.1.** *Fix a strictly monotonic link function $f$ satisfying $f(x) = 1 - f(-x)$. Suppose for a given item set $\mathcal{X}$ with $n \geq D + d + 1$ and $\|x_i\| \leq 1 \, \forall i \in [n]$ that the pairs and users in dataset $\mathcal{S}$ are sampled independently and uniformly at random, and that user responses are sampled according to $\mathbb{P}(y_p^{(k)} = -1) = f\left(-\delta_p(M^*, v_k^*)\right)$ where $M^*, \{v_k^*\}_{k=1}^K$ satisfy the constraints in (5). Let $\widehat{M}, \{\widehat{v}_k\}_{k=1}^K$ be the solution to (5) solved using loss $\ell_f$. Then with probability at least $1 - \delta$,*

$$\frac{1}{n} \sigma_{\min}\left(J[X_{\otimes}^T, X^T]\right)^2 \left(\|\widehat{M} - M^*\|_F^2 + \frac{1}{K} \sum_{k=1}^K \|\widehat{v}_k - v_k^*\|^2\right) \leq$$

$$\frac{4}{C_f^2} \sqrt{\frac{L^2(\lambda_F^2 + K\lambda_v^2)}{|\mathcal{S}|}} \log(d^2 + d + 1) + \frac{\sqrt{8L^2(\lambda_F^2 + K\lambda_v^2)}}{3C_f^2|\mathcal{S}|} \log(d^2 + d + 1) + \frac{1}{C_f^2} \sqrt{\frac{L^2\gamma^2 \log(\frac{2}{\delta})}{2|\mathcal{S}|}},$$

*where $C_f = \min_{z: |z| \leq \gamma} f'(z)$ and $J := I_n - \frac{1}{n}\mathbf{1}_n \mathbf{1}_n^T$ is the centering matrix. Furthermore, if $\mathcal{X}$ is constructed by sampling each item i.i.d. from a distribution $p_X$ with support on the unit ball that is absolutely continuous with respect to the Lebesgue measure, then with probability 1, $\sigma_{\min}\left(J[X_{\otimes}^T, X^T]\right) > 0$.*

*Remark* 4.2. The key conclusion from this result is that since $\sigma_{\min}\left(J[X_{\otimes}^T, X^T]\right) > 0$ almost surely, the recovery error of $\widehat{M}, \{\widehat{v}_k\}_{k=1}^K$ with respect to $M^*, \{v_k^*\}_{k=1}^K$ is upper bounded by a decreasing function of the number of data points, $|\mathcal{S}|$. In other words, the metric and pseudo-ideal points are identifiable from one-bit paired comparisons under an assumed response distribution. As discussed in Section 2, the ideal points $\{u_k^*\}_{k=1}^K$ are then also identifiable as long as $M^*$ is full-rank. We present an analogous result for the case of a low-rank metric in Appendix E, and leave to future work a study of the scaling of $\sigma_{\min}\left(J[X_{\otimes}^T, X^T]\right)$ with respect to $d$ and $n$.

# 5 Experimental results

Below we analyze the performance of the empirical risk minimizers given in eqs. (5) and (7) on both simulated and real-world data,[7] with further details deferred to Appendix F.

---

[7]Code available at `https://github.com/gregcanal/multiuser-metric-preference`

**Simulated experiments:** We first simulate data in a similar setting to Cor. 3.3.1 where $x_i, u_k \sim \mathcal{N}(0, \frac{1}{d}I)$ and $M^* = \frac{d}{\sqrt{r}}LL^T$ where $L \in \mathbb{R}^{d \times r}$ is a random orthogonal matrix. To construct the training dataset, we query a fixed number of randomly selected pairs per user and evaluate prediction accuracy on a held-out test set, where all responses are generated according to a logistic link function (injecting response noise). We evaluate the prediction accuracy of the Frobenius norm regularized optimization in eq. (5) (referred to as **Frobenius metric**), designed for full-rank matrix recovery, as well as the nuclear norm regularized optimization in eq. (7) (referred to as **Nuclear full**), designed for low-rank metrics. We also compare against several ablation methods that modify the constraint sets in (5) and (7): **Nuclear metric**, where $\|M\|_*$ and $\|v_k\|_2$ are constrained; **Nuclear split**, where $\|M\|_*$ and $\|[v_1, \cdots, v_K]\|_*$ are constrained; and **PSD only**, where only $M \succeq 0$ is enforced. We also compare against **Nuclear full, single**, which is equivalent to **Nuclear full** when applied *separately* to each user (learning a unique metric and ideal point), where test accuracy is averaged over all users. To compare performance under a best-case hyperparameter setting, we tune each method's respective constraints using oracle knowledge of $M^*$ and $\{u_k^*\}_{k=1}^K$. Finally, we also evaluate prediction accuracy when the ground-truth parameters are known exactly (i.e., $M = M^*, v_k = -2M^*u_k^*$), which we call **Oracle**. This accuracy gives the "best case" performance, and reflects the inherent response noise in the generated data.

To test a low-rank setting, we set $d = 10$, $r = 1$, $n = 100$, and $K = 10$. We observe that **Nuclear full** outperforms the baseline methods in terms of test accuracy, and is closely followed by **Nuclear split** (Fig. 1a). Interestingly **Nuclear metric**, which also enforces a nuclear norm constraint on $M$, does not perform as well, possibly because it does not encourage the pseudo-ideal points to lie in the same low-rank subspace. While **Nuclear metric** does demonstrate slightly improved metric recovery (Fig. 1b), **Nuclear full** and **Nuclear split** recover higher quality metrics for lower query counts (which is the typical operating regime for human-in-the-loop systems) and exhibit significantly better ideal point recovery (Fig. 1c), illustrating the importance of proper subspace alignment between the pseudo-ideal points. To this end, unlike **Nuclear split**, **Nuclear full** explicitly encourages the pseudo-ideal points to align with the column space of $M$, which may explain its slight advantage. Finally, we note that **Nuclear full, single** results in the worst prediction accuracy, demonstrating the benefit of a common metric model when the underlying metric is in fact shared. While the single user case is not the focus of this work, in Appendix F.4 we compare the performance of **Nuclear full, single** against the methods proposed in [7], which only considers the single user case.

**Color dataset:** We also study the performance of our model on a dataset of pairwise color preferences across multiple respondents ($K = 48$) [24]. In this setting, each color ($n = 37$) is represented as a 3-dimensional vector in CIELAB color space (lightness, red vs. green, blue vs. yellow), which was designed as a uniform space for how humans perceive color [25]. Each respondent was asked to order pairs of color by preference, as described in [26, Sec. 3.1]. Since all $2\binom{37}{2}$ possible pairs (including each pair reversal) were queried for each respondent, we may simulate random pair sampling exactly.

As there are only $d = 3$ features, we constrain the Frobenius norm of the metric and optimize eq. (5) using the hinge loss. Varying the number of pairs queried per user, we plot prediction accuracy on a held-out test set (**Learned M, crowd** in Fig. 1d). As CIELAB is designed to be perceptually uniform, we compare against a solution to eq. (5) that *fixes* $M = I$ and only learns the points $\{v_k\}_{k=1}^{48}$ (**Identity M** in Fig. 1d). This method leads to markedly lower prediction accuracy than simultaneously learning the metric and ideal points; this result suggests that although people's *perception* of color is uniform in this space, their *preferences* are not. We also compare against a baseline that solves the same optimization as eq. (5) *separately* for each individual respondent (learning a unique metric and ideal point per user), with prediction accuracy averaged over all respondents (**Learned M, single** in Fig. 1d). Although learning individual metrics appears to result in better prediction after many queries, in the low-query regime ($< 20$ pairs per user) learning a common metric across all users results in slightly improved performance (see Appendix F for zoomed plot). As $d = 3$ is small relative to the number of queries given to each user, the success of individual metric learning is not unexpected; however, collecting $O(d^2)$ samples per user is generally infeasible for larger $d$ unlike collective metric learning which benefits from crowd amortization. Finally, learning a single metric common to all users allows for insights into the crowd's general measure of color similarity. As can be seen in Fig. 1e, the learned metric is dominated by the "lightness" feature, indicating that people's preferences correspond most strongly to a color's lightness. As an external validation, this is consistent with the findings of Fig. 1 of [24].

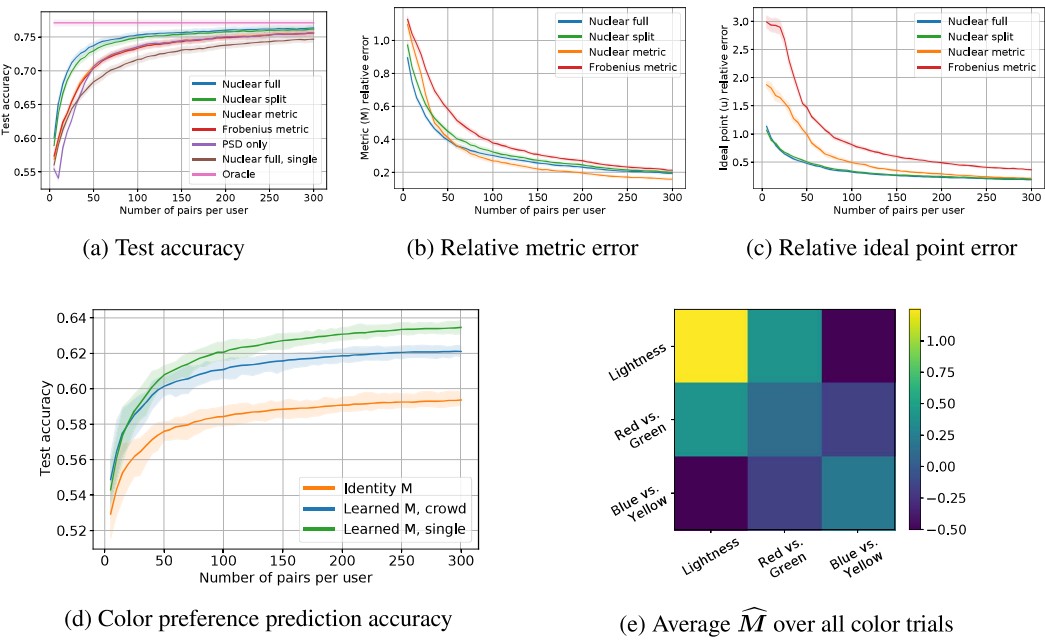

(a) Test accuracy     (b) Relative metric error     (c) Relative ideal point error

(d) Color preference prediction accuracy     (e) Average $\widehat{\boldsymbol{M}}$ over all color trials

Figure 1: (a-c) Normally distributed items with $d = 10$, $r = 1$, $n = 100$ and $K = 10$. Error bars indicate $\pm 1$ standard error about the sample mean. For visual clarity, **PSD only** and **Nuclear full, single** baselines are omitted from (b-c) due to poor performance. (d) Average color preference prediction accuracy, where error bars indicate 2.5% and 97.5% percentiles. (e) Estimated color preference metric. For (a-e), random train/test splitting was repeated over 30 trials.

## 6 Discussion

The main contribution of this work is a model for multi-user simultaneous metric and preference learning consisting of a shared metric that captures the crowd's common perceptions between items, as well as user-specific ideal points that characterize individual preferences. Our core result is that when querying paired comparisons over a large number of users, the sample cost of metric learning is distributed over the crowd, with the total number of queries per user scaling with the rank of the underlying metric.

One interesting avenue for future work is to study more flexible metric models, since as with any Mahalanobis metric model it may not be the case that linear weightings of quadratic feature combinations provide enough flexibility to adequately model certain preference judgements. While it may be possible to generalize the linear metric results studied here to a more general Hilbert space, another avenue to increase model flexibility is to acquire a richer set of features in the item set, which would typically involve increasing the ambient dimension. In this setting, an important consideration would be to ameliorate the requirement that the number of items scales as $\Omega(d^2)$. While this condition is necessary for identifiability if the metric is full-rank, we believe that the item count should only need to scale as $\Omega(dr)$ if the metric has rank $r < d$ (see Appendix C.3 for additional discussion). Furthermore, in the case of metric learning from triplet queries it is known that only $\Omega(r)$ items are required for metric recovery in the low-rank case [12]. This suggests a potential strategy of supplementing paired comparisons with triplet queries to reduce the number of required items.

## Acknowledgements and Disclosure of Funding

We thank the reviewers for their feedback, as well as Karen Schloss for providing the color preference data used in our experiments and for useful discussions and references about color preference learning. This work was partially supported by AFOSR/AFRL grant FA9550-18-1-0166 and ARMY MURI grant W911NF-15-1-0479.

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
