# One for All: Simultaneous Metric and Preference Learning over Multiple Users

**Gregory Canal**
University of Wisconsin-Madison
Madison, WI
gcanal@wisc.edu

**Blake Mason**
Rice University
Houston, TX
bm63@rice.edu

**Ramya Korlakai Vinayak**
University of Wisconsin-Madison
Madison, WI
ramya@ece.wisc.edu

**Robert Nowak**
University of Wisconsin-Madison
Madison, WI
rdnowak@wisc.edu

## Abstract

This paper investigates simultaneous preference and metric learning from a crowd of respondents. A set of items represented by $d$-dimensional feature vectors and paired comparisons of the form "item $i$ is preferable to item $j$" made by each user is given. Our model jointly learns a distance metric that characterizes the crowd's general measure of item similarities along with a latent ideal point for each user reflecting their individual preferences. This model has the flexibility to capture individual preferences, while enjoying a metric learning sample cost that is amortized over the crowd. We first study this problem in a noiseless, continuous response setting (i.e., responses equal to differences of item distances) to understand the fundamental limits of learning. Next, we establish prediction error guarantees for noisy, binary measurements such as may be collected from human respondents, and show how the sample complexity improves when the underlying metric is low-rank. Finally, we establish recovery guarantees under assumptions on the response distribution. We demonstrate the performance of our model on both simulated data and on a dataset of color preference judgments across a large number of users.

## 1  Introduction

In many data-driven recommender systems (e.g., streaming services, online retail), multiple users interact with a set of items (e.g., movies, products) that are common to all users. While each user has their individual preferences over these items, there may exist shared structure in how users *perceive* items when making preference judgments. This is a reasonable assumption, since collections of users typically have shared perceptions of similarity between items regardless of their individual item preferences [1–3]. In this work we develop and analyze models and algorithms for simultaneously learning individual preferences and the common metric by which users make preference judgments.

Specifically, suppose there exists a known, fixed set $\mathcal{X}$ of $n$ items, where each item $i \in 1, \ldots, n$ is parameterized by a feature vector $\boldsymbol{x}_i \in \mathbb{R}^d$. We model the crowd's preference judgments between items as corresponding to a common Mahalanobis distance metric $d_{\boldsymbol{M}}(\boldsymbol{x}, \boldsymbol{y}) = \|\boldsymbol{x} - \boldsymbol{y}\|_{\boldsymbol{M}}$, where $\|\boldsymbol{x}\|_{\boldsymbol{M}} \coloneqq \sqrt{\boldsymbol{x}^T \boldsymbol{M} \boldsymbol{x}}$ and $\boldsymbol{

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

# A  Broader impacts

With the deployment of the ideal point model with a learned metric comes all of the challenges, impacts, and considerations associated with preference learning and recommender systems, such as if the deployed recommender system produces preference estimates and item recommendations that are aligned with the values and goals of the users and society as a whole, and if the item features are selected in a way that is diverse enough to adequately model all users and items. Therefore, we limit our broader impacts discussion to the challenges specific to our model.

As with any statistical model, the utility of our common metric ideal point model is limited by the accuracy to which it captures the patterns observed in the response data. The most immediate question about our model is the appropriateness of the assumption that a single metric is shared among all users. Such a model can prove useful in practice since it directly allows for shared structure between users and shared information between their measurements, and furthermore allows for a direct interpretation of preference at the crowd level, as we demonstrate with color preference data in Section 5. However, with this model comes the implicit assumption that users are homogeneous in their preference judgements. In many recommender systems the assumption of a common preference metric will likely be violated, due to heterogeneous user bases and subpopulations of users.

In general, there are almost certainly individual differences between each user's notion of item similarity and preference, and hence recovery of a single crowd metric should not necessarily be taken to mean that it describes each individual exactly. Although our model does provide a degree of individual flexibility through its use of ideal points (rather than treating the entire crowd's responses as coming from a single user), the result of a common metric violation may be that the learned population metric will fit to the behavior of the majority, or may fail to capture some aspect of each individual user's preference judgements. Our model could certainly be applied individually to each user, such that they each learn their own metric: however, as we demonstrate in our theoretical and empirical results, there is a fundamental sample complexity tradeoff in that to learn individual metrics, many more samples are needed per user, unlike the case of a common metric model where measurements are amortized.

The impacts of such a mismatch on an individual or subpopulation can range from inconvenient, such as in getting poor recommendations for online shopping or streaming services, to actively harmful, such as in receiving poor recommendations for a major decision (e.g., medical) that would otherwise suit the population majority. To prevent such cases, before deploying a common metric model it is important to not only average performance across the entire population (which will reflect the majority), but also evaluate worst-case performance on any given user or subpopulation. Such considerations are especially important if the common metric is not only used for predicting preferences between items, but also used to make inferences about a population by directly examining the metric entries (as we demonstrate for color preferences in Section 5). If the metric only applies to the majority or the population in the aggregate, then such inferences about feature preferences may not be accurate for individual users or subpopulations.

Beyond considering the effects of skewed modeling of individual users or subpopulations, it is important to consider potentially harmful effects of a common metric preference model when arriving at *item rankings*. While the item set examples discussed here include non-human objects such as movies and products, more generally the term "item" may be used in an abstract sense to include people, such as when building recommender systems for an admissions or hiring committee to select between job or school applicants (see [7] for a preference learning example on graduate school admissions data). In this context, the "users" may be a separate population (such as an admissions committee) that is making preference judgements about individual candidates (i.e., the "items"). In such cases, it is critical that extra precautions be taken and considerations be made for any possible biases that may be present across the population of users and reflected in the common metric. For example, if a majority of an admissions committee shared certain implicit biases when making preference decisions between candidates, such biases may be *learned* in a common metric. On the other hand, the existence of a common metric potentially allows for interpretation and insight into the features by which a committee is making its decisions, possibly allowing for intervention and bias mitigation if it is observed that the committee population is sharing a bias with regard to certain candidate features. While our model could be applied to each individual to avoid the challenges described above, there is a fundamental tradeoff in the sample complexity cost per user required to obtain individual models, which our results elucidate.

# B  Related work

Metric learning has received considerable attention and classical techniques are nicely summarized in the monographs [8, 27]. Efficient algorithms exist for a variety of data sources such as class labels [10, 11] and triplet comparisons [12]. Classical metric learning techniques focus on learning linear (Mahalanobis) metrics parametrized by a positive (semi-)definite matrix. In the case of learning linear metrics from triplet observations, [12, 28] establish tight generalization error guarantees. In practice, to handle increasingly complex learning problems, it is common to leverage more expressive, nonlinear metrics that are parametrized by kernels or deep neural networks and we refer the reader to [13–15] for a survey of nonlinear metric learning techniques. The core idea of many kernelized metric learning algorithms is that one can use Kernelized-PCA to reduce the nonlinear metric learning problem over $n$ items to learning a linear metric in $\mathbb{R}^n$ via a kernel trick on the empirical Gram matrix. The downside to this approach is that the learned metric need not apply to new items other than those contained in the original $n$.

To circumvent this issue, works such as [16] have proposed deep metric learning. Intuitively, in the linear case one may factor a metric $M$ as $M = LL^T$ and could instead learn a matrix $L \in \mathbb{R}^{d \times r}$. In the case of deep metric learning, the same principle applies except that $L$ is replaced with a map $\mathcal{L} : \mathbb{R}^d \to \mathbb{R}^r$ given by a deep neural network such that the final metric is $d(x, y) = \|\mathcal{L}(x) - \mathcal{L}(y)\|_2$. While the theory of nonlinear metric learning is less mature, [29, 30] provide generalization guarantees for deep metric learning using neural tangent kernel and Rademacher analyses respectively. Finally, metric learning is a closely related to the problem of ordinal embedding: [17, 31, 32] propose active sampling techniques for ordinal embedding whereas [33] establishes learning guarantees for passive algorithms.

Preference learning from paired comparisons is a well-studied problem spanning machine learning, psychology, and social sciences, and we refer the reader to [9] for a comprehensive summary of approaches and problem statements. Researchers have proposed a multitude of models ranging from classical techniques such as the Bradley-Terry model [34, 35], Plackett-Luce model [36, 37], and Thurstone model [38] to more modern approaches such as preference learning via Siamese networks [39] to fit the myriad of tailored applications of preference learning. In the linear setting, [7, 40–42] among others propose passive learning algorithms whereas [5, 6, 17–19, 43] propose adaptive sampling procedures. [20] perform localization from paired comparisons, and [44] employ a Gaussian process approach for learning pairwise preferences from multiple users.

The ideal point model we consider here is well-established in the single user setting [4–7], and differs fundamentally from models of preference based only on inner products. In the inner product or "attribute" model of preference, items with larger values of $\langle u, x \rangle$ are preferred, in which case an item's preference can be modified simply by scaling its magnitude. On the other hand, the ideal point (i.e., distance-based) model accounts for item magnitude, and it has been demonstrated that distance-based models can more accurately model user preferences [45].

# C  Proofs and additional results for identifiability from unquantized measurements

## C.1  Properties of selection matrices

In this section, we present several theoretical properties of selection matrices (see Definition 2.1) that will be useful for proving the results that follow. We begin with a lemma upper bounding the rank of selection matrices:

**Lemma C.1.** *Let $n \geq 2$. For any $m \times n$ selection matrix $S$, $\mathrm{rank}(S) \leq \min(m, n - 1)$.*

*Proof.* Since $S$ has $m$ rows, $\mathrm{rank}(S) \leq m$. By construction, for any selection matrix $S$ note that $\mathbf{1}_n \in \ker(S)$, where $\mathbf{1}_n$ is the vector of all ones in $\mathbb{R}^n$. To see this, for any $z \in \mathbb{R}^n$ the $i$th element of $Sz$ is given by $z[p_i] - z[q_i]$, and so the $i$th element of $S\mathbf{1}_n$ is $1 - 1 = 0$ and hence $Sz = 0$. Therefore, $\dim(\ker(S)) \geq 1$ and so $\mathrm{rank}(S) \leq n - 1$. $\qquad\square$

We use this result to show a property of full row rank selection matrices that will be useful for their construction and analysis:

**Lemma C.2.** *Let $S$ be an $m \times n$ selection matrix with $n \geq m + 1$, where for each $i \in [m]$ the nonzero indices of the ith row are given by distinct $p_i, q_i \in [n]$ such that $S[i, p_i] = 1$, $S[i, q_i] = -1$. If $\operatorname{rank}(S) = m$, then for every subset $I \subseteq [m]$ of row indices, there exists $i^* \in I$ such that $S[j, p_{i^*}] = 0$ for all $j \in I \setminus \{i^*\}$, or $S[j, q_{i^*}] = 0$ for all $j \in I \setminus \{i^*\}$.*

*Proof.* Let $I \subseteq [m]$ be given, and suppose by contradiction that no such $i^*$ exists, i.e., no measurement in $I$ introduces a new item unseen by any other measurements in $I$. Let $S^{(I)}$ be the $|I| \times n$ selection matrix consisting of the rows in $S$ listed in $I$. Since $S$ is full row rank, its rows are linearly independent, implying that the rows in $S^{(I)}$ are also linearly independent and therefore $S^{(I)}$ has rank $|I|$.

Let $c \leq n$ be the number of columns that $S^{(I)}$ is supported on (i.e., have at least one nonzero entry). By our contradictory assumption, every item measured in $S^{(I)}$ is measured in at least two rows in $I$, and therefore each of these $c$ columns must have at least 2 nonzero entries. This implies that $S^{(I)}$ has at least $2c$ nonzero entries in total. Since each measurement adds exactly 2 nonzero entries to $S^{(I)}$, this means that there are least $2c/2 = c$ measurements and so $|I| \geq c$.

Now consider the $|I| \times c$ matrix $\widetilde{S}^{(I)}$ consisting of $S^{(I)}$ with its zero columns removed. $\operatorname{rank}(\widetilde{S}^{(I)}) = \operatorname{rank}(S^{(I)})$ since $\widetilde{S}^{(I)}$ and $S^{(I)}$ have the same column space. Since $\widetilde{S}^{(I)}$ is itself a $|I| \times c$ selection matrix, we know from Lemma C.1 that $\operatorname{rank}(\widetilde{S}^{(I)}) \leq \min(|I|, c - 1)$. Since we know $|I| \geq c$, $\min(|I|, c - 1) = c - 1$, implying $\operatorname{rank}(\widetilde{S}^{(I)}) \leq c - 1$. But this is a contradiction since we already know $\operatorname{rank}(\widetilde{S}^{(I)}) = \operatorname{rank}(S^{(I)}) = |I| \geq c$. $\qquad\square$

Intuitively, Lemma C.2 says that if $S$ is full row rank, then every subset of rows contains a row that is supported on a column that is zero for all other rows in the subset, i.e., at least one row measures a new item unmeasured by any other row in the subset. This property is related to a selection matrix being incremental (see Definition 2.2) as follows:

**Lemma C.3.** *Let $S$ be an $m \times n$ selection matrix with $n \geq m + 1$, where for each $i \in [m]$ the nonzero indices of the ith row are given by distinct $p_i, q_i \in [n]$ such that $S[i, p_i] = 1$, $S[i, q_i] = -1$. Suppose for every subset $I \subseteq [m]$ of row indices, there exists $i^* \in I$ such that $S[j, p_{i^*}] = 0$ for all $j \in I \setminus \{i^*\}$, or $S[j, q_{i^*}] = 0$ for all $j \in I \setminus \{i^*\}$. Then there exists an $m \times m$ permutation matrix $P$ such that $PS$ is incremental.*

*Proof.* We will construct a sequence of row indices such that permuting the rows of $S$ in the sequence order results in an incremental matrix. Let $I_m := [m]$. By assumption, there exists an index $i_m \in I_m$ such that $S[j, p_{i_m}] = 0$ for all $j \in [m] \setminus \{i_m\}$ or $S[j, q_{i_m}] = 0$ for all $j \in [m] \setminus \{i_m\}$. Now let $1 < m' \leq m$ be given, and suppose by induction that there exists a set of distinct indices $\{i_k\}_{k=m'}^{m}$ such that for all $m' \leq k \leq m$, $S[j, p_{i_k}] = 0$ for all $j \in [m] \setminus \{i_\ell\}_{\ell=k}^{m}$ or $S[j, q_{i_k}] = 0$ for all $j \in [m] \setminus \{i_\ell\}_{\ell=k}^{m}$ (we have shown the case of $m' = m$ above). Let $I_{m'-1} := [m] \setminus \{i_k\}_{k=m'}^{m}$. Then by assumption, there exists an index $i_{m'-1} \in I_{m'-1}$ such that $S[j, p_{i_{m'-1}}] = 0$ for all $j \in [m] \setminus \{i_k\}_{k=m'-1}^{m}$ or $S[j, q_{i_{m'-1}}] = 0$ for all $j \in [m] \setminus \{i_k\}_{k=m'-1}^{m}$. Therefore, combined with the fact that $i_{m'-1} \in [m] \setminus \{i_k\}_{k=m'}^{m}$ along with the inductive assumption on $\{i_k\}_{k=m'}^{m}$, $\{i_k\}_{k=m'-1}^{m}$ constitutes an index set where for all $m' - 1 \leq k \leq m$, $S[j, p_{i_k}] = 0$ for all $j \in [m] \setminus \{i_\ell\}_{\ell=k}^{m}$ or $S[j, q_{i_k}] = 0$ for all $j \in [m] \setminus \{i_\ell\}_{\ell=k}^{m}$.

Taking $m' = 2$, we have proved by induction the existence of an index set $\{i_1, \ldots, i_m\}$ that is a permutation of $[m]$ such that for any $k \in [m]$, $S[j, p_{i_k}] = 0$ for all $j \in [m] \setminus \{i_\ell\}_{\ell=k}^{m}$ or $S[j, q_{i_k}] = 0$ for all $j \in [m] \setminus \{i_\ell\}_{\ell=k}^{m}$. By construction, $[m] \setminus \{i_\ell\}_{\ell=k}^{m} = \{i_\ell\}_{\ell=1}^{k-1}$, so equivalently for any $k \in [m]$, $S[i_j, p_{i_k}] = 0$ for all $j < k$ or $S[i_j, q_{i_k}] = 0$ for all $j < k$.

We can then explicitly construct the $m \times m$ permutation matrix $P$ as

$$P[k, \ell] = \begin{cases} 1 & \ell = i_k \\ 0 & \text{otherwise.} \end{cases}$$

Let $S' = PS$, $p'_k = p_{i_k}$ and $q'_k = q_{i_k}$. $p'_k, q'_k$ are the nonzero column indices of the $k$th row in the permuted selection matrix $S'$, since for any $\ell \in [n]$, $S'[k, \ell] = S[i_k, \ell]$. We then have for any $k \in [m]$, $S'[j, p'_k] = S[i_j, p_{i_k}] = 0$ for all $j < k$ or $S'[j, q'_k] = S[i_j, q_{i_k}] = 0$ for all $j < k$, and hence $S'$ is incremental. $\qquad\square$

Furthermore, if a selection matrix $S$ (more specifically, a permutation thereof) is incremental, then it is also full-rank:

**Lemma C.4.** *Let $S$ be an $m \times n$ selection matrix with $n \geq m + 1$, and suppose there exists an $m \times m$ permutation matrix $P$ such that $PS$ is incremental. Then $S$ is full-rank with $\mathrm{rank}(S) = m$.*

*Proof.* Denoting the $i$th row of $PS$ by $s_i$, since $PS$ is incremental for all $i \in [m]$ there exists a $j$ such that $s_i[j] \neq 0$ and $s_\ell[j] = 0$ for all $\ell < i$. Hence, for all $i \in [m]$, $s_i$ does not lie in the span of $\{s_\ell\}_{\ell < i}$. Starting at $i = 2$, this implies that $s_1$ and $s_2$ are linearly independent. Let $m' < m$ be given, and assume by induction that $\{s_\ell\}_{\ell \leq m'}$ are linearly independent. Since by assumption $s_{m'+1}$ does not lie in the span of $\{s_\ell\}_{\ell \leq m'}$, the entire set $\{s_\ell\}_{\ell \leq m'+1}$ is linearly independent. Taking $m' = m - 1$, we have by induction that the rows of $PS$ (i.e., $\{s_\ell\}_{\ell \leq m}$) are linearly independent, and since these rows are just a permutation of the rows in $S$, the $m$ rows in $S$ are also linearly independent and so $\mathrm{rank}(S) = m$. $\qquad\square$

We summarize the above lemmas in the following corollary:

**Corollary C.4.1.** *Let $S$ be an $m \times n$ selection matrix with $n \geq m + 1$, where for each $i \in [m]$ the nonzero indices of the $i$th row are given by distinct $p_i, q_i \in [n]$ such that $S[i, p_i] = 1$, $S[i, q_i] = -1$. Then the following are equivalent:*

(a) $\mathrm{rank}(S) = m$.

(b) *For every subset $I \subseteq [m]$ of row indices, there exists $i^* \in I$ such that $S[j, p_{i^*}] = 0$ for all $j \in I \setminus \{i^*\}$, or $S[j, q_{i^*}] = 0$ for all $j \in I \setminus \{i^*\}$.*

(c) *There exists an $m \times m$ permutation matrix $P$ such that $PS$ is incremental.*

*Proof.* By Lemma C.2, $(a) \implies (b)$. By Lemma C.3, $(b) \implies (c)$. By Lemma C.4, $(c) \implies (a)$. Combining these implications, $(a) \iff (b) \iff (c)$. $\qquad\square$

Another useful corollary lower bounds the number of columns a selection matrix must be supported on, depending on its rank:

**Corollary C.4.2.** *Let $S$ be a rank $r$, $m \times n$ selection matrix with $m \geq r$ and $n \geq r + 1$. Then at least $r + 1$ columns of $S$ have at least one nonzero entry.*

*Proof.* Since $\mathrm{rank}(S) = r$, there exists an index set of $r$ linearly independent rows of $S$, which we denote by $I \subseteq [m]$. Let $S'$ be the $r \times n$ submatrix of $S$ consisting of the rows indexed by $I$: since its rows are linearly independent, $\mathrm{rank}(S') = r$. From Corollary C.4.1, there exists a permutation $P$ of the rows in $S'$ such that $PS'$ is incremental. Since the first row of $PS'$ introduces two items and the remaining $r - 1$ rows each introduce at least one new item, $PS'$ must be supported on at least $2 + (r - 1) = r + 1$ columns. Since the rows in $PS'$ are contained in $S$, $S$ must also be supported on at least $r + 1$ columns. $\qquad\square$

When studying random selection matrices in Appendix C.2, it will be useful to understand a particular graph constructed from the rows of a selection matrix $S$. For $p, q \in [n]$ and $p \neq q$, let $s_{(p,q)}$ denote a vector in $\mathbb{R}^n$ given by

$$s_{(p,q)}[j] = \begin{cases} 1 & j = p \\ -1 & j = q \\ 0 & \text{otherwise.} \end{cases} \qquad (8)$$

Consider a set of $r$ vectors $S := \{s_i\}_{i=1}^r \subset \mathbb{R}^n$ in the form given by eq. (8). We can construct a graph $G_S = (V_S, E_S)$ from this set as follows: $V_S = [r]$ denotes the vertices of this graph (with vertex $i \in [r]$ corresponding to row $s_i$), and $E_S$ denotes the edge set. We define the connectivity of $G_S$ by an $r \times r$ adjacency matrix $A_S$, where

$$A_S[i, j] = \begin{cases} 1 & \exists k \in [n] \text{ s.t. } s_i[k] \neq 0 \wedge s_j[k] \neq 0 \\ 0 & \text{otherwise.} \end{cases}$$

In other words, vectors $s_i$ and $s_j$ are adjacent on $G_S$ if they have overlapping support. We say that vertices $i$ and $j$ are *linked* on $G$ if $A[i, j] = 1$, or if there exists a finite sequence of distinct indices

$\{k_\ell\}_{\ell=1}^T \subseteq [r] \setminus \{i, j\}$ such that $A[i, k_1] = A[k_1, k_2] = \cdots = A[k_{T-1}, k_T] = A[k_T, j] = 1$. Denote the set of linked vertex pairs by

$$C_S = \{(i, j) : i, j \in [r], \, i, j \text{ linked on } G_S\}.$$

We start with a lemma concerning the span of $S$ in how it relates to connectivity on $G_S$:

**Lemma C.5.** *Let* $S := \{s_i\}_{i=1}^r$ *denote a set of linearly independent vectors in* $\mathbb{R}^n$ *in the form eq. (8), with* $n \geq r + 1$. *For given* $p, q \in [n]$ *with* $p \neq q$, *if* $s_{(p,q)} \in \text{span}(\{s_i\}_{i=1}^r)$ *then there exists a linked vertex pair* $(i_p, i_q) \in C_S$ *such that* $s_{i_p}[p] \neq 0$ *and* $s_{i_q}[q] \neq 0$.

*Proof.* If $s_{(p,q)} \in \text{span}(\{s_i\}_{i=1}^r)$, there exist scalars $\{\beta_i\}_{i=1}^r$ not all equal to zero such that $s_{(p,q)} = \sum_{i=1}^r \beta_i s_i$, i.e.,

$$1 = \sum_{i=1}^r \beta_i s_i[p] \tag{9}$$

$$-1 = \sum_{i=1}^r \beta_i s_i[q] \tag{10}$$

$$0 = \sum_{i=1}^r \beta_i s_i[j] \quad j \neq p, q. \tag{11}$$

From eq. (9), we know there exists a $i_p \in [r]$ such that $\beta_{i_p} \neq 0$ and $s_{i_p}[p] \neq 0$: otherwise, $\beta_i s_i[p] = 0$ for all $i \in [r]$ which would result in the summation in eq. (9) being 0. Let $j_1$ denote the other index supported by $s_{i_p}$, i.e., $j_1 \neq p$ and $s_{i_p}[j_1] \neq 0$. If $j_1 = q$, then there trivially exists $i_q = i_p$ such that $s_{i_q}[q] = s_{i_p}[j_1] \neq 0$. Clearly this choice of $(i_p, i_q)$ is linked on $G_S$ since $i_p = i_q$, which would give us the desired result.

Now suppose $j_1 \neq q$. Recalling that $j_1 \neq p$ as well, from eq. (11) we have

$$0 = \sum_{i=1}^r \beta_i s_i[j_1]$$
$$= \underbrace{\beta_{i_p}}_{\neq 0} \underbrace{s_{i_p}[j_1]}_{\neq 0} + \sum_{i \in [r] \setminus \{i_p\}} \beta_i s_i[j_1], \tag{12}$$

which implies that there exists $i_1 \in [r] \setminus \{i_p\}$ such that $\beta_{i_1} \neq 0$ and $s_{i_1}[j_1] \neq 0$; otherwise, $\beta_i s_i[j_1] = 0$ for all $i \in [r] \setminus \{i_p\}$ which would result in a contradiction in eq. (12). Note that $(i_1, i_p)$ are linked on $G_S$, since they are both supported on index $j_1$.

Now, suppose by induction that for a given $1 \leq T \leq r - 1$ there exist distinct vertices $\{i_1 \ldots, i_T\} \in [r] \setminus \{i_p\}$ and distinct item indices $\{j_1, \ldots, j_T\} \in [n] \setminus \{p, q\}$ such that $s_{i_p}[j_1] \neq 0$, $s_{i_k}[j_k] \neq 0$ and $s_{i_k}[j_{k+1}] \neq 0$ for $k < T - 1$, $s_{i_T}[j_T] \neq 0$, $(i_p, i_T)$ are linked on $G_S$, and $\beta_{i_k} \neq 0$ for $k \in [T]$. Above we have shown the existence of such sets for the the base case of $T = 1$.

Let $j_{T+1}$ be the other item index supported on $s_{i_T}$, i.e., $j_{T+1} \neq j_T$ and $s_{i_T}[j_{T+1}] \neq 0$. If $j_{T+1} = q$, then we can set $i_q = i_T$ and we have found an $i_q$ linked to $i_p$ on $G_S$ (since $i_T = i_q$ is linked to $i_p$ on $G_S$ by inductive assumption) and $s_{i_q}[q] = s_{i_T}[j_{T+1}] \neq 0$. Otherwise, since $\{s_{i_k}\}_{k=1}^T \cup \{s_{i_p}\}$ are $T + 1$ linearly independent vectors in the form eq. (8), from Corollary C.4.2 we have that $\{s_{i_k}\}_{k=1}^T \cup \{s_{i_p}\}$ are collectively supported on at least $T + 2$ indices in $[n]$. Hence, if $j_{T+1} \neq q$, then we must have $j_{T+1} \in [n] \setminus (\{p, q\} \cup \{j_k\}_{k=1}^T)$. From eq. (11), we then have

$$0 = \sum_{i=1}^r \beta_i s_i[j_{T+1}]$$
$$= \beta_{i_p} \underbrace{s_{i_p}[j_{T+1}]}_{0} + \sum_{k=1}^{T-1} \beta_{i_k} \underbrace{s_{i_k}[j_{T+1}]}_{0} + \underbrace{\beta_{i_T}}_{\neq 0} \underbrace{s_{i_T}[j_{T+1}]}_{\neq 0} + \sum_{i \in [r] \setminus (i_p \cup \{i_k\}_{k=1}^T)} \beta_i s_i[j_{T+1}],$$
$$= \underbrace{\beta_{i_T}}_{\neq 0} \underbrace{s_{i_T}[j_{T+1}]}_{\neq 0} + \sum_{i \in [r] \setminus (i_p \cup \{i_k\}_{k=1}^T)} \beta_i s_i[j_{T+1}], \tag{13}$$

which implies that there exists $i_{T+1} \in [r] \setminus (i_p \cup \{i_k\}_{k=1}^T)$ such that $\beta_{i_{T+1}} \neq 0$ and $s_{i_{T+1}}[j_{T+1}] \neq 0$; otherwise, $\beta_i s_{i_{T+1}}[j_{T+1}] = 0$ for all $[r] \setminus (i_p \cup \{i_k\}_{k=1}^T)$, which would result in a contradiction in eq. (13). Note that if $T = r - 1$, the existence of such an $i_{T+1}$ is impossible since in that case $[r] \setminus (i_p \cup \{i_k\}_{k=1}^T) = \varnothing$; hence, if $T = r - 1$, it must be the case that $j_{T+1} = q$ as described above.

If $T < r - 1$ and $j_{T+1} \neq q$, then such an $i_{T+1} \in [r] \setminus (i_p \cup \{i_k\}_{k=1}^T)$ exists. Note that $i_{T+1}$ and $i_p$ are linked on $G_S$, since $i_{T+1}$ and $i_T$ share an item (i.e., $j_{T+1}$) and $i_T$ and $i_p$ are linked on $G_S$ by inductive assumption. Hence, we have constructed sets $\{i_k\}_{i=1}^{T+1}$ and $\{j_k\}_{i=1}^{T+1}$ that fulfill the inductive assumption for $T' = T + 1$.

Therefore, there must exist a $1 \leq T^* \leq r - 1$ such that $j_{T^*+1} = q$, in which case we can take $i_q = i_{T^*}$ and thus have identified an $i_q$ that is linked to $i_p$ on $G_S$ (since $i_{T^*} = i_q$ is linked to $i_p$ on $G_S$) and satisfies $s_{i_q}[q] = s_{i_{T^*}}[j_{T^*+1}] \neq 0$. $\qquad\square$

## C.2 Characterizing random selection matrices

In this section, we explore how many measurements and items are required for a randomly constructed selection matrix to have full-rank.

To answer this question, first we establish a fundamental result concerning how many item pairs sampled uniformly at random are required (on average or with high probability) in order for a selection matrix to be of a certain rank. We start by bounding the probability that, for an existing selection matrix $S$ with rank $r$, an additional row $s$ constructed by selecting two items uniformly at random lies within the row space of $S$. This is equivalent to the probability that the concatenation of $s$ with $S$ is a rank $r + 1$ matrix; bounding this probability will then allow us to bound the number of such appended rows needed to increase the rank of $S$ to some desired value greater than $r$.

**Lemma C.6.** *Suppose $S$ is an $m \times n$ selection matrix with rank $r \leq \min(m, n-1)$. Let $s \in \mathbb{R}^n$ be constructed by sampling two integers, $p$ and $q$, uniformly and without replacement from $[n]$ (and statistically independent of $S$) and setting $s = s_{(p,q)}$. Then*

$$\frac{2r}{n(n-1)} \leq \mathbb{P}(s \in \text{rowsp}(S) \mid S) \leq \frac{(r+1)r}{n(n-1)}. \tag{14}$$

We defer the proof of Lemma C.6 to the end of the section.

With the above result, we can work towards characterizing the probability that a selection matrix with pairs sampled uniformly at random has a particular rank. To make our results as general as possible, assume that we have a known "seed" $m \times n$ selection matrix $S_0$ with rank $r_0 \leq \min(m, n-2)$, and that we append $m$ randomly sampled rows to $S_0$ where each row is constructed by sampling two integers uniformly at random without replacement (and statistically independent from previous measurements and $S_0$) from $[n]$; denote these $m$ rows as $m \times n$ selection matrix $S$. We are interested in characterizing the probability that $\begin{bmatrix} S_0 \\ S \end{bmatrix}$ has rank $r > r_0$. We are only interested in $r_0 \leq \min(m, n-2)$, since if $r_0 = n - 1$ then from Lemma C.1 $S_0$ already has the maximum rank possible for a selection matrix with $n$ columns, and so we cannot increase its rank with additional random measurements.

Let $s_i$ denote the $i$th row of $S$. We will take an approach similar in spirit to the coupon collector problem by first defining a notion of a "failure" and "success" with regards to measuring new rows. After having queried $i - 1$ random paired comparisons given by rows $\{s_j\}_{j=1}^{i-1}$, we say that sampling a new selection row $s_i$ "fails" if it lies in the span of the selection matrix thus far, and "succeeds" if it lies outside this span. More precisely[8], define failure event $E_i = 0$ if $s_i \in \text{rowsp}(S_0) \cup \text{span}(\{s_j\}_{j=1}^{i-1})$ and success event $E_i = 1$ otherwise. Clearly, $\dim(\text{rowsp}(S_0) \cup \text{span}(\{s_j\}_{j=1}^{i})) = \dim(\text{rowsp}(S_0) \cup \text{span}(\{s_j\}_{j=1}^{i-1})) + 1$ if and only if $E_i = 1$. For $i \geq 1$, let $M_i = \min(\{k : \dim(\text{rowsp}(S_0) \cup \text{span}(\{s_j\}_{j=1}^{k})) = r_0 + i\}) = \min(\{k : \sum_{j=1}^k E_j = i\})$. Note that for any $i \geq 1$, $E_{M_i} = 1$; otherwise, $i = \sum_{j=1}^{M_i} E_j = \sum_{j=1}^{M_i-1} E_j + 0 < i$ by definition of $M_i$, which would be a contradiction. $M_{r-r_0}$ for $r > r_0$ is exactly the quantity we are interested in, since

---

[8]In the following statements concerning probability events, $S_0$ is assumed to be fixed and known.

it is the number of random measurements (beyond those already in $r_0$) needed for $r - r_0$ successes in total, i.e., for the cumulative selection matrix $\begin{bmatrix} S_0 \\ S \end{bmatrix}$ to be rank $r$.

To analyze $M_i$ for $1 \le i \le r - r_0$, let $C_i = M_i - M_{i-1}$ denote the number of measurements until the first success after already having had $i - 1$ successes, where $C_1 = M_1$. Then

$$M_i = (M_i - M_{i-1}) + (M_{i-1} - M_{i-2}) + (M_{i-2} + \cdots + (M_2 - M_1) + M_1 = \sum_{j=1}^{i} C_j.$$

Given $C_1, \ldots, C_{i-1}$ (and hence $M_{i-1}$), we note by definition that for any $c \ge 1$,

$$C_i > c \iff E_j = 0 \text{ for all } M_{i-1} + 1 \le j \le M_{i-1} + c. \tag{15}$$

Now suppose we condition on the event $C_1 = c_1, \ldots, C_{i-1} = c_{i-1}$, which we denote for shorthand by $c_1, \ldots, c_{i-1}$. We have

$$\mathbb{P}(C_i > c \mid c_1, \ldots, c_{i-1}) = \mathbb{P}\left( \bigcap_{k=M_{i-1}+1}^{M_{i-1}+c} E_k = 0 \;\middle|\; c_1, \ldots, c_{i-1} \right) \tag{16}$$

$$= \prod_{k=M_{i-1}+1}^{M_{i-1}+c} \mathbb{P}\left( E_k = 0 \;\middle|\; \bigcap_{\ell=M_{i-1}+1}^{k-1} (E_\ell = 0), c_1, \ldots, c_{i-1} \right), \tag{17}$$

where eq. (16) follows from eq. (15). For a fixed $k \in (M_{i-1} + 1) \ldots (M_{i-1} + c)$ let

$$S_k = \left\{ \{s_\ell\}_{\ell=1}^{k-1} : \bigcap_{\ell=M_{i-1}+1}^{k-1} (E_\ell = 0), C_1 = c_1, \ldots, C_{i-1} = c_{i-1} \right\},$$

i.e., $S_k$ is the set of all possible row sets $\{s_\ell\}_{\ell=1}^{k-1}$ that result in the events $\bigcap_{\ell=M_{i-1}+1}^{k-1} (E_\ell = 0), C_1 = c_1, \ldots, C_{i-1} = c_{i-1}$ (recall that by definition these events are deterministic when conditioned on $\{s_\ell\}_{\ell=1}^{k-1}$). A natural result of this set definition is

$$\{s_\ell\}_{\ell=1}^{k-1} \notin S_k \implies \mathbb{P}\left( \{s_\ell\}_{\ell=1}^{k-1} \;\middle|\; \bigcap_{\ell=M_{i-1}+1}^{k-1} (E_\ell = 0), c_1, \ldots, c_{i-1} \right) = 0, \tag{18}$$

We then have

$$\mathbb{P}\left( E_k = 0 \;\middle|\; \bigcap_{\ell=M_{i-1}+1}^{k-1} (E_\ell = 0), c_1, \ldots, c_{i-1} \right)$$

$$= \sum_{\{s_\ell\}_{\ell=1}^{k-1} \in S_k} \mathbb{P}(E_k = 0 \mid \{s_\ell\}_{\ell=1}^{k-1}, \bigcap_{\ell=M_{i-1}+1}^{k-1}(E_\ell = 0), c_1, \ldots, c_{i-1}) \mathbb{P}(\{s_\ell\}_{\ell=1}^{k-1} \mid \bigcap_{\ell=M_{i-1}+1}^{k-1}(E_\ell = 0), c_1, \ldots, c_{i-1}) \tag{19}$$

$$= \sum_{\{s_\ell\}_{\ell=1}^{k-1} \in S_k} \mathbb{P}(E_k = 0 \mid \{s_\ell\}_{\ell=1}^{k-1}) \mathbb{P}(\{s_\ell\}_{\ell=1}^{k-1} \mid \bigcap_{\ell=M_{i-1}+1}^{k-1} (E_\ell = 0), c_1, \ldots, c_{i-1}) \tag{20}$$

$$= \sum_{\{s_\ell\}_{\ell=1}^{k-1} \in S_k} \mathbb{P}(s_k \in \mathrm{rowsp}(S_0) \cup \mathrm{span}(\{s_\ell\}_{\ell=1}^{k-1}) \mid \{s_\ell\}_{\ell=1}^{k-1}) \mathbb{P}(\{s_\ell\}_{\ell=1}^{k-1} \mid \bigcap_{\ell=M_{i-1}+1}^{k-1} (E_\ell = 0), c_1, \ldots, c_{i-1})$$

$$\le \frac{(r_0 + i)(r_0 + i - 1)}{n(n-1)} \sum_{\{s_\ell\}_{\ell=1}^{k-1} \in S_k} \mathbb{P}(\{s_\ell\}_{\ell=1}^{k-1} \mid \bigcap_{\ell=M_{i-1}+1}^{k-1} (E_\ell = 0), c_1, \ldots, c_{i-1}) \tag{21}$$

$$= \frac{(r_0 + i)(r_0 + i - 1)}{n(n-1)} (1) \tag{22}$$

and so

$$\mathbb{P}\left( E_k = 0 \;\middle|\; \bigcap_{\ell=M_{i-1}+1}^{k-1} (E_\ell = 0), C_1 = c_1, \ldots, C_{i-1} = c_{i-1} \right) \le \frac{(r_0 + i)(r_0 + i - 1)}{n(n-1)}.$$

In the above, eq. (19) is a result of eq. (18), eq. (20) is since

$$\{s_\ell\}_{\ell=1}^{k-1} \in S_k \implies \bigcap_{\ell=M_{i-1}+1}^{k-1} (E_\ell = 0), c_1, \ldots, c_{i-1},$$

eq. (21) is from Lemma C.6 combined with the fact that since $\{s_\ell\}_{\ell=1}^{k-1} \in S_k$, $\dim(\mathrm{rowsp}(S_0) \cup \mathrm{span}(\{s_\ell\}_{\ell=1}^{k-1})) = r_0 + i - 1$, and eq. (22) is from eq. (18).

Continuing from eq. (17), this implies

$$\mathbb{P}(C_i > c \mid C_1 = c_1, \dots, C_{i-1} = c_{i-1}) \leq \left( \frac{(r_0 + i)(r_0 + i - 1)}{n(n-1)} \right)^c,$$

and so $\mathbb{P}(C_i \leq c \mid C_1 = c_1, \dots, C_{i-1} = c_{i-1}) \geq 1 - \left( \frac{(r_0+i)(r_0+i-1)}{n(n-1)} \right)^c$.

Now, consider a set of $r - r_0$ *independent* random variables, $B_1, \dots, B_{r-r_0}$, with each $B_i \in \{1, 2, 3, \dots\}$ distributed according to a geometric distribution with probability of success given by $p_i = 1 - \frac{(r_0+i)(r_0+i-1)}{n(n-1)}$. We will relate the statistics of $B_i$ to $C_i$ in order to construct a tail bound on $M_{r-r_0}$, our quantity of interest. Recalling the c.d.f. of geometric distributions, we have for $1 \leq i \leq r - r_0$,

$$\mathbb{P}(B_i \leq c \mid B_1, \dots, B_{i-1}) = \mathbb{P}(B_i \leq c) = 1 - (1 - p_i)^c = 1 - \left( \frac{(r_0 + i)(r_0 + i - 1)}{n(n-1)} \right)^c,$$

and so $\mathbb{P}(C_i \leq c \mid C_1, \dots, C_{i-1}) \geq \mathbb{P}(B_i \leq c)$ for all possible $C_1, \dots, C_{i-1}$.

Let $B := \sum_{i=1}^{r-r_0} B_i$. [46] presents a tail bound for the sum of independent geometric random variables, which we can apply to $B$. Let $X = \sum_{i=1}^{j} X_i$ be the sum of $j$ independent geometric random variables, each with parameter $0 < p_i \leq 1$. Define $\mu := \mathbb{E}[X] = \sum_{i=1}^{j} \frac{1}{p_i}$. Then from [46], for any $\lambda \geq 1$,

$$\mathbb{P}(X \geq \lambda \mu) \leq e^{1-\lambda}.$$

In our case, $X_i = B_i$, $j = r - r_0$, and

$$\mu = \mathbb{E}[B] = \sum_{i=1}^{r-r_0} \frac{1}{1 - \frac{(r_0+i)(r_0+i-1)}{n(n-1)}},$$

and so for any $\lambda \geq 1$,

$$\mathbb{P}(B \geq \lambda \mu) \leq e^{1-\lambda}. \tag{23}$$

To translate eq. (23) into a more interpretable tail bound, Let $0 < \delta < 1$ be given. If we choose $\lambda = 1 + \ln \frac{1}{\delta}$ (noting that $\lambda > 1$), then

$$\mathbb{P}\left( B \geq \left(1 + \ln \frac{1}{\delta}\right) \left( \sum_{i=1}^{r-r_0} \frac{1}{1 - \frac{(r_0+i)(r_0+i-1)}{n(n-1)}} \right) \right) \leq \delta. \tag{24}$$

If we can relate the statistics of $B$ to those of $M_{r-r_0}$, then we can potentially apply eq. (24) to construct a tail bound on $M_{r-r_0}$; the following lemma will provide the link we need. In the following, for a sequence $\{X_k\}_{k=i}^{j}$ let $X_{i:j} := \{X_k\}_{k=i}^{j}$.

**Lemma C.7.** *Let $\{X_i\}_{i=1}^{r}$ and $\{Y_i\}_{i=1}^{r}$ be two sets of random positive integers ($X_i, Y_i \in \mathbb{N} \; \forall i \in [r]$), where for $u_{1:i-1} \in \mathbb{N}^{i-1}$, $\{X_i\}_{i=1}^{r}$ is characterized by the distribution*

$$F_{X_i \mid X_{1:i-1}}(u \mid u_{1:i-1}) := \mathbb{P}(X_i \leq u \mid X_{1:i-1} = u_{1:i-1})$$

*and the $\{Y_i\}_{i=1}^{r}$ are statistically independent, so that for any $u_{1:i-1} \in \mathbb{N}^{i-1}$*

$$\mathbb{P}(Y_i \leq u \mid Y_{1:i-1} = u_{1:i-1}) = \mathbb{P}(Y_i \leq u) =: F_{Y_i}(u).$$

*Let $X := \sum_{i=1}^{r} X_i$ and $Y := \sum_{i=1}^{r} Y_i$, with $F_X(x) := \mathbb{P}(X \leq x)$ and $F_Y(y) := \mathbb{P}(Y \leq y)$. Suppose for all $i \in [r]$, $u \in \mathbb{R}$, and $u_{1:i-1} \in \mathbb{N}^{i-1}$, we have $F_{X_i \mid X_{1:i-1}}(u \mid u_{1:i-1}) \geq F_{Y_i}(u)$. Then $F_X(u) \geq F_Y(u)$ for all $u \in \mathbb{R}$ and $\mathbb{E}[X] \leq \mathbb{E}[Y]$.*

We defer the proof of Lemma C.7 to the end of the section.

**Corollary C.7.1.** *For all $c \in \mathbb{R}$, $\mathbb{P}(M_{r-r_0} > c) \leq \mathbb{P}(B > c)$ and $\mathbb{E}[M_{r-r_0}] \leq \mathbb{E}[B]$.*

*Proof.* $\{B_i\}_{i=1}^{r-r_0}$ are statistically independent, and we know $\mathbb{P}(C_i \leq c \mid C_1, \ldots, C_{i-1}) \geq \mathbb{P}(B_i \leq c)$ for all $c$ and all possible $C_1, \ldots, C_{i-1}$. Therefore by Lemma C.7, $\mathbb{P}(M_{r-r_0} \leq c) \geq \mathbb{P}(B \leq c)$ and so $\mathbb{P}(M_{r-r_0} > c) \leq \mathbb{P}(B > c)$ and $\mathbb{E}[M_{r-r_0}] \leq \mathbb{E}[B]$. $\qquad\square$

Combining Corollary C.7.1 with eq. (24), for any $0 < \delta < 1$ we have

$$\mathbb{P}\Big(M_{r-r_0} > \Big(1 + \ln\frac{1}{\delta}\Big)\Big(\sum_{i=1}^{r-r_0} \frac{1}{1 - \frac{(r_0+i)(r_0+i-1)}{n(n-1)}}\Big)\Big) \leq \mathbb{P}\Big(B > \Big(1 + \ln\frac{1}{\delta}\Big)\Big(\sum_{i=1}^{r-r_0} \frac{1}{1 - \frac{(r_0+i)(r_0+i-1)}{n(n-1)}}\Big)\Big)$$
$$\leq \delta$$

and

$$\mathbb{E}[M_{r-r_0}] \leq \sum_{i=1}^{r-r_0} \frac{1}{1 - \frac{(r_0+i)(r_0+i-1)}{n(n-1)}}.$$

In other words, with probability at least $1 - \delta$, $\Big(1 + \ln\frac{1}{\delta}\Big)\Big(\sum_{i=1}^{r-r_0} \frac{1}{1 - \frac{(r_0+i)(r_0+i-1)}{n(n-1)}}\Big)$ additional random measurements are sufficient to construct a rank $r$ selection matrix from $n$ items and a seed matrix $\boldsymbol{S}_0$ of rank $r_0$. We formalize the above facts in the following theorem:

**Theorem C.8.** *Let $\boldsymbol{S}_0$ be a given $m \times n$ selection matrix with rank $1 \leq r_0 \leq \min(m, n-2)$. Let $r_0 < r \leq n - 1$ be given. Consider the following random sampling procedure: at sampling time $i \geq 1$, let $\boldsymbol{s}_i = s_{(p,q)} \in \mathbb{R}^n$ where $p$ is sampled uniformly at random from $[n]$, $q$ is sampled uniformly at random from $[n] \setminus \{p\}$, and where each $\boldsymbol{s}_i$ is sampled independently from $\boldsymbol{s}_j$ for $j \neq i$ and from $\boldsymbol{S}_0$. Let $\boldsymbol{S}$ be the selection matrix constructed by concatenating the vectors $\boldsymbol{s}_i$ into rows. Suppose rows are appended to $\boldsymbol{S}$ until $\mathrm{rank}\big(\big[\begin{smallmatrix} \boldsymbol{S}_0 \\ \boldsymbol{S} \end{smallmatrix}\big]\big) = r$, at which point sampling halts. Let $M$ be the total number of rows in $\boldsymbol{S}$ resulting from this process. Then for any $0 < \delta < 1$,*

$$\mathbb{P}\Big(M > \Big(1 + \ln\frac{1}{\delta}\Big)\Big(\sum_{i=r_0+1}^{r} \frac{1}{1 - \frac{i(i-1)}{n(n-1)}}\Big)\Big) \leq \delta$$

*and*

$$\mathbb{E}[M] \leq \sum_{i=r_0+1}^{r} \frac{1}{1 - \frac{i(i-1)}{n(n-1)}}.$$

**Proof of Lemma C.6:** If $r = n - 1$, then by Lemma C.1 $\boldsymbol{S}$ already has maximal rank and so its row space spans $\mathbb{R}^n$. In this case, $\mathbb{P}(\boldsymbol{s} \in \mathrm{rowsp}(\boldsymbol{S}) \mid \boldsymbol{S}) = 1$, and so the upper bound of the inequality is tight at 1. The lower bound is satisfied since, as $n \geq 2$ by definition of selection matrices, $2r/(n(n-1)) = 2/n \leq 1 \leq \mathbb{P}(\boldsymbol{s} \in \mathrm{rowsp}(\boldsymbol{S}) \mid \boldsymbol{S})$ and so is true.

Otherwise, assume $r \leq n - 2$. Since $\boldsymbol{S}$ is rank $r$, there exists a set of $r$ linearly independent rows, which we denote by $\{\boldsymbol{s}_i\}_{i=1}^{r}$, such that $\mathrm{rowsp}(\boldsymbol{S}) = \mathrm{span}(\{\boldsymbol{s}_i\}_{i=1}^{r})$. Without loss of generality, for each $\boldsymbol{s}_i$ assume that $p_i < q_i$, where $\boldsymbol{s}_i[p_i] = 1$ and $\boldsymbol{s}_i[q_i] = -1$: this assumption does not affect the span of $\{\boldsymbol{s}_i\}_{i=1}^{r}$, since for $p, q \in [n]$ with $p \neq q$, $\boldsymbol{s}_{(p,q)} = -\boldsymbol{s}_{(q,p)}$. In a slight abuse of notation, let $\boldsymbol{S} \setminus \{\boldsymbol{s}_i\}_{i=1}^{r}$ denote the remaining rows in $\boldsymbol{S}$. We therefore have

$$\mathbb{P}(\boldsymbol{s} \in \mathrm{rowsp}(\boldsymbol{S}) \mid \boldsymbol{S}) = \mathbb{P}(\boldsymbol{s} \in \mathrm{span}(\{\boldsymbol{s}_i\}_{i=1}^{r}) \mid \boldsymbol{S}) = \mathbb{P}(\boldsymbol{s} \in \mathrm{span}(\{\boldsymbol{s}_i\}_{i=1}^{r}) \mid \{\boldsymbol{s}_i\}_{i=1}^{r}, \boldsymbol{S} \setminus \{\boldsymbol{s}_i\}_{i=1}^{r})$$
$$= \mathbb{P}(\boldsymbol{s} \in \mathrm{span}(\{\boldsymbol{s}_i\}_{i=1}^{r}) \mid \{\boldsymbol{s}_i\}_{i=1}^{r}),$$

where the last equality follows from the fact that $\boldsymbol{s}$ is statistically independent of $\boldsymbol{S}$.

Without loss of generality, suppose $p < q$. This does not affect our calculation of $\mathbb{P}(\boldsymbol{s} \in \mathrm{span}(\{\boldsymbol{s}_i\}_{i=1}^{r}) \mid \{\boldsymbol{s}_i\}_{i=1}^{r})$, since $\boldsymbol{s}_{(p,q)} \in \mathrm{span}(\{\boldsymbol{s}_i\}_{i=1}^{r}) \iff \boldsymbol{s}_{(q,p)} \in \mathrm{span}(\{\boldsymbol{s}_i\}_{i=1}^{r})$, as $\boldsymbol{s}_{(p,q)} = -\boldsymbol{s}_{(q,p)}$. Therefore, we can calculate $\boldsymbol{s}_{(p,q)} \in \mathrm{span}(\{\boldsymbol{s}_i\}_{i=1}^{r})$ directly by counting which among the $\binom{n}{2}$ equally likely pairs with $p < q$ lies in the span of $\{\boldsymbol{s}_i\}_{i=1}^{r}$. Precisely, let $Q := \{(p,q) : p, q \in [n], \ p < q, \ \boldsymbol{s}_{(p,q)} \in \mathrm{span}(\{\boldsymbol{s}_i\}_{i=1}^{r})\}$. Then

$$\mathbb{P}(\boldsymbol{s} \in \mathrm{span}(\{\boldsymbol{s}_i\}_{i=1}^{r}) \mid \{\boldsymbol{s}_i\}_{i=1}^{r}) = \frac{|Q|}{\binom{n}{2}}.$$

With these preliminaries established, we can easily lower bound $\mathbb{P}(\boldsymbol{s} \in \mathrm{span}(\{\boldsymbol{s}_i\}_{i=1}^{r}) \mid \{\boldsymbol{s}_i\}_{i=1}^{r})$: since $\boldsymbol{s}_i \in \mathrm{span}(\{\boldsymbol{s}_j\}_{j=1}^{r})$ for each $i \in [r]$, $Q$ contains the item pairs indexing the support of each

$\{s_j\}_{j=1}^r$. Furthermore, since $\{s_i\}_{i=1}^r$ are linearly independent, they must be distinct (i.e., for every $i, j \in [r]$, $s_i \neq s_j$) and so $Q$ contains at least $r$ distinct item pairs, i.e., $|Q| \geq r$. Hence,

$$\mathbb{P}(s \in \text{span}(\{s_i\}_{i=1}^r) \mid \{s_i\}_{i=1}^r) = \frac{|Q|}{\binom{n}{2}} \geq \frac{r}{\binom{n}{2}} = \frac{2r}{n(n-1)},$$

proving the lower bound in the inequality.

Letting $S := \{s_i\}_{i=1}^r$, we will upper bound $|Q|$ by analyzing the graph $G_S$ (with the graph construction introduced in Appendix C.1) with linked vertex pairs $C_S$. Let $I_S$ denote the set of distinct *item* pairs corresponding to linked vertices on $G_S$, i.e.,

$$I_S := \{(p, q) : p, q \in [n], \ p < q, \ \exists (i_p, i_q) \in C_S, \ s_{i_p}[p] \neq 0, \ s_{i_q}[q] \neq 0\}.$$

From Lemma C.5, $(p, q) \in Q \implies (p, q) \in I_S$ and so $|Q| \leq |I_S|$ and

$$\mathbb{P}(s \in \text{span}(\{s_i\}_{i=1}^r) \mid \{s_i\}_{i=1}^r) = \frac{|Q|}{\binom{n}{2}} \leq \frac{|I_S|}{\binom{n}{2}}. \tag{25}$$

We can therefore upper bound $\mathbb{P}(s \in \text{span}(\{s_i\}_{i=1}^r) \mid \{s_i\}_{i=1}^r)$ by upper bounding $|I_S|$.

To proceed, without loss of generality that suppose $G_S$ has exactly $c$ distinct subgraphs ($c \in [r]$) $G_1 = (V_1, E_1), \dots G_c = (V_c, E_c)$ where $V_1, \dots, V_c$ are a partition of $[r]$, such that for every $k$ and every $i, j \in V_k$, vertices $i$ and $j$ are linked on $G_S$ (and hence $G_k$), and for every $k, \ell \in [c]$ with $k \neq \ell$ and every $i \in V_k, j \in V_\ell$, vertices $i$ and $j$ are not linked on $G_S$. We next define item pairs according to which subgraph they pertain to: let

$$I_k := \{(p, q) : p, q \in [n], \ p < q, \ \exists (i_p, i_q) \in V_k \text{ s.t. } s_{i_p}[p] \neq 0, \ s_{i_q}[q] \neq 0\}. \tag{26}$$

Note that for any given item pair $(p, q) \in I_S$ with corresponding row indices $(i_p, i_q) \in C_S$ such that $s_{i_p}[p] \neq 0$ and $s_{i_q}[q] \neq 0$, there must exist $k \in [c]$ such that $i_p, i_q \in V_k$. Hence, $I_S = \bigcup_{k=1}^c I_k$ and therefore

$$|I_S| \leq \sum_{k=1}^c |I_k|. \tag{27}$$

To calculate $|I_k|$, first define $N_k$ to be the number of items supported by subgraph $G_k$:

$$N_k := \{i : i \in [n], \ \exists j \in V_k \text{ s.t. } s_j[i] \neq 0\}.$$

Since all vertices in $V_k$ are linked on $G_k$ (by construction), $|I_k|$ is exactly equal to all possible pair permutations of items in $N_k$, i.e., $|I_k| = \binom{|N_k|}{2}$. Let $r_k := |V_k|$ denote the number of vertices in subgraph $G_k$, noting that $\sum_{k=1}^c r_k = r$. For each $k \in [c]$ we then have $|N_k| = r_k + 1$: to see this, consider the selection matrix $S_{V_k}$ constructed from the rows $\{s_i\}_{i \in V_k}$, and note that the rows $\{s_i\}_{i \in V_k}$ are linearly independent by construction (since $\{s_i\}_{i \in V_k} \subseteq \{s_i\}_{i=1}^r$). Furthermore, suppose without loss of generality that $S_{V_k}$ is incremental: since $\{s_i\}_{i \in V_k}$ are linearly independent, by Corollary C.4.1 we can always find a permutation of these rows such that the resulting matrix $S_{V_k}$ is incremental. We will show below that each row of $S_{V_k}$ introduces *exactly* one new item.

Let $S_{V_k}^{(t)}$ denote the submatrix of $S_{V_k}$ consisting of the first $t$ rows: note that each $S_{V_k}^{(t)}$ is also incremental. Denote the $t$th row of $S_{V_k}$ by $s^{(t)}$. Suppose by contradiction that there exists $1 < i \leq r_k$ such that $s^{(i)}$ introduces exactly two new items that are not supported in $S_{V_k}^{(i-1)}$, and consider any $j < i$. Since every row index in $V_k$ is linked on $G_k$, there must exist at least one finite sequence of distinct indices $\{k_\ell\}_{\ell=1}^T \subseteq [r_k] \setminus \{i, j\}$ such that $s^{(i)}$ and $s^{(k_1)}$ share an item, each $s^{(k_\ell)}$ shares an item with $s^{(k_{\ell-1})}$ for $1 < \ell \leq T$, and $s^{(k_T)}$ shares an item with $s^{(j)}$. Note that $k_1 > i$, since $s^{(i)}$ cannot share an item directly with any row in $S_{V_k}^{(i-1)}$ (due to our contradictory assumption). Let $k^* = \max_{\ell \in [T]} k_\ell$; we know from the above argument that $k^* \geq k_1 > i > j$ and so $k^* > i, j, \{k_\ell\}_{\ell=1}^T \setminus \{k^*\}$. By definition of $\{k_\ell\}_{\ell=1}^T$, $s^{(k^*)}$ shares an item with two distinct indices $k_a, k_b \in (\{k_\ell\}_{\ell=1}^T \cup \{i, j\}) \setminus k^*$: let $p_a$ be the item index shared with $k_a$ and $p_b$ denote the item index shared with $k_b$. Since $k^* > i, j, \{k_\ell\}_{\ell=1}^T \setminus \{k^*\}$, $k_a < k^*$ and $k_b < k^*$, and therefore both $p_a$ and $p_b$ must appear in $S_{V_k}^{(k^*-1)}$. However, this is a contradiction since $S_{V_k}^{(k^*)}$ is incremental meaning that $s^{(k^*)}$ must introduce at least one new item. Therefore, there cannot exist index $1 < i \leq r_k$ such that

$s^{(i)}$ introduces exactly two new items that are not supported in $S_{V_k}^{(i-1)}$. Hence, hence the first row of $S_{V_k}$ introduces 2 new items and each subsequent row ($r_k - 1$ additional rows in total) introduces exactly one new item, resulting in $2 + r_k - 1 = r_k + 1$ supported columns in total, i.e., $|N_k| = r_k + 1$ and so $|I_k| = \binom{|N_k|}{2} = \binom{r_k+1}{2}$.

Therefore, by eq. (27),

$$|I_S| \leq \sum_{k=1}^{c} \binom{r_k + 1}{2} = \frac{1}{2} \sum_{k=1}^{c} (r_k + 1) r_k,$$

where we recall that $\sum_{k=1}^{c} r_k = r$ and $c \in [r]$. To get an upper bound on $|I_S|$ that only depends on $r$, we can maximize this bound over the choice of $\{r_k\}_{k=1}^{c}$ and $c$. We propose that $c = 1$ and hence $r_1 = r$ maximizes this bound: consider any other $c \in [r]$ and $\{r_k\}_{k=1}^{c}$ such that $\sum_{k=1}^{c} r_k = r$. We have

$$(r+1)r - \sum_{k=1}^{c}(r_k+1)r_k = r^2 + r - \sum_{k=1}^{c}(r_k^2 + r_k)$$

$$= \left( r^2 - \sum_{k=1}^{c} r_k^2 \right) + r - \sum_{k=1}^{c} r_k$$

$$= r^2 - \sum_{k=1}^{c} r_k^2$$

$$\geq r^2 - \sum_{k=1}^{c} r_k^2 - \sum_{k \neq \ell} r_k r_\ell$$

$$= r^2 - \left( \sum_{k=1}^{c} r_k \right)^2$$

$$= 0.$$

Hence, for any $c \in [r]$ and $\{r_k\}_{k=1}^{c}$ such that $\sum_{k=1}^{c} r_k = r$,

$$\frac{1}{2} \sum_{k=1}^{c} (r_k + 1) r_k \leq \frac{1}{2}(r+1)r,$$

and so $|I_S| \leq \frac{1}{2}(r+1)r$. Recalling eq. (25), we therefore have

$$\mathbb{P}(s \in \mathrm{span}(\{s_i\}_{i=1}^{r}) \mid \{s_i\}_{i=1}^{r}) \leq \frac{1}{2} \frac{(r+1)r}{\binom{n}{2}} = \frac{(r+1)r}{n(n-1)}.$$

**Proof of Lemma C.7:** Let $X^{(i)} := \sum_{j=i}^{r} X_j$ and $Y^{(i)} := \sum_{j=i}^{r} Y_j$: note that $X = X^{(1)}$ and $Y = Y^{(1)}$. Let

$$F_{X^{(i)}|X_{1:i-1}}(u \mid u_{1:i-1}) = \mathbb{P}(X^{(i)} \leq u \mid X_{1:i-1} = u_{1:i-1})$$

and

$$F_{Y^{(i)}}(u) := \mathbb{P}(Y^{(i)} \leq u).$$

We will prove by induction that $F_{X^{(1)}}(u) \geq F_{Y^{(1)}}(u)$ for all $u \in \mathbb{R}$, and therefore $F_X(u) \geq F_Y(u)$. Starting at $i = r$, we have by assumption that for all $u \in \mathbb{R}$, and $u_{1:r-1} \in \mathbb{N}^{r-1}$,

$$F_{X^{(r)}|X_{1:r-1}}(u \mid u_{1:r-1}) = F_{X_r|X_{1:r-1}}(u \mid u_{1:r-1}) \geq F_{Y_r}(u) = F_{Y^{(r)}}(u).$$

Now, let $1 \leq m < r$ be given, and suppose by induction that for any $u_{1:m} \in \mathbb{N}^m$, we have

$$F_{X^{(m+1)}|X_{1:m}}(u \mid u_{1:m}) \geq F_{Y^{(m+1)}}(u).$$

Expanding $F_{X^{(m)}|X_{1:m-1}}(u \mid u_{1:m-1})$,

$$F_{X^{(m)}|X_{1:m-1}}(u \mid u_{1:m-1}) =$$
$$= \mathbb{P}(X^{(m)} \leq u \mid X_{1:m-1} = u_{1:m-1})$$

$$= \mathbb{P}\Big(\sum_{j=m}^{r} X_j \le u \mid X_{1:m-1} = u_{1:m-1}\Big)$$

$$= \mathbb{P}\Big(\sum_{j=m+1}^{r} X_j \le u - X_m \mid X_{1:m-1} = u_{1:m-1}\Big)$$

$$= \sum_{v=1}^{\infty} \mathbb{P}\Big(\sum_{j=m+1}^{r} X_j \le u - X_m \mid X_{1:m-1} = u_{1:m-1}, X_m = v\Big) \mathbb{P}(X_m = v \mid X_{1:m-1} = u_{1:m-1})$$

$$= \sum_{v=1}^{\infty} F_{X^{(m+1)}|X_{1:m}}(u - v \mid u_{1:m-1}, v)(F_{X_m|X_{1:m-1}}(v \mid u_{1:i-1}) - F_{X_m|X_{1:m-1}}(v - 1 \mid u_{1:i-1})).$$

Similarly we have

$$F_{Y^{(m)}}(u) = \mathbb{P}(Y^{(m)} \le u)$$

$$= \mathbb{P}\Big(\sum_{j=m}^{r} Y_j \le u\Big)$$

$$= \mathbb{P}\Big(\sum_{j=m+1}^{r} Y_j \le u - Y_m\Big)$$

$$= \sum_{v=1}^{\infty} \mathbb{P}\Big(\sum_{j=m+1}^{r} Y_j \le u - Y_m \mid Y_m = v\Big) \mathbb{P}(Y_m = v)$$

$$= \sum_{v=1}^{\infty} \mathbb{P}\Big(\sum_{j=m+1}^{r} Y_j \le u - v\Big) \mathbb{P}(Y_m = v) \tag{28}$$

$$= \sum_{v=1}^{\infty} F_{Y^{(m+1)}}(u - v)(F_{Y_m}(v) - F_{Y_m}(v - 1)),$$

where eq. (28) follows from the fact that $\{Y_i\}_{i=1}^{r}$ are statistically independent. Therefore, letting $u_{1:m-1} \in \mathbb{N}^{m-1}$ be given,

$$F_{X^{(m)}|X_{1:m-1}}(u \mid u_{1:m-1}) - F_{Y^{(m)}}(u)$$

$$= \sum_{v=1}^{\infty} F_{X^{(m+1)}|X_{1:m}}(u - v \mid u_{1:m-1}, v)(F_{X_m|X_{1:m-1}}(v \mid u_{1:i-1})$$

$$- F_{X_m|X_{1:m-1}}(v - 1 \mid u_{1:i-1})) - F_{Y^{(m+1)}}(u - v)(F_{Y_m}(v) - F_{Y_m}(v - 1))$$

$$\ge \sum_{v=1}^{\infty} F_{Y^{(m+1)}}(u - v)(F_{X_m|X_{1:m-1}}(v \mid u_{1:i-1}) - F_{X_m|X_{1:m-1}}(v - 1 \mid u_{1:i-1})) -$$

$$F_{Y^{(m+1)}}(u - v)(F_{Y_m}(v) - F_{Y_m}(v - 1)) \tag{29}$$

$$= \sum_{v=1}^{\infty} F_{Y^{(m+1)}}(u - v)(F_{X_m|X_{1:m-1}}(v \mid u_{1:i-1}) - F_{Y_m}(v))$$

$$- \sum_{v=1}^{\infty} F_{Y^{(m+1)}}(u - v)(F_{X_m|X_{1:m-1}}(v - 1 \mid u_{1:i-1}) - F_{Y_m}(v - 1))$$

$$= \sum_{v=1}^{\infty} F_{Y^{(m+1)}}(u - v)(F_{X_m|X_{1:m-1}}(v \mid u_{1:i-1}) - F_{Y_m}(v))$$

$$- \sum_{z=0}^{\infty} F_{Y^{(m+1)}}(u - z - 1)(F_{X_m|X_{1:m-1}}(z \mid u_{1:i-1}) - F_{Y_m}(z)) \quad \text{where } z := v - 1$$

$$= \sum_{v=1}^{\infty} (F_{Y^{(m+1)}}(u - v) - F_{Y^{(m+1)}}(u - v - 1))(F_{X_m|X_{1:m-1}}(v \mid u_{1:i-1}) -$$

$$F_{Y_m}(v)) - F_{Y^{(m+1)}}(u - 1)(F_{X_m|X_{1:m-1}}(0 \mid u_{1:i-1}) - F_{Y_m}(0))$$

$$= \sum_{v=1}^{\infty} (F_{Y^{(m+1)}}(u - v) - F_{Y^{(m+1)}}(u - v - 1))(F_{X_m|X_{1:m-1}}(v \mid u_{1:i-1}) - F_{Y_m}(v)) \tag{30}$$

$$\geq 0 \tag{31}$$

where eq. (29) is by inductive assumption, eq. (30) is because $X_m, Y_m$ are non-negative and so

$$F_{X_m|X_{1:m-1}}(0 \mid u_{1:i-1}) = F_{Y_m|Y_{1:m-1}}(0 \mid v_{1:i-1})) = 0,$$

and eq. (31) is due to the fact that $F_{Y^{(m+1)}}(u)$ is non-decreasing and hence for all $v \geq 1$,

$$F_{Y^{(m+1)}}(u-v) - F_{Y^{(m+1)}}(u-v-1) \geq 0,$$

and by assumption we have

$$F_{X_m|X_{1:m-1}}(v \mid u_{1:i-1}) - F_{Y_m}(v) \geq 0.$$

Taking $m = 1$, we have $F_{X^{(1)}}(u) - F_{Y^{(1)}}(u) \geq 0$ i.e., $F_X(u) \geq F_Y(u)$.

Using the fact that $\mathbb{E}[X] = \sum_{u=0}^{\infty} \mathbb{P}(X > u) = \sum_{u=0}^{\infty}(1 - F_X(u))$ and similarly $\mathbb{E}[Y] = \sum_{u=0}^{\infty}(1 - F_Y(u))$, we have

$$\mathbb{E}[X] = \sum_{u=0}^{\infty}(1 - F_X(u))$$
$$\leq \sum_{u=0}^{\infty}(1 - F_Y(u))$$
$$= \mathbb{E}[Y].$$

### C.3  Proof of Proposition 2.1 and additional discussion

If $\mathbf{\Gamma}$ has full column rank, then its $D + dK$ columns are linearly independent and so $\mathrm{rank}(\mathbf{\Gamma}) = D+dK$. Since the rank of $\mathbf{\Gamma}$ is upper bounded by its number of rows, we require $\sum_{k=1}^{K} m_k \geq D+dK$. Next we will show in turn that each condition in Proposition 2.1 is necessary for $\mathbf{\Gamma}$ to have full column rank:

**(a)**  In order for all $D + dK$ columns in $\mathbf{\Gamma}$ to be linearly independent, it must be the case that for each $k \in [K]$, the columns corresponding to user $k$ are linearly independent, given by

$$\begin{bmatrix} \mathbf{0}_{m_1,d} \\ \vdots \\ \mathbf{S}_k \mathbf{X}^T \\ \vdots \\ \mathbf{0}_{m_K,d} \end{bmatrix}.$$

Clearly this is only possible if the $d$ columns in $\mathbf{S}_k \mathbf{X}^T$ are linearly independent (since padding by zeros does not affect linear independence of columns), i.e., $\mathrm{rank}(\mathbf{S}_k \mathbf{X}^T) = d$. Since $\mathrm{rank}(\mathbf{S}_k \mathbf{X}^T) \leq \mathrm{rank}(\mathbf{S}_k)$, we require $\mathrm{rank}(\mathbf{S}_k) \geq d$, which implies $m_k \geq d$ since $\mathbf{S}_k$ has $m_k$ rows.

**(b)**  Since $\mathrm{rank}(\mathbf{\Gamma}) = D + dK$, $\mathbf{\Gamma}$ must have $D + dK$ linearly independent rows. Observing eq. (3), each user's block of $m_k$ rows is given by

$$\begin{bmatrix} \mathbf{S}_k \mathbf{X}_{\otimes}^T & \mathbf{0}_{m_k,d} & \cdots & \mathbf{S}_k \mathbf{X}^T & \cdots & \mathbf{0}_{m_k,d} \end{bmatrix}, \tag{32}$$

which has the same column space, and therefore the same rank, as $\mathbf{S}_k \begin{bmatrix} \mathbf{X}_{\otimes}^T & \mathbf{X}^T \end{bmatrix}$. Therefore, the number of linearly independent rows in eq. (32) is equal to the rank of $\mathbf{S}_k \begin{bmatrix} \mathbf{X}_{\otimes}^T & \mathbf{X}^T \end{bmatrix}$, and so the number of linearly independent rows in $\mathbf{\Gamma}$ is upper bounded by $\sum_{k=1}^{K} \mathrm{rank}(\mathbf{S}_k \begin{bmatrix} \mathbf{X}_{\otimes}^T & \mathbf{X}^T \end{bmatrix})$, which for $\mathbf{\Gamma}$ with full column rank must be at least $D + dK$. Since $\mathrm{rank}(\mathbf{S}_k \begin{bmatrix} \mathbf{X}_{\otimes}^T & \mathbf{X}^T \end{bmatrix}) \leq \mathrm{rank}(\mathbf{S}_k)$, we have $\sum_{k=1}^{K} \mathrm{rank}(\mathbf{S}_k \begin{bmatrix} \mathbf{X}_{\otimes}^T & \mathbf{X}^T \end{bmatrix}) \leq \sum_{k=1}^{K} \mathrm{rank}(\mathbf{S}_k)$ and therefore we also require $\sum_{k=1}^{K} \mathrm{rank}(\mathbf{S}_k) \geq D + dK$.

**(c)** Consider any $\boldsymbol{\eta} \in \mathbb{R}^D$ and $\boldsymbol{v} \in \mathbb{R}^d$. Recalling eq. (3), multiplying $\boldsymbol{\Gamma}$ by $[\boldsymbol{\eta}^T \underbrace{\boldsymbol{v}^T \cdots \boldsymbol{v}^T}_{K \text{ times}}]^T$ is equivalent to

$$\boldsymbol{\Gamma} \begin{bmatrix} \boldsymbol{\eta} \\ \boldsymbol{v} \\ \vdots \\ \boldsymbol{v} \end{bmatrix} = \begin{bmatrix} \boldsymbol{S}_1 \boldsymbol{X}_\otimes^T \boldsymbol{\eta} + \boldsymbol{S}_1 \boldsymbol{X}^T \boldsymbol{v} \\ \vdots \\ \boldsymbol{S}_K \boldsymbol{X}_\otimes^T \boldsymbol{\eta} + \boldsymbol{S}_K \boldsymbol{X}^T \boldsymbol{v} \end{bmatrix} = \begin{bmatrix} \boldsymbol{S}_1 \\ \vdots \\ \boldsymbol{S}_K \end{bmatrix} \begin{bmatrix} \boldsymbol{X}_\otimes^T & \boldsymbol{X}^T \end{bmatrix} \begin{bmatrix} \boldsymbol{\eta} \\ \boldsymbol{v} \end{bmatrix} = \boldsymbol{S}_T \begin{bmatrix} \boldsymbol{X}_\otimes^T & \boldsymbol{X}^T \end{bmatrix} \begin{bmatrix} \boldsymbol{\eta} \\ \boldsymbol{v} \end{bmatrix}. \quad (33)$$

By the rank-nullity theorem, $\ker(\boldsymbol{S}_T \begin{bmatrix} \boldsymbol{X}_\otimes^T & \boldsymbol{X}^T \end{bmatrix})$ is trivial if and only if $\mathrm{rank}(\boldsymbol{S}_T \begin{bmatrix} \boldsymbol{X}_\otimes^T & \boldsymbol{X}^T \end{bmatrix}) = D + d$ (recall that $\begin{bmatrix} \boldsymbol{X}_\otimes^T & \boldsymbol{X}^T \end{bmatrix}$ has $D + d$ columns). Therefore if $\mathrm{rank}(\boldsymbol{S}_T \begin{bmatrix} \boldsymbol{X}_\otimes^T & \boldsymbol{X}^T \end{bmatrix}) < D + d$, there exists a $\begin{bmatrix} \boldsymbol{\eta} \\ \boldsymbol{v} \end{bmatrix} \neq \boldsymbol{0}$ such that $\boldsymbol{S}_T \begin{bmatrix} \boldsymbol{X}_\otimes^T & \boldsymbol{X}^T \end{bmatrix} \begin{bmatrix} \boldsymbol{\eta} \\ \boldsymbol{v} \end{bmatrix} = \boldsymbol{0}$ and therefore exists a nonzero vector in $\mathbb{R}^{D+dK}$ given by $[\boldsymbol{\eta}^T \underbrace{\boldsymbol{v}^T \cdots \boldsymbol{v}^T}_{K \text{ times}}]^T$ such that

$$\boldsymbol{\Gamma} \begin{bmatrix} \boldsymbol{\eta} \\ \boldsymbol{v} \\ \vdots \\ \boldsymbol{v} \end{bmatrix} = \boldsymbol{0},$$

which would imply that $\boldsymbol{\Gamma}$ is rank deficient. Therefore, we require $\mathrm{rank}(\boldsymbol{S}_T \begin{bmatrix} \boldsymbol{X}_\otimes^T & \boldsymbol{X}^T \end{bmatrix}) = D + d$, and since $\mathrm{rank}(\boldsymbol{S}_T \begin{bmatrix} \boldsymbol{X}_\otimes^T & \boldsymbol{X}^T \end{bmatrix}) \leq \min(\mathrm{rank}(\boldsymbol{S}_T), \mathrm{rank}(\begin{bmatrix} \boldsymbol{X}_\otimes^T & \boldsymbol{X}^T \end{bmatrix}))$ this implies $\mathrm{rank}(\boldsymbol{S}_T) \geq D + d$ and $\mathrm{rank}(\begin{bmatrix} \boldsymbol{X}_\otimes^T & \boldsymbol{X}^T \end{bmatrix}) \geq D + d$. Since $\boldsymbol{S}_T$ is itself a selection matrix, by Lemma C.1 we require $n \geq D + d + 1$ in order for $\mathrm{rank}(\boldsymbol{S}_T) \geq D + d$. $\qquad \square$

In the setting where the metric $\boldsymbol{M}$ is low-rank with rank $r < d$, we conjecture that the required item scaling grows as $O(rd)$ rather than $O(d^2)$, which is a much gentler increase especially when $r \ll d$. Our intuition for this conjecture is as follows: the requirement for $O(d^2)$ items in the full-rank metric case comes from part (c) above, which says that if a hypothetical *single* user existed that answered the queries assigned to *all* users, then such a system would require $O(d^2)$ measurements due to the $O(d^2)$ degrees of freedom in the metric. Due to properties of selection matrices, we require the same or greater order of items as measurements since intuitively one new item is required per independent measurement (see Lemma C.1). If $\boldsymbol{M}$ is rank $r < d$, there are only $dr$ degrees of freedom in $\boldsymbol{M}$, and hence we believe that part (c) above would only require $O(dr)$ independent measurements for the hypothetical single user and therefore only $O(dr)$ items. Concretely, by rewriting the unquantized measurements in eq. (2) as a matrix inner product between $\begin{bmatrix} \boldsymbol{M} & \boldsymbol{v}_1 & \dots & \boldsymbol{v}_K \end{bmatrix}$ and a corresponding measurement matrix only depending on the items (see eq. (40) in the proof of Theorem 3.1 for an example of this technique), and using low-rank matrix recovery techniques such as those described in [47], we believe that one can show only $O(dr)$ independent measurements are required for the hypothetical single user and therefore only $O(dr)$ items are required.

## C.4 Proof of Proposition 2.2

We first permute the rows of $\boldsymbol{\Gamma}$ as follows (row permutations do not change matrix rank): first, define $\boldsymbol{\Gamma}^{(1)}$ as

$$\boldsymbol{\Gamma}^{(1)} := \begin{bmatrix} \boldsymbol{S}_1^{(1)} \boldsymbol{X}_\otimes^T & \boldsymbol{S}_1^{(1)} \boldsymbol{X}^T & \boldsymbol{0}_{d,d} & \cdots & \boldsymbol{0}_{d,d} \\ \boldsymbol{S}_2^{(1)} \boldsymbol{X}_\otimes^T & \boldsymbol{0}_{d,d} & \boldsymbol{S}_2^{(1)} \boldsymbol{X}^T & \cdots & \boldsymbol{0}_{d,d} \\ \vdots & \vdots & \vdots & \vdots & \vdots \\ \boldsymbol{S}_K^{(1)} \boldsymbol{X}_\otimes^T & \boldsymbol{0}_{d,d} & \boldsymbol{0}_{d,d} & \cdots & \boldsymbol{S}_K^{(1)} \boldsymbol{X}^T \end{bmatrix} \quad (34)$$

and $\boldsymbol{\Gamma}^{(2)}$ as

$$\boldsymbol{\Gamma}^{(2)} := \boldsymbol{P} \begin{bmatrix} \boldsymbol{S}_1^{(2)} \boldsymbol{X}_\otimes^T & \boldsymbol{S}_1^{(2)} \boldsymbol{X}^T & \boldsymbol{0}_{m_1-d,d} & \cdots & \boldsymbol{0}_{m_1-d,d} \\ \boldsymbol{S}_2^{(2)} \boldsymbol{X}_\otimes^T & \boldsymbol{0}_{m_2-d,d} & \boldsymbol{S}_2^{(2)} \boldsymbol{X}^T & \cdots & \boldsymbol{0}_{m_2-d,d} \\ \vdots & \vdots & \vdots & \vdots & \vdots \\ \boldsymbol{S}_K^{(2)} \boldsymbol{X}_\otimes^T & \boldsymbol{0}_{m_K-d,d} & \boldsymbol{0}_{m_K-d,d} & \cdots & \boldsymbol{S}_K^{(2)} \boldsymbol{X}^T \end{bmatrix}. \quad (35)$$

Finally, define $\widehat{\boldsymbol{\Gamma}} = \begin{bmatrix} \boldsymbol{\Gamma}^{(1)} \\ \boldsymbol{\Gamma}^{(2)} \end{bmatrix}$. Since $\widehat{\boldsymbol{\Gamma}}$ is simply a permutation of the rows in $\boldsymbol{\Gamma}$, $\mathrm{rank}(\widehat{\boldsymbol{\Gamma}}) = \mathrm{rank}(\boldsymbol{\Gamma})$. Therefore, if we show that the $D + dK$ rows in $\widehat{\boldsymbol{\Gamma}}$ are linearly independent and hence $\mathrm{rank}(\widehat{\boldsymbol{\Gamma}}) = D + dK$, we will have shown that $\mathrm{rank}(\boldsymbol{\Gamma}) = D + dK$ and so $\boldsymbol{\Gamma}$ is full column rank.

We will start by examining the rows in $\boldsymbol{\Gamma}^{(1)}$. For $k \in [K]$, let $\boldsymbol{Q}_k^{(1)} := \boldsymbol{S}_k^{(1)} \boldsymbol{X}^T$; evaluating this matrix product, the $i$th row of $\boldsymbol{Q}_k^{(1)}$ is given by $\boldsymbol{x}_{p_{k,i}} - \boldsymbol{x}_{q_{k,i}}$, where $p_{k,i}$ indexes the $+1$ entry in the $i$th row of $\boldsymbol{S}_k^{(1)}$ and $q_{k,i}$ indexes the $-1$ entry in the $i$th row of $\boldsymbol{S}_k^{(1)}$. Since each $\boldsymbol{x}_i$ is i.i.d. distributed according to $p_X$, and $p_X$ is absolutely continuous with respect to the Lebesgue measure, for any $i \neq j$ we have that $\boldsymbol{z}_{i,j} := \boldsymbol{x}_i - \boldsymbol{x}_j$ is distributed according to some distribution $p_Z$ (which does not depend on $i$ or $j$ since $\boldsymbol{x}_i$ is i.i.d. for all $i$) that is also absolutely continuous with respect to the Lebesgue measure.

Inspecting the first row of $\boldsymbol{Q}_k^{(1)}$, we then have

$$\mathbb{P}(\boldsymbol{x}_{p_{k,1}} - \boldsymbol{x}_{q_{k,1}} = \boldsymbol{0}) = \mathbb{P}_Z(\boldsymbol{z} = \boldsymbol{0}) = 0,$$

where the last equality follows since $\mu(\{\boldsymbol{0}\}) = 0$, where $\mu$ is the Lebesgue measure, and $p_Z$ is absolutely continuous. Hence, with probability 1, $\boldsymbol{x}_{p_{k,1}} - \boldsymbol{x}_{q_{k,1}}$ is nonzero and spans a 1-dimensional subspace of $\mathbb{R}^d$.

Let $\boldsymbol{w}$ be a vector orthogonal to $\boldsymbol{x}_{p_{k,1}} - \boldsymbol{x}_{q_{k,1}}$, and consider $\boldsymbol{x}_{p_{k,2}} - \boldsymbol{x}_{q_{k,2}}$, which is the second row of $\boldsymbol{Q}_k^{(1)}$. Since by assumption $\boldsymbol{S}_k^{(1)}$ is incremental, at least one of $p_{k,2}$ or $q_{k,2}$ is not equal to $p_{k,1}$ or $q_{k,1}$. Suppose that both $p_{k,2}, q_{k,2} \notin \{p_{k,1}, q_{k,1}\}$. Then

$$\mathbb{P}(\boldsymbol{w}^T(\boldsymbol{x}_{p_{k,2}} - \boldsymbol{x}_{q_{k,2}}) = 0 \mid \boldsymbol{x}_{p_{k,1}}, \boldsymbol{x}_{q_{k,1}}) = \mathbb{P}_Z(\boldsymbol{w}^T \boldsymbol{z} = 0 \mid \boldsymbol{x}_{p_{k,1}}, \boldsymbol{x}_{q_{k,1}}) = 0,$$

where the last equality follows from the fact that $\mu(\{\boldsymbol{w}^T \boldsymbol{z} = 0 : \boldsymbol{z} \in \mathbb{R}^d\}) = 0$ and $p_Z$ is absolutely continuous.

Now suppose that exactly one of $p_{k,2}$ or $q_{k,2}$ is not equal to $p_{k,1}$ or $q_{k,1}$. Without loss of generality, suppose $q_{k,2}$ is equal to $p_{k,1}$ or $q_{k,1}$ (the same argument holds if this were true for $p_{k,2}$ instead). Then

$$\mathbb{P}(\boldsymbol{w}^T(\boldsymbol{x}_{p_{k,2}} - \boldsymbol{x}_{q_{k,2}}) = 0 \mid \boldsymbol{x}_{p_{k,1}}, \boldsymbol{x}_{q_{k,1}}) = \mathbb{P}(\boldsymbol{w}^T \boldsymbol{x}_{p_{k,2}} - \boldsymbol{w}^T \boldsymbol{x}_{q_{k,2}} = 0 \mid \boldsymbol{x}_{p_{k,1}}, \boldsymbol{x}_{q_{k,1}}, \boldsymbol{x}_{q_{k,2}})$$

$$= \mathbb{P}_X(\boldsymbol{w}^T \boldsymbol{x} - c = 0 \mid \boldsymbol{x}_{p_{k,1}}, \boldsymbol{x}_{q_{k,1}}, \boldsymbol{x}_{q_{k,2}}) \quad \text{where } c \text{ is a constant.}$$

$$= 0,$$

where the first equality follows from the fact that $q_{k,2} \in \{p_{k,1}, q_{k,1}\}$, the second equality follows from the fact that when conditioned on $\boldsymbol{x}_{q_{k,2}}$, $\boldsymbol{w}^T \boldsymbol{x}_{q_{k,2}}$ is a constant (which we denote by $c$) and that $\boldsymbol{x}_{p_{k,2}}$ is distributed as $p_X$ and is independent of other $\boldsymbol{x}_j$ for $j \neq p_{k,2}$, and the final equality follows from the fact that $\mu(\{\boldsymbol{w}^T \boldsymbol{x} - c = 0 : \boldsymbol{x} \in \mathbb{R}^d\}) = 0$ and $p_X$ is absolutely continuous.

In either scenario, $\mathbb{P}(\boldsymbol{w}^T(\boldsymbol{x}_{p_{k,2}} - \boldsymbol{x}_{q_{k,2}}) = 0 \mid \boldsymbol{x}_{p_{k,1}}, \boldsymbol{x}_{q_{k,1}}) = 0$. Hence, when conditioned on $\boldsymbol{x}_{p_{k,1}}$ and $\boldsymbol{x}_{q_{k,1}}$, with probability 1 $\boldsymbol{x}_{p_{k,2}} - \boldsymbol{x}_{q_{k,2}}$ includes a component orthogonal to $\boldsymbol{x}_{p_{k,1}} - \boldsymbol{x}_{q_{k,1}}$ and therefore does not lie in the span of $\boldsymbol{x}_{p_{k,1}} - \boldsymbol{x}_{q_{k,1}}$. Denote the first $j$ rows of $\boldsymbol{Q}_k^{(1)}$ as

$$\boldsymbol{Q}_k^{(1)}[1:j] = \begin{bmatrix} (\boldsymbol{x}_{p_{k,1}} - \boldsymbol{x}_{q_{k,1}})^T \\ \vdots \\ (\boldsymbol{x}_{p_{k,j}} - \boldsymbol{x}_{q_{k,j}})^T \end{bmatrix}.$$

Then, from the above argument, $\mathbb{P}(\mathrm{rank}(\boldsymbol{Q}_k^{(1)}[1:2]) = 2 \mid \boldsymbol{x}_{p_{k,1}}, \boldsymbol{x}_{q_{k,1}}) = 1$. This is true for any $\boldsymbol{x}_{p_{k,1}}, \boldsymbol{x}_{q_{k,1}}$ satisfying $\boldsymbol{x}_{p_{k,1}} - \boldsymbol{x}_{q_{k,1}} \neq \boldsymbol{0}$, which we know occurs with probability 1, and so marginalizing over this event we have $\mathbb{P}(\mathrm{rank}(\boldsymbol{Q}_k^{(1)}[1:2]) = 2) = 1$.

If $d > 2$, let $2 \leq m < d$ be given, and suppose by induction that $\mathbb{P}(\mathrm{rank}(\boldsymbol{Q}_k^{(1)}[1:m]) = m) = 1$. The rows of $\boldsymbol{Q}_k^{(1)}[1:m]$ are constructed from vectors $\boldsymbol{x}_i$ where $i \in M$ and $M := \{p_{k,j}\}_{j=1}^m \cup \{q_{k,j}\}_{j=1}^m$.

Let $\boldsymbol{w}$ be a vector in the orthogonal subspace to $\mathrm{rowsp}(\boldsymbol{Q}_k^{(1)}[1:m])$. Consider row $m+1$ of $\boldsymbol{Q}_k^{(1)}$, given by $\boldsymbol{x}_{p_{k,m+1}} - \boldsymbol{x}_{q_{k,m+1}}$. Since $\boldsymbol{S}_k^{(1)}$ is incremental, at least one of $p_{k,m+1}$ or $q_{k,m+1}$ is not in

$M$. First suppose this is true for both $p_{k,m+1}$ and $q_{k,m+1}$. Then by similar arguments as above,

$$\mathbb{P}(\boldsymbol{w}^T(\boldsymbol{x}_{p_{k,m+1}} - \boldsymbol{x}_{q_{k,m+1}}) = 0 \mid \{\boldsymbol{x}_i\}_{i \in M}) = \mathbb{P}_Z(\boldsymbol{w}^T \boldsymbol{z} = 0 \mid \{\boldsymbol{x}_i\}_{i \in M}) = 0.$$

Now, instead suppose without loss of generality that only $q_{k,m+1} \in M$ (an identical argument holds for $p_{k,m+1} \in M$). Then by similar arguments as above,

$$\mathbb{P}(\boldsymbol{w}^T(\boldsymbol{x}_{p_{k,m+1}} - \boldsymbol{x}_{q_{k,m+1}}) = 0 \mid \{\boldsymbol{x}_i\}_{i \in M}) = \mathbb{P}(\boldsymbol{w}^T \boldsymbol{x}_{p_{k,m+1}} - \boldsymbol{w}^T \boldsymbol{x}_{q_{k,m+1}} = 0 \mid \boldsymbol{x}_{q_{k,m+1}}, \{\boldsymbol{x}_i\}_{i \in M})$$
$$= \mathbb{P}_X(\boldsymbol{w}^T \boldsymbol{x} - c = 0 \mid \boldsymbol{x}_{q_{k,m+1}}, \{\boldsymbol{x}_i\}_{i \in M}) \quad \text{where } c \text{ is a constant.}$$
$$= 0.$$

In either scenario, $\mathbb{P}(\boldsymbol{w}^T(\boldsymbol{x}_{p_{k,m+1}} - \boldsymbol{x}_{q_{k,m+1}}) = 0 \mid \{\boldsymbol{x}_i\}_{i \in M}) = 0$. Hence, when conditioned on $\{\boldsymbol{x}_i\}_{i \in M}$, $\boldsymbol{x}_{p_{k,m+1}} - \boldsymbol{x}_{q_{k,m+1}}$ includes a component orthogonal to the row space of $\boldsymbol{Q}_k^{(1)}[1:m]$ and therefore does not lie in this row space. In other words,

$$\mathbb{P}(\text{rank}(\boldsymbol{Q}_k^{(1)}[1:m+1]) = m+1 \mid \{\boldsymbol{x}_i\}_{i \in M}) = 1.$$

This is true for any $\{\boldsymbol{x}_i\}_{i \in M}$ satisfying $\text{rank}(\boldsymbol{Q}_k^{(1)}[1:m]) = m$, which by inductive assumption is true with probability 1 and so marginalizing over this event we have $\mathbb{P}(\text{rank}(\boldsymbol{Q}_k^{(1)}[1:m+1]) = m+1) = 1$. Taking $m = d-1$, and noting that $\boldsymbol{Q}_k^{(1)}[1:d] = \boldsymbol{Q}_k^{(1)}$, $\mathbb{P}(\text{rank}(\boldsymbol{Q}_k^{(1)}) = d) = 1$. Since this is true for all $k \in [K]$, by the union bound we have

$$\mathbb{P}\left( \bigcup_{k \in [K]} (\text{rank}(\boldsymbol{Q}_k^{(1)}) < d) \right) \leq \sum_{k \in [K]} \mathbb{P}(\text{rank}(\boldsymbol{Q}_k^{(1)}) < d) \leq \sum_{k \in [K]} 0 = 0,$$

and therefore with probability 1, $\text{rank}(\boldsymbol{Q}_k^{(1)}) = d$ simultaneously for all $k \in [K]$.

Consider the following matrix:

$$\boldsymbol{Q}^{(1)} := \begin{bmatrix} \boldsymbol{Q}_1^{(1)} & \boldsymbol{0}_{d,d} & \cdots & \boldsymbol{0}_{d,d} \\ \boldsymbol{0}_{d,d} & \boldsymbol{Q}_2^{(1)} & \cdots & \boldsymbol{0}_{d,d} \\ \vdots & \vdots & \vdots & \vdots \\ \boldsymbol{0}_{d,d} & \boldsymbol{0}_{d,d} & \cdots & \boldsymbol{Q}_K^{(1)} \end{bmatrix}.$$

Each consecutive block of $d$ rows in $\boldsymbol{Q}^{(1)}$ is clearly orthogonal, and since with probability 1 each $\boldsymbol{Q}_k^{(1)}$ is simultaneously full row rank, with probability 1 we have that $\boldsymbol{Q}^{(1)}$ is full row rank (and hence is invertible since it is square). Inspecting eq. (34), we can write $\boldsymbol{\Gamma}^{(1)} = \begin{bmatrix} \boldsymbol{R}^{(1)} & \boldsymbol{Q}^{(1)} \end{bmatrix}$ where $\boldsymbol{R}^{(1)}$ is a $Kd \times D$ submatrix. Since $\boldsymbol{Q}^{(1)}$ is rank $Kd$, the column space of $\boldsymbol{\Gamma}^{(1)}$ is also of dimension at least $Kd$ and therefore $\boldsymbol{\Gamma}^{(1)}$ is full row rank since it has $Kd$ rows. In other words, we have shown that with probability 1 the rows of $\boldsymbol{\Gamma}^{(1)}$ are linearly independent. We will now show linear independence for the remaining rows in $\widehat{\boldsymbol{\Gamma}}$ (i.e., $\boldsymbol{\Gamma}^{(2)}$), which completes our proof. Specifically, we will proceed through the remaining $D$ measurements row by row, and inductively show how each cumulative set of rows is linearly independent.

First, we define some additional notation: for any vector $\boldsymbol{w} \in \mathbb{R}^{D+Kd}$, let $\psi_k(\boldsymbol{w})$ be the subvector limited to the column indices of $\boldsymbol{\Gamma}$ involving user $k$, i.e.,

$$\psi_k(\boldsymbol{w}) := \begin{bmatrix} \boldsymbol{w}[1:D] \\ \boldsymbol{w}[D+(k-1)d+1:D+kd] \end{bmatrix}.$$

Let $\boldsymbol{r}_i$ denote the $i$th row of $\boldsymbol{\Gamma}^{(2)}$, let $k_i \in [K]$ denote the user that this row corresponds to (i.e., $\boldsymbol{r}_i$ is supported on columns $1:D$ and $D+(k_i-1)d+1:D+k_id$), and let $j_{i,1}, j_{i,2}$ denote the first and second items selected at this measurement. We require this flexible definition of the user and items in row $\boldsymbol{r}_i$, since the permutation $\boldsymbol{P}$ has arbitrarily scrambled the users that each row in $\boldsymbol{\Gamma}^{(2)}$ corresponds to. Finally, let

$$\phi(\boldsymbol{x}) = \begin{bmatrix} \boldsymbol{x} \otimes_S \boldsymbol{x} \\ \boldsymbol{x} \end{bmatrix},$$

which is a vector in $\mathbb{R}^{D+d}$. For any nonzero vector $\boldsymbol{\mu} \in \mathbb{R}^{D+d}$, $\boldsymbol{\mu}^T \phi(\boldsymbol{x})$ is a nontrivial polynomial in $\boldsymbol{x}$. With this notation defined, for any vector $\boldsymbol{w} \in \mathbb{R}^{D+Kd}$ we see from eq. (35) that

$$\boldsymbol{w}^T \boldsymbol{r}_i = \psi_{k_i}(\boldsymbol{w})^T (\phi(\boldsymbol{x}_{j_{i,1}}) - \phi(\boldsymbol{x}_{j_{i,2}})).$$

Next, we establish a fact about the orthogonal subspace to $\mathrm{rowsp}(\boldsymbol{\Gamma}^{(1)})$. Let $E_0 := \mathrm{span}(\{\boldsymbol{e}_i\}_{i=1}^D)$ where $\boldsymbol{e}_i$ is the $i$th standard basis vector in $\mathbb{R}^{D+dK}$. It is a fact that for every $\boldsymbol{w} \in \mathrm{rowsp}(\boldsymbol{\Gamma}^{(1)})^\perp$, $\mathrm{proj}_{E_0} \boldsymbol{w} \neq \boldsymbol{0}$. In other words, $\boldsymbol{w}$ has at least one nonzero element in its first $D$ entries. Suppose this were not true: then for some nonzero $\boldsymbol{w}' \in \mathbb{R}^{Kd}$, we would have

$$\boldsymbol{w} = \begin{bmatrix} \boldsymbol{0} \\ \boldsymbol{w}' \end{bmatrix}.$$

Since $\boldsymbol{w}' \in \mathbb{R}^{Kd}$ and the rows of $\boldsymbol{Q}^{(1)}$ are a basis for $\mathbb{R}^{Kd}$ (since $\boldsymbol{Q}^{(1)}$ is invertible), we have $\boldsymbol{w}' = (\boldsymbol{Q}^{(1)})^T \boldsymbol{\beta}$ for some $\boldsymbol{\beta} \in \mathbb{R}^{Kd}$. Consider $\boldsymbol{r} = (\boldsymbol{\Gamma}^{(1)})^T \boldsymbol{\beta}$, which is clearly in $\mathrm{rowsp}(\boldsymbol{\Gamma}^{(1)})$. Expanding $\boldsymbol{\Gamma}^{(1)}$, we have

$$\boldsymbol{r} = \begin{bmatrix} (\boldsymbol{R}^{(1)})^T \boldsymbol{\beta} \\ (\boldsymbol{Q}^{(1)})^T \boldsymbol{\beta} \end{bmatrix} = \begin{bmatrix} (\boldsymbol{R}^{(1)})^T \boldsymbol{\beta} \\ \boldsymbol{w}' \end{bmatrix},$$

and so

$$\boldsymbol{w}^T \boldsymbol{r} = \boldsymbol{0}^T (\boldsymbol{R}^{(1)})^T \boldsymbol{\beta} + (\boldsymbol{w}')^T \boldsymbol{w}' = \|\boldsymbol{w}'\|_2^2 > 0,$$

where the last inequality follows since $\boldsymbol{w}' \neq \boldsymbol{0}$. This is a contradiction since by definition, $\boldsymbol{w}^T \boldsymbol{r} = 0$ for every $\boldsymbol{r} \in \mathrm{rowsp}(\boldsymbol{\Gamma}^{(1)})$.

Let $\boldsymbol{w}$ be a vector in $\mathrm{rowsp}(\boldsymbol{\Gamma}^{(1)})^\perp$ not equal to the zero vector, and let $J$ denote the item indices on which each $\boldsymbol{S}_k^{(1)}$ is supported across all $k \in [K]$, i.e.,

$$J = \{j : j \in [n], \exists k \in [K], i \in [d] \text{ s.t. } \boldsymbol{S}_k^{(1)}[i,j] \neq 0\}.$$

Consider the first row of $\boldsymbol{\Gamma}^{(2)}$. By the incremental assumption, at least one of $j_{1,1}$ or $j_{1,2}$ is not found in $J$. First suppose that both are not found in $J$. Then

$$\mathbb{P}(\boldsymbol{w}^T \boldsymbol{r}_1 = 0 \mid \{\boldsymbol{x}_i\}_{i \in J}) = \mathbb{P}(\psi_{k_1}(\boldsymbol{w})^T (\phi(\boldsymbol{x}_{j_{1,1}}) - \phi(\boldsymbol{x}_{j_{1,2}})) = 0 \mid \{\boldsymbol{x}_i\}_{i \in J}). \quad (36)$$

As an aside, if $\boldsymbol{x}_i, \boldsymbol{x}_j$ are i.i.d. distributed according to absolutely continuous distribution $p_X$, then the joint distribution $p_{\boldsymbol{x}_i, \boldsymbol{x}_j}(\boldsymbol{x}_i, \boldsymbol{x}_j) = p_X(\boldsymbol{x}_i) p_X(\boldsymbol{x}_j)$ is also absolutely continuous. Also note that for any $\boldsymbol{\mu} \in \mathbb{R}^{D+d}$,

$$\boldsymbol{\mu}^T (\phi(\boldsymbol{x}_i) - \phi(\boldsymbol{x}_j)) = \begin{bmatrix} \boldsymbol{\mu}^T & -\boldsymbol{\mu}^T \end{bmatrix} \begin{bmatrix} \phi(\boldsymbol{x}_i) \\ \phi(\boldsymbol{x}_j) \end{bmatrix}. \quad (37)$$

Since $\begin{bmatrix} \boldsymbol{x}_i \\ \boldsymbol{x}_j \end{bmatrix} \otimes_S \begin{bmatrix} \boldsymbol{x}_i \\ \boldsymbol{x}_j \end{bmatrix}$ contains all terms in $\boldsymbol{x}_i \otimes_S \boldsymbol{x}_i$ and $\boldsymbol{x}_j \otimes_S \boldsymbol{x}_j$, $\phi(\begin{bmatrix} \boldsymbol{x}_i \\ \boldsymbol{x}_j \end{bmatrix})$ contains $\phi(\boldsymbol{x}_i)$ and $\phi(\boldsymbol{x}_j)$. Therefore, we can view eq. (37) as being a polynomial in $\begin{bmatrix} \boldsymbol{x}_i \\ \boldsymbol{x}_j \end{bmatrix}$. If $\boldsymbol{\mu} \neq \boldsymbol{0}$, then $\begin{bmatrix} \boldsymbol{\mu} \\ -\boldsymbol{\mu} \end{bmatrix} \neq \boldsymbol{0}$ and so $\boldsymbol{\mu}^T (\phi(\boldsymbol{x}_i) - \phi(\boldsymbol{x}_j))$ can be viewed as a nontrivial polynomial in $\begin{bmatrix} \boldsymbol{x}_i \\ \boldsymbol{x}_j \end{bmatrix}$. Since $p_{\boldsymbol{x}_i, \boldsymbol{x}_j}$ is absolutely continuous, for any nonzero $\boldsymbol{\mu}$ we have

$$\mathbb{P}(\boldsymbol{\mu}^T (\phi(\boldsymbol{x}_i) - \phi(\boldsymbol{x}_j)) = 0) = 0,$$

since the set of roots for a nontrivial polynomial is a set of Lebesgue measure 0.

Returning to eq. (36), we then have

$$\mathbb{P}(\boldsymbol{w}^T \boldsymbol{r}_1 = 0 \mid \{\boldsymbol{x}_i\}_{i \in J}) = \mathbb{P}(\psi_{k_1}(\boldsymbol{w})^T (\phi(\boldsymbol{x}_{j_{1,1}}) - \phi(\boldsymbol{x}_{j_{1,2}})) = 0 \mid \{\boldsymbol{x}_i\}_{i \in J}) = 0,$$

which follows from the fact that $\psi_{k_1}(\boldsymbol{w})$ is nonzero: recall from the fact presented above that $\mathrm{proj}_{E_0}(\boldsymbol{w}) \neq \boldsymbol{0}$, so $\psi_k(\boldsymbol{w}) \neq \boldsymbol{0}$ for any $k \in [K]$.

Now, instead suppose without loss of generality that $j_{1,1} \notin J$ and $j_{1,2} \in J$ (an identical argument holds for $j_{1,1} \in J$ and $j_{1,2} \notin J$). Then by similar arguments as above,

$$\mathbb{P}(\boldsymbol{w}^T \boldsymbol{r}_1 = 0 \mid \{\boldsymbol{x}_i\}_{i \in J}) = \mathbb{P}(\psi_{k_1}(\boldsymbol{w})^T (\phi(\boldsymbol{x}_{j_{1,1}}) - \phi(\boldsymbol{x}_{j_{1,2}})) = 0 \mid \{\boldsymbol{x}_i\}_{i \in J})$$

$$= \mathbb{P}(\psi_{k_1}(\boldsymbol{w})^T \phi(\boldsymbol{x}_{j_{1,1}}) - \psi_{k_1}(\boldsymbol{w})^T \phi(\boldsymbol{x}_{j_{1,2}}) = 0 \mid \boldsymbol{x}_{j_{1,2}} \cup \{\boldsymbol{x}_i\}_{i \in J})$$
$$= \mathbb{P}_X(\psi_{k_1}(\boldsymbol{w})^T \phi(\boldsymbol{x}) - c = 0 \mid \boldsymbol{x}_{j_{1,2}} \cup \{\boldsymbol{x}_i\}_{i \in J}) \quad \text{where } c \text{ is a constant.}$$
$$= 0.$$

The last equality follows since $\psi_{k_1}(\boldsymbol{w}) \neq \boldsymbol{0}$ and so $\psi_{k_1}(\boldsymbol{w})^T \phi(\boldsymbol{x}) - c$ is a nontrivial polynomial in $\boldsymbol{x}$, along with the fact that $p_X$ is absolutely continuous.

In either scenario, $\mathbb{P}(\boldsymbol{w}^T \boldsymbol{r}_1 = 0 \mid \{\boldsymbol{x}_i\}_{i \in J}) = 0$. Hence, when conditioned on $\{\boldsymbol{x}_i\}_{i \in J}$, $\boldsymbol{r}_1$ includes a component orthogonal to $\mathrm{rowsp}(\boldsymbol{\Gamma}^{(1)})$ and therefore does not lie in this row space. In other words,

$$\mathbb{P}\Big(\mathrm{rank}\Big(\big[(\boldsymbol{\Gamma}^{(1)})^T \quad \boldsymbol{r}_1\big]\Big) = Kd + 1 \mid \{\boldsymbol{x}_i\}_{i \in J}\Big) = 1.$$

This is true for any $\{\boldsymbol{x}_i\}_{i \in J}$ resulting in $\boldsymbol{\Gamma}^{(1)}$ being full-rank, which we know occurs with probability 1 and so by marginalizing we have $\mathbb{P}\Big(\mathrm{rank}\Big(\big[(\boldsymbol{\Gamma}^{(1)})^T \quad \boldsymbol{r}_1\big]\Big) = Kd + 1\Big) = 1$.

If $d > 1$, let $m < D$ be given and suppose by induction that

$$\mathbb{P}\Big(\mathrm{rank}\Big(\big[(\boldsymbol{\Gamma}^{(1)})^T \quad \boldsymbol{r}_1 \quad \cdots \quad \boldsymbol{r}_m\big]\Big) = Kd + m\Big) = 1.$$

Let $\boldsymbol{w}$ be a vector in $(\mathrm{rowsp}(\boldsymbol{\Gamma}^{(1)}) \cup \mathrm{span}(\{\boldsymbol{r}_i\}_{i=1}^m))^\perp$ not equal to the zero vector. Note that $\boldsymbol{w} \in \mathrm{rowsp}(\boldsymbol{\Gamma}^{(1)})^\perp$ as well, and so by the above, $\mathrm{proj}_{E_0}(\boldsymbol{w}) \neq \boldsymbol{0}$ and so $\psi_k(\boldsymbol{w}) \neq \boldsymbol{0}$ for any $k \in [K]$. Reusing notation, let

$$J = \{j \colon j \in [n], \exists k \in [K], i \in [d] \text{ s.t. } \boldsymbol{S}_k^{(1)}[i,j] \neq 0\} \cup \{j \colon j \in [n], \exists i \in [m] \; j = j_{i,1} \vee j = j_{i,2}\}$$

denote the set of all item indices in $\widehat{\boldsymbol{\Gamma}}$ measured up through and including the $m$th measurement of $\boldsymbol{\Gamma}^{(2)}$. Consider row $m + 1$ of $\boldsymbol{\Gamma}^{(2)}$. By the incremental assumption, at least one of $j_{m+1,1}$ or $j_{m+1,2}$ is not found in $J$. First suppose that both are not found in $J$. Then

$$\mathbb{P}(\boldsymbol{w}^T \boldsymbol{r}_{m+1} = 0 \mid \{\boldsymbol{x}_i\}_{i \in J}) = \mathbb{P}(\psi_{k_{m+1}}(\boldsymbol{w})^T (\phi(\boldsymbol{x}_{j_{m+1,1}}) - \phi(\boldsymbol{x}_{j_{m+1,2}})) = 0 \mid \{\boldsymbol{x}_i\}_{i \in J})$$
$$= 0,$$

due to a similar argument as above.

Now, instead suppose without loss of generality that $j_{m+1,1} \notin J$ and $j_{m+1,2} \in J$ (an identical argument holds for $j_{m+1,1} \in J$ and $j_{m+1,2} \notin J$). Then by similar arguments as above,

$$\mathbb{P}(\boldsymbol{w}^T \boldsymbol{r}_{m+1} = 0 \mid \{\boldsymbol{x}_i\}_{i \in J}) = \mathbb{P}(\psi_{k_{m+1}}(\boldsymbol{w})^T (\phi(\boldsymbol{x}_{j_{m+1,1}}) - \phi(\boldsymbol{x}_{j_{m+1,2}})) = 0 \mid \{\boldsymbol{x}_i\}_{i \in J})$$
$$= \mathbb{P}(\psi_{k_{m+1}}(\boldsymbol{w})^T \phi(\boldsymbol{x}_{j_{m+1,1}}) - \psi_{k_{m+1}}(\boldsymbol{w})^T \phi(\boldsymbol{x}_{j_{m+1,2}}) = 0 \mid \boldsymbol{x}_{j_{m+1,2}} \cup \{\boldsymbol{x}_i\}_{i \in J})$$
$$= \mathbb{P}_X(\psi_{k_{m+1}}(\boldsymbol{w})^T \phi(\boldsymbol{x}) - c = 0 \mid \boldsymbol{x}_{j_{m+1,2}} \cup \{\boldsymbol{x}_i\}_{i \in J}) \quad \text{where } c \text{ is a constant.}$$
$$= 0.$$

The last equality follows since $\psi_{k_{m+1}}(\boldsymbol{w}) \neq \boldsymbol{0}$ and so $\psi_{k_{m+1}}(\boldsymbol{w})^T \phi(\boldsymbol{x}) - c$ is a nontrivial polynomial in $\boldsymbol{x}$.

In either scenario, $\mathbb{P}(\boldsymbol{w}^T \boldsymbol{r}_{m+1} = 0 \mid \{\boldsymbol{x}_i\}_{i \in J}) = 0$. Hence, when conditioned on $\{\boldsymbol{x}_i\}_{i \in J}$, $\boldsymbol{r}_{m+1}$ includes a component orthogonal to $\mathrm{rowsp}(\boldsymbol{\Gamma}^{(1)}) \cup \mathrm{span}(\{\boldsymbol{r}_i\}_{i=1}^m)$ and therefore does not lie in the span of the previous rows. In other words,

$$\mathbb{P}\Big(\mathrm{rank}\Big(\big[(\boldsymbol{\Gamma}^{(1)})^T \quad \boldsymbol{r}_1 \quad \cdots \quad \boldsymbol{r}_{m+1}\big]\Big) = Kd + m + 1 \mid \{\boldsymbol{x}_i\}_{i \in J}\Big) = 1.$$

This is true for any $\{\boldsymbol{x}_i\}_{i \in J}$ satisfying $\mathrm{rank}\Big(\big[(\boldsymbol{\Gamma}^{(1)})^T \quad \boldsymbol{r}_1 \quad \cdots \quad \boldsymbol{r}_m\big]\Big) = Kd + m$, which by inductive assumption occurs with probability 1 and so by marginalizing we have $\mathbb{P}\Big(\mathrm{rank}\Big(\big[(\boldsymbol{\Gamma}^{(1)})^T \quad \boldsymbol{r}_1 \quad \cdots \quad \boldsymbol{r}_{m+1}\big]\Big) = Kd + m + 1\Big) = 1$. Taking $m = D - 1$, we have with probability 1 that

$$\big[(\boldsymbol{\Gamma}^{(1)})^T \quad \boldsymbol{r}_1 \quad \cdots \quad \boldsymbol{r}_D\big] = \big[(\boldsymbol{\Gamma}^{(1)})^T \quad (\boldsymbol{\Gamma}^{(2)})^T\big] = \widehat{\boldsymbol{\Gamma}}^T$$

is full-rank, and so $\boldsymbol{\Gamma}$ has full column rank.

## C.5   Proof of results in the single user case

**Necessary conditions:**   When $K = 1$, the three conditions in Proposition 2.1 are equivalent to $\text{rank}(S) \geq D + d$: in the single user case, $S_T = S$, and so (c) directly states that $\text{rank}(S) \geq D + d$. (b) also translates to $\text{rank}(S) \geq D + d$ when $K = 1$. The condition $\text{rank}(S) \geq d$ in (a) is subsumed by $\text{rank}(S) \geq D + d$.

**Sufficient conditions:**   Suppose $\text{rank}(S) \geq D + d$. By definition, there exists a set of $D + d$ linearly independent rows in $S$: denote the $D + d \times n$ submatrix of $S$ defined by these rows as $S'$. Since $S'$ is full row rank by construction, by Corollary C.4.1 there exists a permutation $P$ such that $PS'$ is incremental. Define $S^{(1)}$ as the first $d$ rows of $PS'$, and $S^{(2)}$ as the remaining $D$ rows of $PS'$. Then $\begin{bmatrix} S^{(1)} \\ S^{(2)} \end{bmatrix}$ satisfies the conditions of Proposition 2.2 and therefore if each $x_i$ is sampled i.i.d. according to $p_X$ then $PS'[\, x_\otimes^T \ x^T \,]$ has full column rank with probability 1 — and therefore full row rank since it is square. Since the rows of this matrix are simply a permuted subset of the rows in $S[\, x_\otimes^T \ x^T \,]$, we also have that $S[\, x_\otimes^T \ x^T \,]$ has rank $D + d$ and therefore full column rank with probability 1.

**Random construction:**   We will use the results of Appendix C.2 to choose a number of random measurements and items such that a single-user selection matrix $S$ has rank at least $D + d$ with high probability, to satisfy the conditions described above. Let failure probability $0 < \delta < 1$ be given, and suppose $S$ is constructed by drawing $m_T$ item index pairs uniformly and independent at random among $n$ items. After drawing a single measurement, $S$ will immediately have rank 1. According to Theorem C.8 with $r_0 = 1$ and $r = D + d$, with probability at least $1 - \delta$ the total number of additional required measurements $M$ is less than $\left( 1 + \ln \frac{1}{\delta} \right) \left( \sum_{i=2}^{D+d} \frac{1}{1 - \frac{i(i-1)}{n(n-1)}} \right)$.

To make this quantity more manageable, note that $\frac{1}{1 - \frac{i(i-1)}{n(n-1)}}$ is an increasing function of $i$. Hence, if we choose a constant $U$ such that $\frac{1}{1 - \frac{(D+d)(D+d-1)}{n(n-1)}} \leq U$, then for every $2 \leq i \leq D + d$ we also have $\frac{1}{1 - \frac{i(i-1)}{n(n-1)}} \leq U$. To arrive at such a $U$, suppose that $(n - 1)^2 \geq (1 + \gamma)(D + d)^2$ for some $\gamma > 0$, i.e., $n \geq \sqrt{1 + \gamma}(D + d) + 1$. Then

$$\frac{(D + d)(D + d - 1)}{n(n - 1)} \leq \frac{(D + d)^2}{(n - 1)^2} \leq \frac{1}{1 + \gamma},$$

and so

$$\frac{1}{1 - \frac{(D+d)(D+d-1)}{n(n-1)}} \leq \frac{1}{1 - \frac{1}{1+\gamma}} = \frac{1 + \gamma}{\gamma},$$

and we can set $U = \frac{1+\gamma}{\gamma}$. Therefore,

$$\sum_{i=2}^{D+d} \frac{1}{1 - \frac{i(i-1)}{n(n-1)}} \leq \frac{1 + \gamma}{\gamma}(D + d - 1),$$

and so with probability at least $1 - \delta$, $M < \left( 1 + \ln \frac{1}{\delta} \right) \frac{1+\gamma}{\gamma}(D + d - 1)$. To choose a convenient value for $\gamma$, we can let $\gamma = \frac{1}{2}(1 + \sqrt{5})$, in which case $\frac{\gamma+1}{\gamma} = \sqrt{1 + \gamma} = \frac{1}{2}(1 + \sqrt{5}) \approx 1.62$. So, if $n \geq \frac{1}{2}(1 + \sqrt{5})(D + d) + 1$, and $m_T \geq \left\lceil \frac{1}{2}(1 + \sqrt{5}) \left( 1 + \ln \frac{1}{\delta} \right)(D + d - 1) \right\rceil + 1$ random measurements are taken, then with probability at least $1 - \delta$, $\text{rank}(S) \geq D + d$. Hence, with high probability, $n = \Omega(D + d)$ and $m_T = \Omega(D + d)$ random measurements result in a selection matrix $S$ with rank at least $D + d$. Once such a matrix with rank at least $D + d$ is fixed after sampling, if each $x_i$ is sampled i.i.d. according to $p_X$ then as described above $\Gamma$ will be full column rank with probability 1. Together, the process of independently sampling $S$ and $\{x_i\}_{i=1}^{n}$ results in $\Gamma$ having full column rank with high probability.

## C.6   Constructions and counterexamples

**Counterexample for necessary conditions being sufficient:**   Below we demonstrate a counterexample where the conditions in Proposition 2.1 are met, but the system results in a $\Gamma$ matrix that is

not full column rank. In this example, $d = 2$ (and so $D = 3$), $K = 3$, $m_k = d + D/K = 3$, and $n = D + d + 1 = 6$. Consider the selection matrices below:

$$S_1 = \begin{bmatrix} 0 & 0 & 1 & 0 & -1 & 0 \\ 1 & 0 & 0 & 0 & 0 & -1 \\ 1 & 0 & 0 & -1 & 0 & 0 \end{bmatrix}$$

$$S_2 = \begin{bmatrix} 0 & 1 & 0 & 0 & -1 & 0 \\ 1 & 0 & -1 & 0 & 0 & 0 \\ 0 & 0 & 0 & 0 & 1 & -1 \end{bmatrix}$$

$$S_3 = \begin{bmatrix} 0 & 1 & 0 & 0 & -1 & 0 \\ 1 & 0 & -1 & 0 & 0 & 0 \\ 0 & 1 & 0 & 0 & 0 & -1 \end{bmatrix}.$$

By inspection, $\mathrm{rank}(S_k) = 3 \geq d$ for each $k$, $\sum_k \mathrm{rank}(S_k) = 9 = D + dK$, and $\mathrm{rank}(S_T) = 5 = D + d$. Yet, when we numerically sample $x_i \sim \mathcal{N}(0, I)$ and verify that $\mathrm{rank}(S_k X^T) = 2 = d$, $\sum_k \mathrm{rank}(S_k \begin{bmatrix} X_\otimes^T & X^T \end{bmatrix}) = D + dK$, and $\mathrm{rank}(S_T \begin{bmatrix} X_\otimes^T & X^T \end{bmatrix}) = 5$, we still find that $\Gamma$ is rank deficient. This counterexample illustrates that the conditions of Proposition 2.1 are *not sufficient* for identifiability.

**Incremental condition construction:** Here we construct a selection matrix scheme that satisfies the properties of Proposition 2.2 while only using the minimal number of measurements per user (i.e., $m_k = d + D/K$) and items (i.e., $n = D + d + 1$). For each $k \in [K]$, let

$$S_k^{(1)} = d \text{ rows} \left\{ \begin{bmatrix} 1 & -1 & 0 & \cdots & 0 & 0 & 0 \\ 0 & 1 & -1 & \cdots & 0 & 0 & 0 \\ 0 & 0 & 1 & \cdots & 0 & 0 & 0 \\ \vdots & \vdots & \vdots & \vdots & \vdots & \vdots & \vdots & \mathbf{0}_{d,D} \\ 0 & 0 & 0 & \cdots & -1 & 0 & 0 \\ 0 & 0 & 0 & \cdots & 1 & -1 & 0 \\ 0 & 0 & 0 & \cdots & 0 & 1 & -1 \end{bmatrix} \right.,$$

and define

$$S^{(2)} = D \text{ rows} \left\{ \begin{bmatrix} \mathbf{0}_{D,d} & \begin{matrix} 1 & -1 & 0 & \cdots & 0 & 0 & 0 \\ 0 & 1 & -1 & \cdots & 0 & 0 & 0 \\ 0 & 0 & 1 & \cdots & 0 & 0 & 0 \\ \vdots & \vdots & \vdots & \vdots & \vdots & \vdots & \vdots \\ 0 & 0 & 0 & \cdots & -1 & 0 & 0 \\ 0 & 0 & 0 & \cdots & 1 & -1 & 0 \\ 0 & 0 & 0 & \cdots & 0 & 1 & -1 \end{matrix} \end{bmatrix} \right..$$

By observation, for each $k \in [K]$ we have

$$\begin{bmatrix} S_k^{(1)} \\ S^{(2)} \end{bmatrix} = D + d \text{ rows} \left\{ \begin{bmatrix} 1 & -1 & 0 & \cdots & 0 & 0 & 0 \\ 0 & 1 & -1 & \cdots & 0 & 0 & 0 \\ 0 & 0 & 1 & \cdots & 0 & 0 & 0 \\ \vdots & \vdots & \vdots & \vdots & \vdots & \vdots & \vdots \\ 0 & 0 & 0 & \cdots & -1 & 0 & 0 \\ 0 & 0 & 0 & \cdots & 1 & -1 & 0 \\ 0 & 0 & 0 & \cdots & 0 & 1 & -1 \end{bmatrix} \right.,$$

which by observation is incremental. Assuming for simplicity that $D/K$ is an integer, for each $k \in [K]$ let $S_k^{(2)}$ be the submatrix defined by rows $(k-1)(D/K) + 1$ through $k(D/K)$ of $S^{(2)}$, i.e., each user is allotted $D/K$ nonoverlapping rows of $S^{(2)}$. Finally, for each $k \in [K]$ let

$$S_k = \begin{bmatrix} S_k^{(1)} \\ S_k^{(2)} \end{bmatrix}.$$

By observation each $S_k^{(1)}$ has rank $d$, and by construction each $\begin{bmatrix} S_k^{(1)} \\ S^{(2)} \end{bmatrix}$ is incremental, and therefore the conditions of Proposition 2.2 are satisfied.

**Counterexample for incremental sufficiency conditions being exhaustive:** Below we demonstrate a counterexample where the matrix $\mathbf{\Gamma}$ is full column rank, yet the conditions in Proposition 2.2 are not met, demonstrating that they are not an exhaustive set of sufficiency conditions.

In this example, $d = 2$, $K = 2$, and $n = 6$, with selection matrices given by

$$
\mathbf{S}_1 = \begin{matrix} (1a) \\ (1b) \\ (1c) \\ (1d) \end{matrix} \begin{bmatrix} 1 & -1 & 0 & 0 & 0 & 0 \\ 0 & 1 & -1 & 0 & 0 & 0 \\ 0 & 0 & 1 & -1 & 0 & 0 \\ 0 & 0 & 0 & 1 & -1 & 0 \end{bmatrix}
$$

$$
\mathbf{S}_2 = \begin{matrix} (2a) \\ (2b) \\ (2c) \end{matrix} \begin{bmatrix} 1 & -1 & 0 & 0 & 0 & 0 \\ 0 & 0 & 1 & -1 & 0 & 0 \\ 0 & 0 & 0 & 0 & 1 & -1 \end{bmatrix}.
$$

By observation, one cannot partition these selection matrices according to the conditions in Proposition 2.2. To see this, we can attempt to partition these selection matrices according to these conditions. First note that rows (1a) and (2a) are equal, as well as rows (1c) and (2b), which implies that (1a,c) and (2a,b) must belong to $\mathbf{S}_1^{(1)}$ and $\mathbf{S}_2^{(1)}$ respectively. Otherwise, there would exist at least one repeated pair in the matrix $\begin{bmatrix} \mathbf{S}_k^{(1)} \\ \mathbf{S}^{(2)} \end{bmatrix}$ for $k = 1$ or $k = 2$, which would violate condition (b) of Proposition 2.2.

Therefore, $\mathbf{S}^{(2)}$ must consist of rows (1b), (1d), and (2c). While (1d) is surely incremental with respect to $\mathbf{S}_1^{(1)}$, and (2c) is surely incremental with respect to $\mathbf{S}_2^{(1)}$, (1b) overlaps with both (1a) and (1c) and therefore cannot possibly be incremental with respect to $\mathbf{S}_1^{(1)}$, and hence the conditions in Proposition 2.2 are not met.

Yet, in simulation we find with normally distributed items that the $\mathbf{\Gamma}$ matrix resulting from the above selection scheme is in fact full column rank.

## C.7 Conjectured sufficiency conditions

We conjecture that a set of conditions similar to that of Proposition 2.2 are sufficient for identifiability under items sampled according to a distribution that is absolutely continuous with respect to the Lebesgue measure. We list these conditions below:

**Conjectured sufficiency conditions:** Let $K \geq 1$, and suppose $m_k > d \ \forall \ k \in [K]$, $m_T = D + dK$, and $n \geq D + d + 1$. Suppose that for each $k \in [K]$, there exists a $d \times n$ selection matrix $\mathbf{S}_k^{(1)}$ and $m_k - d \times n$ selection matrix $\mathbf{S}_k^{(2)}$ such that $\mathbf{S}_k = [\,(\mathbf{S}_k^{(1)})^T \ (\mathbf{S}_k^{(2)})^T\,]^T$, and that the following are true:

(a) For all $k \in [K]$, $\mathrm{rank}(\mathbf{S}_k^{(1)}) = d$

(b) Defining the $D \times n$ selection matrix $\mathbf{S}^{(2)}$ as $\mathbf{S}^{(2)} := [\,(\mathbf{S}_1^{(2)})^T \ \cdots \ (\mathbf{S}_K^{(2)})^T\,]^T$, for each $k \in [K]$, $\begin{bmatrix} \mathbf{S}_k^{(1)} \\ \mathbf{S}^{(2)} \end{bmatrix}$ is full row rank

Intuitively, these conditions replace the permutation condition in Proposition 2.2 with a more general condition concerning only the rank. In fact, due to Corollary C.4.1, condition (b) in Proposition 2.2 implies the second condition above. These conditions capture the intuition that each user is allocated $d$ independent measurements to identity their own pseudo-ideal point (condition (a) above), and then collectively the set of users answers an additional $D$ independent measurements to identify the metric (condition (b) above). As long as the individual measurements do not "overlap" with the collective measurements (captured by the rank condition in condition (b)), then the collective set of measurements should be rich enough to identify the metric and all pseudo-ideal points, even if each individual user has overlapping measurements in their $\mathbf{S}_k^{(1)}$ selection matrices. Empirically, we find that the above conditions appear to be sufficient for identifiability, at least with normally distributed items.

Furthermore, the above conditions would provide a convenient avenue to study randomly selected unquantized measurements among multiple users. As we demonstrated in Appendix C.5, we can use Theorem C.8 to bound the number of measurements needed for a selection matrix to be full-rank. We can apply these tools to the multiuser case as follows: first, sample on the order $\Omega(D)$ randomly

selected pairs among $\Omega(D + d)$ items to construct a selection matrix $\boldsymbol{S}^{(2)}$ that is rank $D$ with high probability. Then, using $\boldsymbol{S}^{(2)}$ as a seed matrix in Theorem C.8, sample on the order of $d$ additional measurements per user in selection matrix $\boldsymbol{S}_k^{(1)}$, so that for each individual user conditions (a-b) above are satisfied with high probability. The key insight is that if the measurements in $\boldsymbol{S}^{(2)}$ are evenly distributed between users, and the number of samples taken in $\boldsymbol{S}^{(2)}$ and each $\boldsymbol{S}_k^{(1)}$ is fixed ahead of time (and non-adaptive), then the above sampling process is simply equivalent to sampling pairs uniformly at random for each individual user. Yet, with high probability they should satisfy the above conditions, which if are sufficient for identifiability should result in a measurement matrix $\boldsymbol{\Gamma}$ that is full column rank.

# D  Proofs of prediction and generalization results

## D.1  Proof of Theorem 3.1

We start by expanding the excess risk between the empirical and true optimizers:

$$
\begin{aligned}
R(\widehat{\boldsymbol{M}}, &\{\widehat{\boldsymbol{v}}_k\}_{k=1}^K) - R(\boldsymbol{M}^*, \{\boldsymbol{v}_k^*\}_{k=1}^K) \\
&= R(\widehat{\boldsymbol{M}}, \{\widehat{\boldsymbol{v}}_k\}_{k=1}^K) - \widehat{R}(\widehat{\boldsymbol{M}}, \{\widehat{\boldsymbol{v}}_k\}_{k=1}^K) + \widehat{R}(\widehat{\boldsymbol{M}}, \{\widehat{\boldsymbol{v}}_k\}_{k=1}^K) \\
&\qquad - \widehat{R}(\boldsymbol{M}^*, \{\boldsymbol{v}_k^*\}_{k=1}^K) + \widehat{R}(\boldsymbol{M}^*, \{\boldsymbol{v}_k^*\}_{k=1}^K) - R(\boldsymbol{M}^*, \{\boldsymbol{v}_k^*\}_{k=1}^K) \\
&\leq R(\widehat{\boldsymbol{M}}, \{\widehat{\boldsymbol{v}}_k\}_{k=1}^K) - \widehat{R}(\widehat{\boldsymbol{M}}, \{\widehat{\boldsymbol{v}}_k\}_{k=1}^K) + \widehat{R}(\boldsymbol{M}^*, \{\boldsymbol{v}_k^*\}_{k=1}^K) - R(\boldsymbol{M}^*, \{\boldsymbol{v}_k^*\}_{k=1}^K) \qquad (38) \\
&\leq 2 \sup_{\boldsymbol{M}, \{\boldsymbol{v}_k\}_{k=1}^K} |\widehat{R}(\boldsymbol{M}, \{\boldsymbol{v}_k\}_{k=1}^K) - R(\boldsymbol{M}, \{\boldsymbol{v}_k\}_{k=1}^K)| \\
&\leq 2 \mathbb{E} \left[ \sup_{\boldsymbol{M}, \{\boldsymbol{v}_k\}_{k=1}^K} \left| \widehat{R}(\boldsymbol{M}, \{\boldsymbol{v}_k\}_{k=1}^K) - R(\boldsymbol{M}, \{\boldsymbol{v}_k\}_{k=1}^K) \right| \right] + \sqrt{\frac{8L^2\gamma^2 \log(2/\delta)}{|\mathcal{S}|}} \qquad (39)
\end{aligned}
$$

where (38) follows from the fact that $\widehat{\boldsymbol{M}}, \{\boldsymbol{v}_k\}$ are the empirical risk minimizers, and (39) follows from the Bounded differences inequality (also known as McDiarmid's Inequality, see [48]) since for two data points $(p, k, y_p)$ and $(p', k', y_{p'})$ we have that

$$
\begin{aligned}
\ell \left( y_p (\boldsymbol{x}_i^T \boldsymbol{M} \boldsymbol{x}_i - \boldsymbol{x}_j^T \boldsymbol{M} \boldsymbol{x}_j + (\boldsymbol{x}_i - \boldsymbol{x}_j)^T \boldsymbol{v}_k) \right) \\
- \ell \left( y_{p'} (\boldsymbol{x}_{i'}^T \boldsymbol{M} \boldsymbol{x}_{i'} - \boldsymbol{x}_{j'}^T \boldsymbol{M} \boldsymbol{x}_{j'} + (\boldsymbol{x}_{i'} - \boldsymbol{x}_{j'})^T \boldsymbol{v}_{k'}) \right) \leq 2L\gamma
\end{aligned}
$$

by Lipschitz-ness of $\ell$ and the definition of $\gamma$ in eq. (5). The expectation in eq. (39) is with respect to the dataset $\mathcal{S}$. Next, using symmetrization, contraction, and introducing Rademacher random variables $\varepsilon_p$ with $\mathbb{P}(\varepsilon_p = 1) = \mathbb{P}(\varepsilon_p = -1) = 1/2$ for all data points in $\mathcal{S}$, we have that (with expectations taken with respect to both $\{\varepsilon_p\}$ and $\mathcal{S}$)

$$
\mathbb{E} \left[ \sup_{\boldsymbol{M}, \{\boldsymbol{v}_k\}_{k=1}^K} |\widehat{R}(\boldsymbol{M}, \{\boldsymbol{v}_k\}_{k=1}^K) - R(\boldsymbol{M}, \{\boldsymbol{v}_k\}_{k=1}^K)| \right]
$$

$$
\leq \frac{2L}{|\mathcal{S}|} \mathbb{E} \left[ \sup_{\boldsymbol{M}, \{\boldsymbol{v}_k\}_{k=1}^K} \left| \sum_{\mathcal{S}} \varepsilon_p y_p (\boldsymbol{x}_i^T \boldsymbol{M} \boldsymbol{x}_i - \boldsymbol{x}_j^T \boldsymbol{M} \boldsymbol{x}_j + (\boldsymbol{x}_i - \boldsymbol{x}_j)^T \boldsymbol{v}_k) \right| \right]
$$

$$
= \frac{2L}{|\mathcal{S}|} \mathbb{E} \left[ \sup_{\boldsymbol{M}, \{\boldsymbol{v}_k\}_{k=1}^K} \left| \sum_{\mathcal{S}} \varepsilon_p (\boldsymbol{x}_i^T \boldsymbol{M} \boldsymbol{x}_i - \boldsymbol{x}_j^T \boldsymbol{M} \boldsymbol{x}_j + (\boldsymbol{x}_i - \boldsymbol{x}_j)^T \boldsymbol{v}_k) \right| \right] \quad \text{since } \mathbb{P}(\varepsilon_p y_p = 1) = 1/2
$$

$$
= \frac{2L}{|\mathcal{S}|} \mathbb{E} \left[ \sup_{\boldsymbol{M}, \{\boldsymbol{v}_k\}_{k=1}^K} \left| \left\langle \sum_{\mathcal{S}} \varepsilon_p \begin{bmatrix} \boldsymbol{x}_i \boldsymbol{x}_i^T - \boldsymbol{x}_j \boldsymbol{x}_j^T & \boldsymbol{0} & \cdots & \underbrace{\boldsymbol{x}_i - \boldsymbol{x}_j}_{\text{column } d + k} & \cdots & \boldsymbol{0} \end{bmatrix}, \begin{bmatrix} \boldsymbol{M} & \boldsymbol{v}_1 & \cdots & \boldsymbol{v}_K \end{bmatrix} \right\rangle \right| \right]
$$
$$
\tag{40}
$$

$$
\overset{\text{Cauchy-Schwarz}}{\leq} \frac{2L}{|\mathcal{S}|} \mathbb{E} \left[ \sup_{\boldsymbol{M}, \{\boldsymbol{v}_k\}_{k=1}^K} \left\| \sum_{\mathcal{S}} \varepsilon_p \begin{bmatrix} \boldsymbol{x}_i \boldsymbol{x}_i^T - \boldsymbol{x}_j \boldsymbol{x}_j^T & \boldsymbol{0} & \cdots & \underbrace{\boldsymbol{x}_i - \boldsymbol{x}_j}_{\text{column } d + k} & \cdots & \boldsymbol{0} \end{bmatrix} \right\|_F \left\| \begin{bmatrix} \boldsymbol{M} & \boldsymbol{v}_1 & \cdots & \boldsymbol{v}_K \end{bmatrix} \right\|_F \right]
$$

$$
= \frac{2L}{|\mathcal{S}|} \left( \sup_{\boldsymbol{M}, \{\boldsymbol{v}_k\}_{k=1}^K} \left\| \begin{bmatrix} \boldsymbol{M} & \boldsymbol{v}_1 & \cdots & \boldsymbol{v}_K \end{bmatrix} \right\|_F \right) \mathbb{E} \left[ \left\| \sum_{\mathcal{S}} \varepsilon_p \begin{bmatrix} \text{vec}(\boldsymbol{x}_i \boldsymbol{x}_i^T - \boldsymbol{x}_j \boldsymbol{x}_j^T) \\ \boldsymbol{x}_i - \boldsymbol{x}_j \end{bmatrix} \right\|_2 \right]
$$

$$\leq \frac{2L}{|\mathcal{S}|}\sqrt{\lambda_F^2 + K\lambda_v^2}\;\mathbb{E}\left[\left\|\sum_{\mathcal{S}}\varepsilon_p\begin{bmatrix}\mathrm{vec}(\boldsymbol{x}_i\boldsymbol{x}_i^T - \boldsymbol{x}_j\boldsymbol{x}_j^T)\\ \boldsymbol{x}_i - \boldsymbol{x}_j\end{bmatrix}\right\|_2\right]. \tag{41}$$

Next we employ Matrix Bernstein to bound

$$\mathbb{E}\left[\left\|\sum_{\mathcal{S}}\varepsilon_p\begin{bmatrix}\mathrm{vec}(\boldsymbol{x}_i\boldsymbol{x}_i^T - \boldsymbol{x}_j\boldsymbol{x}_j^T)\\ \boldsymbol{x}_i - \boldsymbol{x}_j\end{bmatrix}\right\|_2\right],$$

which is a sum of zero-mean random vectors in $\mathbb{R}^{d^2+d}$ (recall that each $\varepsilon_p \in \{-1,1\}$ with equal probability). First note that under the assumption $\|\boldsymbol{x}_i\| \leq B\;\forall\,i$, we have

$$\left\|\begin{bmatrix}\mathrm{vec}(\boldsymbol{x}_i\boldsymbol{x}_i^T - \boldsymbol{x}_j\boldsymbol{x}_j^T)\\ \boldsymbol{x}_i - \boldsymbol{x}_j\end{bmatrix}\right\|_2^2 = \|\boldsymbol{x}_i\boldsymbol{x}_i^T - \boldsymbol{x}_j\boldsymbol{x}_j^T\|_F^2 + \|\boldsymbol{x}_i - \boldsymbol{x}_j\|_2^2$$

$$\leq (\|\boldsymbol{x}_i\boldsymbol{x}_i^T\|_F + \|\boldsymbol{x}_j\boldsymbol{x}_j^T\|_F)^2 + (\|\boldsymbol{x}_i\|_2 + \|\boldsymbol{x}_j\|_2)^2$$

$$\leq (\|\boldsymbol{x}_i\|_2^2 + \|\boldsymbol{x}_j\|_2^2)^2 + (\|\boldsymbol{x}_i\|_2 + \|\boldsymbol{x}_j\|_2)^2$$

$$\leq 4(B^4 + B^2),$$

and therefore

$$\left\|\begin{bmatrix}\mathrm{vec}(\boldsymbol{x}_i\boldsymbol{x}_i^T - \boldsymbol{x}_j\boldsymbol{x}_j^T)\\ \boldsymbol{x}_i - \boldsymbol{x}_j\end{bmatrix}\right\|_2 \leq 2B\sqrt{B^2 + 1} =: C_B.$$

We also have

$$\left\|\mathbb{E}\left[\sum_{\mathcal{S}}\left(\varepsilon_p\begin{bmatrix}\mathrm{vec}(\boldsymbol{x}_i\boldsymbol{x}_i^T - \boldsymbol{x}_j\boldsymbol{x}_j^T)\\ \boldsymbol{x}_i - \boldsymbol{x}_j\end{bmatrix}\right)^T\left(\varepsilon_p\begin{bmatrix}\mathrm{vec}(\boldsymbol{x}_i\boldsymbol{x}_i^T - \boldsymbol{x}_j\boldsymbol{x}_j^T)\\ \boldsymbol{x}_i - \boldsymbol{x}_j\end{bmatrix}\right)\right]\right\|$$

$$= \mathbb{E}\left[\sum_{\mathcal{S}}\begin{bmatrix}\mathrm{vec}(\boldsymbol{x}_i\boldsymbol{x}_i^T - \boldsymbol{x}_j\boldsymbol{x}_j^T)\\ \boldsymbol{x}_i - \boldsymbol{x}_j\end{bmatrix}^T\begin{bmatrix}\mathrm{vec}(\boldsymbol{x}_i\boldsymbol{x}_i^T - \boldsymbol{x}_j\boldsymbol{x}_j^T)\\ \boldsymbol{x}_i - \boldsymbol{x}_j\end{bmatrix}\right]$$

$$= \mathbb{E}\left[\sum_{\mathcal{S}}(\|\boldsymbol{x}_i\boldsymbol{x}_i^T - \boldsymbol{x}_j\boldsymbol{x}_j^T\|_F^2 + \|\boldsymbol{x}_i - \boldsymbol{x}_j\|_2^2)\right]$$

$$\leq 4(B^4 + B^2)|\mathcal{S}|$$

$$= C_B^2|\mathcal{S}|,$$

and

$$\left\|\mathbb{E}\left[\sum_{\mathcal{S}}\left(\varepsilon_p\begin{bmatrix}\mathrm{vec}(\boldsymbol{x}_i\boldsymbol{x}_i^T - \boldsymbol{x}_j\boldsymbol{x}_j^T)\\ \boldsymbol{x}_i - \boldsymbol{x}_j\end{bmatrix}\right)\left(\varepsilon_p\begin{bmatrix}\mathrm{vec}(\boldsymbol{x}_i\boldsymbol{x}_i^T - \boldsymbol{x}_j\boldsymbol{x}_j^T)\\ \boldsymbol{x}_i - \boldsymbol{x}_j\end{bmatrix}\right)^T\right]\right\|$$

$$= \left\|\mathbb{E}\left[\sum_{\mathcal{S}}\begin{bmatrix}\mathrm{vec}(\boldsymbol{x}_i\boldsymbol{x}_i^T - \boldsymbol{x}_j\boldsymbol{x}_j^T)\\ \boldsymbol{x}_i - \boldsymbol{x}_j\end{bmatrix}\begin{bmatrix}\mathrm{vec}(\boldsymbol{x}_i\boldsymbol{x}_i^T - \boldsymbol{x}_j\boldsymbol{x}_j^T)\\ \boldsymbol{x}_i - \boldsymbol{x}_j\end{bmatrix}^T\right]\right\|$$

$$\leq \mathbb{E}\left[\left\|\sum_{\mathcal{S}}\begin{bmatrix}\mathrm{vec}(\boldsymbol{x}_i\boldsymbol{x}_i^T - \boldsymbol{x}_j\boldsymbol{x}_j^T)\\ \boldsymbol{x}_i - \boldsymbol{x}_j\end{bmatrix}\begin{bmatrix}\mathrm{vec}(\boldsymbol{x}_i\boldsymbol{x}_i^T - \boldsymbol{x}_j\boldsymbol{x}_j^T)\\ \boldsymbol{x}_i - \boldsymbol{x}_j\end{bmatrix}^T\right\|\right] \quad \text{by convexity of } \|\cdot\| \text{ and Jensen's inequality}$$

$$\leq \mathbb{E}\left[\sum_{\mathcal{S}}\left\|\begin{bmatrix}\mathrm{vec}(\boldsymbol{x}_i\boldsymbol{x}_i^T - \boldsymbol{x}_j\boldsymbol{x}_j^T)\\ \boldsymbol{x}_i - \boldsymbol{x}_j\end{bmatrix}\begin{bmatrix}\mathrm{vec}(\boldsymbol{x}_i\boldsymbol{x}_i^T - \boldsymbol{x}_j\boldsymbol{x}_j^T)\\ \boldsymbol{x}_i - \boldsymbol{x}_j\end{bmatrix}^T\right\|\right] \quad \text{By the triangle inequality}$$

$$\leq \mathbb{E}\left[\sum_{\mathcal{S}}\left\|\begin{bmatrix}\mathrm{vec}(\boldsymbol{x}_i\boldsymbol{x}_i^T - \boldsymbol{x}_j\boldsymbol{x}_j^T)\\ \boldsymbol{x}_i - \boldsymbol{x}_j\end{bmatrix}\right\|_2^2\right]$$

$$\leq C_B^2|\mathcal{S}|,$$

and so

$$\max\left\{\left\|\mathbb{E}\left[\sum_{\mathcal{S}}\left(\varepsilon_p\begin{bmatrix}\mathrm{vec}(\boldsymbol{x}_i\boldsymbol{x}_i^T - \boldsymbol{x}_j\boldsymbol{x}_j^T)\\ \boldsymbol{x}_i - \boldsymbol{x}_j\end{bmatrix}\right)^T\left(\varepsilon_p\begin{bmatrix}\mathrm{vec}(\boldsymbol{x}_i\boldsymbol{x}_i^T - \boldsymbol{x}_j\boldsymbol{x}_j^T)\\ \boldsymbol{x}_i - \boldsymbol{x}_j\end{bmatrix}\right)\right]\right\|,$$

$$\left\| \mathbb{E}\left[ \sum_{\mathcal{S}} \left( \varepsilon_p \begin{bmatrix} \mathrm{vec}(\boldsymbol{x}_i\boldsymbol{x}_i^T - \boldsymbol{x}_j\boldsymbol{x}_j^T) \\ \boldsymbol{x}_i - \boldsymbol{x}_j \end{bmatrix} \right)\left( \varepsilon_p \begin{bmatrix} \mathrm{vec}(\boldsymbol{x}_i\boldsymbol{x}_i^T - \boldsymbol{x}_j\boldsymbol{x}_j^T) \\ \boldsymbol{x}_i - \boldsymbol{x}_j \end{bmatrix} \right)^T \right]\right\| \right\} \le C_B^2|\mathcal{S}|.$$

Therefore, by Theorem 6.1.1 of [49]

$$\mathbb{E}\left[\left\| \sum_{\mathcal{S}} \varepsilon_p \begin{bmatrix} \mathrm{vec}(\boldsymbol{x}_i\boldsymbol{x}_i^T - \boldsymbol{x}_j\boldsymbol{x}_j^T) \\ \boldsymbol{x}_i - \boldsymbol{x}_j \end{bmatrix}\right\|_2 \right] \le \sqrt{2C_B^2|\mathcal{S}|\log(d^2 + d + 1)} + \frac{C_B}{3}\log(d^2 + d + 1).$$

Plugging into eq. (41) and then eq. (39), we have with probability greater than $1 - \delta$,

$$R(\widehat{\boldsymbol{M}}, \{\widehat{\boldsymbol{v}}_k\}_{k=1}^K) - R(\boldsymbol{M}^*, \{\boldsymbol{v}_k^*\}_{k=1}^K)$$
$$\le 4L\sqrt{\lambda_F^2 + K\lambda_v^2}\left( \sqrt{\frac{2C_B^2}{|\mathcal{S}|}\log(d^2 + d + 1)} + \frac{C_B}{3|\mathcal{S}|}\log(d^2 + d + 1) \right) + \sqrt{\frac{8L^2\gamma^2\log(2/\delta)}{|\mathcal{S}|}}.$$

Taking $B = 1$, we have the desired result.

### D.2 Proof of Theorem 3.3

The proof is identical to that of Theorem 3.1 up until eq. (40), where instead of applying Cauchy-Schwarz we apply the matrix Hölder's inequality:

$$\frac{2L}{|\mathcal{S}|}\mathbb{E}\left[ \sup_{\boldsymbol{M},\{\boldsymbol{v}_k\}_{k=1}^K} \left| \left\langle \sum_{\mathcal{S}} \varepsilon_p \begin{bmatrix} \boldsymbol{x}_i\boldsymbol{x}_i^T - \boldsymbol{x}_j\boldsymbol{x}_j^T & \boldsymbol{0} & \cdots & \underbrace{\boldsymbol{x}_i - \boldsymbol{x}_j}_{\text{column } d + k} & \cdots & \boldsymbol{0} \end{bmatrix}, \begin{bmatrix} \boldsymbol{M} & \boldsymbol{v}_1 & \cdots & \boldsymbol{v}_K \end{bmatrix} \right\rangle \right| \right]$$

$$\le \frac{2L}{|\mathcal{S}|}\mathbb{E}\left[ \sup_{\boldsymbol{M},\{\boldsymbol{v}_k\}_{k=1}^K} \left\| \sum_{\mathcal{S}} \varepsilon_p \begin{bmatrix} \boldsymbol{x}_i\boldsymbol{x}_i^T - \boldsymbol{x}_j\boldsymbol{x}_j^T & \boldsymbol{0} & \cdots & \underbrace{\boldsymbol{x}_i - \boldsymbol{x}_j}_{\text{column } d + k} & \cdots & \boldsymbol{0} \end{bmatrix} \right\| \left\| \begin{bmatrix} \boldsymbol{M} & \boldsymbol{v}_1 & \cdots & \boldsymbol{v}_K \end{bmatrix} \right\|_* \right]$$

$$= \frac{2L}{|\mathcal{S}|}\left( \sup_{\boldsymbol{M},\{\boldsymbol{v}_k\}_{k=1}^K} \left\| \begin{bmatrix} \boldsymbol{M} & \boldsymbol{v}_1 & \cdots & \boldsymbol{v}_K \end{bmatrix} \right\|_* \right)\mathbb{E}\left[\left\| \sum_{\mathcal{S}} \varepsilon_p \begin{bmatrix} \boldsymbol{x}_i\boldsymbol{x}_i^T - \boldsymbol{x}_j\boldsymbol{x}_j^T & \boldsymbol{0} & \cdots & \underbrace{\boldsymbol{x}_i - \boldsymbol{x}_j}_{\text{column } d + k} & \cdots & \boldsymbol{0} \end{bmatrix} \right\| \right]$$

$$\le \frac{2L\lambda_*}{|\mathcal{S}|}\mathbb{E}\left[\left\| \sum_{p\in\mathcal{S}} \varepsilon_p \begin{bmatrix} \boldsymbol{x}_i\boldsymbol{x}_i^T - \boldsymbol{x}_j\boldsymbol{x}_j^T & \boldsymbol{0} & \cdots & \underbrace{\boldsymbol{x}_i - \boldsymbol{x}_j}_{\text{column } d + k} & \cdots & \boldsymbol{0} \end{bmatrix} \right\| \right].$$

In a similar manner to Appendix D.1, we can apply Matrix Bernstein to bound $\mathbb{E}\left[\left\| \sum_{\mathcal{S}} \varepsilon_p \boldsymbol{Z}_{ij}^{(k)} \right\| \right]$, where for conciseness we have defined

$$\boldsymbol{Z}_{ij}^{(k)} := \begin{bmatrix} \boldsymbol{x}_i\boldsymbol{x}_i^T - \boldsymbol{x}_j\boldsymbol{x}_j^T & \boldsymbol{0} & \cdots & \underbrace{\boldsymbol{x}_i - \boldsymbol{x}_j}_{\text{column } d + k} & \cdots & \boldsymbol{0} \end{bmatrix}.$$

First note that

$$\|\varepsilon_p \boldsymbol{Z}_{ij}^{(k)}\| \le \|\boldsymbol{x}_i\boldsymbol{x}_i^T - \boldsymbol{x}_j\boldsymbol{x}_j^T\| + \|\boldsymbol{x}_i - \boldsymbol{x}_j\|$$
$$\le \|\boldsymbol{x}_i\boldsymbol{x}_i^T\| + \|\boldsymbol{x}_j\boldsymbol{x}_j^T\| + \|\boldsymbol{x}_i\| + \|\boldsymbol{x}_j\|$$
$$= \|\boldsymbol{x}_i\|_2^2 + \|\boldsymbol{x}_j\|_2^2 + \|\boldsymbol{x}_i\|_2 + \|\boldsymbol{x}_j\|_2$$
$$\le 2(B^2 + B),$$

where we have used the fact that the operator norm of a vector is simply the $\ell_2$ norm, along with the assumption that $\|\boldsymbol{x}_i\|_2 \le B \; \forall i$ for a constant $B > 0$ (taken to be 1 in the theorem statement). We next bound the matrix variance of the sum $\sum_{\mathcal{S}} \varepsilon_p \boldsymbol{Z}_{ij}^{(k)}$, defined as $v := \max\{\|\sum_{\mathcal{S}} \mathbb{E}[\varepsilon_p \boldsymbol{Z}_{ij}^{(k)}(\varepsilon_p \boldsymbol{Z}_{ij}^{(k)})^T]\|, \|\sum_{\mathcal{S}} \mathbb{E}[(\varepsilon_p \boldsymbol{Z}_{ij}^{(k)})^T \varepsilon_p \boldsymbol{Z}_{ij}^{(k)}]\|\}$, which is equal to $|\mathcal{S}|\max\{\|\mathbb{E}[\boldsymbol{Z}_{ij}^{(k)}(\boldsymbol{Z}_{ij}^{(k)})^T]\|, \|\mathbb{E}[(\boldsymbol{Z}_{ij}^{(k)})^T \boldsymbol{Z}_{ij}^{(k)}]\|\}$ since $\varepsilon_p \in \{-1, 1\}$ and each data point in $\mathcal{S}$ is i.i.d.

Towards bounding $v$, we have the following technical lemma, proved later in this section:

**Lemma D.1.** *For $k \sim \mathrm{Unif}([K])$ and $1 \le i < j \le n$ such that $(i,j)$ is chosen uniformly at random from the set of $\binom{n}{2}$ unique pairs,*

$$\mathbb{E}\left[\boldsymbol{Z}_{ij}^{(k)}(\boldsymbol{Z}_{ij}^{(k)})^T\right] = \frac{2}{n(n-1)}\left[\boldsymbol{X}(n\boldsymbol{D}-\boldsymbol{G})\boldsymbol{X}^T + n\boldsymbol{X}\boldsymbol{X}^T - n^2\overline{\boldsymbol{x}}\,\overline{\boldsymbol{x}}^T\right]$$

*where $\boldsymbol{G} := \boldsymbol{X}^T\boldsymbol{X}$, $\boldsymbol{D} := \mathrm{diag}([\|\boldsymbol{x}_1\|^2, \ldots, \|\boldsymbol{x}_n\|^2])$, and $\overline{\boldsymbol{x}} := \frac{1}{n}\sum_{i=1}^n \boldsymbol{x}_i$. Furthermore,*

$$\mathbb{E}\left[(\boldsymbol{Z}_{ij}^{(k)})^T\boldsymbol{Z}_{ij}^{(k)}\right] = \frac{2}{n(n-1)}\begin{bmatrix} \boldsymbol{X}(n\boldsymbol{D}-\boldsymbol{G})\boldsymbol{X}^T & \frac{n}{K}\cdot\sum_{\ell=1}^n(\boldsymbol{x}_\ell-\overline{\boldsymbol{x}})^T\boldsymbol{x}_\ell\cdot\boldsymbol{x}_\ell\boldsymbol{1}_K^T \\ \frac{n}{K}\cdot\boldsymbol{1}_K\sum_{\ell=1}^n(\boldsymbol{x}_\ell-\overline{\boldsymbol{x}})^T\boldsymbol{x}_\ell\cdot\boldsymbol{x}_\ell^T & \frac{n}{K}\cdot\left(\|\boldsymbol{X}\|_F^2 - n\|\overline{\boldsymbol{x}}\|^2\right)\boldsymbol{I}_K \end{bmatrix},$$

*and*

$$\max\{\|\mathbb{E}[\boldsymbol{Z}_{ij}^{(k)}(\boldsymbol{Z}_{ij}^{(k)})^T]\|, \|\mathbb{E}[(\boldsymbol{Z}_{ij}^{(k)})^T\boldsymbol{Z}_{ij}^{(k)}]\|\} \le \left(4(B^2+1) + \frac{4\min(d,n)}{K}\right)\frac{\|\boldsymbol{X}\|^2}{n} + \frac{16B^3}{\sqrt{K}}.$$

Therefore, we have

$$v \le |\mathcal{S}|\left[\left(4(B^2+1) + \frac{4\min(d,n)}{K}\right)\frac{\|\boldsymbol{X}\|^2}{n} + \frac{16B^3}{\sqrt{K}}\right].$$

Noting that $\boldsymbol{Z}_{ij}^{(k)}$ is $d \times d + K$, from Theorem 6.1.1 in [49],

$$\mathbb{E}\left[\left\|\sum_{\mathcal{S}}\varepsilon_p \begin{bmatrix} \boldsymbol{x}_i\boldsymbol{x}_i^T - \boldsymbol{x}_j\boldsymbol{x}_j^T & \boldsymbol{0} & \cdots & \underbrace{\boldsymbol{x}_i - \boldsymbol{x}_j}_{\text{column } d+k} & \cdots & \boldsymbol{0} \end{bmatrix}\right\|\right]$$

$$= \mathbb{E}\left[\left\|\sum_{\mathcal{S}}\varepsilon_p\boldsymbol{Z}_{ij}^{(k)}\right\|\right]$$

$$\le \sqrt{2v\log(2d+K)} + \frac{2(B^2+B)}{3}\log(2d+K)$$

$$\le \sqrt{2|\mathcal{S}|\log(2d+K)\left[\left(4(B^2+1) + \frac{4\min(d,n)}{K}\right)\frac{\|\boldsymbol{X}\|^2}{n} + \frac{16B^3}{\sqrt{K}}\right]} + \frac{2(B^2+B)}{3}\log(2d+K),$$

and so, continuing where we left off at the proof of Theorem 3.1,

$$\mathbb{E}\left[\sup_{\boldsymbol{M},\{\boldsymbol{v}_k\}_{k=1}^K}|\widehat{R}(\boldsymbol{M},\{\boldsymbol{v}_k\}_{k=1}^K) - R(\boldsymbol{M},\{\boldsymbol{v}_k\}_{k=1}^K)|\right]$$

$$\le 2L\sqrt{\frac{2\lambda_*^2\log(2d+K)}{|\mathcal{S}|}\left[\left(4(B^2+1) + \frac{4\min(d,n)}{K}\right)\frac{\|\boldsymbol{X}\|^2}{n} + \frac{16B^3}{\sqrt{K}}\right]} + \frac{4L(B^2+B)\lambda_*}{3|\mathcal{S}|}\log(2d+K).$$

Combining this with the first part of the proof of Theorem 3.1 (which, as we mentioned is identical here), with probability at least $1 - \delta$

$$R(\widehat{\boldsymbol{M}},\{\widehat{\boldsymbol{v}}_k\}_{k=1}^K) - R(\boldsymbol{M}^*,\{\boldsymbol{v}_k^*\}_{k=1}^K)$$

$$\le 2L\sqrt{\frac{2\lambda_*^2\log(2d+K)}{|\mathcal{S}|}\left[\left(4(B^2+1) + \frac{4\min(d,n)}{K}\right)\frac{\|\boldsymbol{X}\|^2}{n} + \frac{16B^3}{\sqrt{K}}\right]} + \tag{42}$$

$$\frac{4L(B^2+B)\lambda_*}{3|\mathcal{S}|}\log(2d+K) + \sqrt{\frac{8L^2\gamma^2\log(2/\delta)}{|\mathcal{S}|}}.$$

Taking $B = 1$, we have the desired result.

**Proof of Lemma D.1:** Break $\boldsymbol{Z}_{i,j}^{(k)}$ into submatrices

$$\boldsymbol{A}_{ij} := \boldsymbol{x}_i\boldsymbol{x}_i^T - \boldsymbol{x}_j\boldsymbol{x}_j^T$$

and

$$\boldsymbol{B}_{ij}^{(k)} := \begin{bmatrix} \boldsymbol{0} & \cdots & \underbrace{\boldsymbol{x}_i - \boldsymbol{x}_j}_{\text{column } k} & \cdots & \boldsymbol{0} \end{bmatrix}.$$

We proceed by computing the expected products of all submatrix combinations. Throughout, we will be summing over all $\binom{n}{2}$ item pairs. To reduce constant factors to track, we will use the fact that $\sum_{i=1}^{n-1}\sum_{j=i+1}^{n} \boldsymbol{Q}_{ij} = \frac{1}{2}\sum_{i=1}^{n}\sum_{j\neq i} \boldsymbol{Q}_{ij} = \frac{1}{2}\sum_{i\neq j} \boldsymbol{Q}_{ij}$ for matrices $\boldsymbol{Q}_{ij}$ satisfying $\boldsymbol{Q}_{ij} = \boldsymbol{Q}_{ji}$.

**Step 1:** Computing $\mathbb{E}\left[\boldsymbol{A}_{ij}^T\boldsymbol{A}_{ij}\right] = \mathbb{E}\left[\boldsymbol{A}_{ij}\boldsymbol{A}_{ij}^T\right]$

Note that the above equality holds by symmetry of $\boldsymbol{A}_{ij}$. Define $\boldsymbol{E}_{ij} := \boldsymbol{e}_i\boldsymbol{e}_i^T - \boldsymbol{e}_j\boldsymbol{e}_j$, and note that $\boldsymbol{A}_{ij} = \boldsymbol{X}\boldsymbol{E}_{ij}\boldsymbol{X}^T$. Therefore

$$
\begin{aligned}
\sum_{i=1}^{n-1}\sum_{j=i+1}^{n} \boldsymbol{A}_{ij}^T\boldsymbol{A}_{ij} &= \frac{1}{2}\sum_{i\neq j}\boldsymbol{A}_{ij}^T\boldsymbol{A}_{ij} \\
&= \frac{1}{2}\sum_{i\neq j}\boldsymbol{X}\boldsymbol{E}_{ij}\boldsymbol{X}^T\boldsymbol{X}\boldsymbol{E}_{ij}\boldsymbol{X}^T \\
&= \frac{1}{2}\boldsymbol{X}\left(\sum_{i\neq j}\boldsymbol{E}_{ij}\boldsymbol{X}^T\boldsymbol{X}\boldsymbol{E}_{ij}\right)\boldsymbol{X}^T \\
&= \frac{1}{2}\boldsymbol{X}\left(\sum_{i\neq j}\boldsymbol{E}_{ij}\boldsymbol{G}\boldsymbol{E}_{ij}\right)\boldsymbol{X}^T \\
&= \frac{1}{2}\boldsymbol{X}\left(\sum_{i\neq j}\|\boldsymbol{x}_i\|^2\boldsymbol{e}_i\boldsymbol{e}_i^T + \|\boldsymbol{x}_j\|^2\boldsymbol{e}_j\boldsymbol{e}_j^T - \boldsymbol{x}_i^T\boldsymbol{x}_j(\boldsymbol{e}_i\boldsymbol{e}_j^T + \boldsymbol{e}_j\boldsymbol{e}_i^T)\right)\boldsymbol{X}^T \\
&= \frac{1}{2}\boldsymbol{X}\left(2(n-1)\boldsymbol{D} - \sum_{i\neq j}\boldsymbol{x}_i^T\boldsymbol{x}_j(\boldsymbol{e}_i\boldsymbol{e}_j^T + \boldsymbol{e}_j\boldsymbol{e}_i^T)\right)\boldsymbol{X}^T \\
&= \boldsymbol{X}\left(n\boldsymbol{D} - \boldsymbol{G}\right)\boldsymbol{X}^T.
\end{aligned}
$$

As $\boldsymbol{A}_{ij}$ does not depend on the random variable $k$,

$$
\mathbb{E}\left[\boldsymbol{A}_{ij}^T\boldsymbol{A}_{ij}\right] = \frac{1}{\binom{n}{2}}\sum_{i=1}^{n-1}\sum_{j=i+1}^{n}\boldsymbol{A}_{ij}^T\boldsymbol{A}_{ij} = \frac{2}{n(n-1)}\boldsymbol{X}\left(n\boldsymbol{D}-\boldsymbol{G}\right)\boldsymbol{X}^T.
$$

**Step 2:** Computing $\mathbb{E}[\boldsymbol{A}_{ij}^T\boldsymbol{B}_{ij}^{(k)}]$

$$
\boldsymbol{A}_{ij}^T\boldsymbol{B}_{ij}^{(k)} = \left(\boldsymbol{x}_i\boldsymbol{x}_i^T - \boldsymbol{x}_j\boldsymbol{x}_j^T\right)\begin{bmatrix}\boldsymbol{0} & \cdots & \underbrace{\boldsymbol{x}_i - \boldsymbol{x}_j}_{\text{column } k} & \cdots & \boldsymbol{0}\end{bmatrix}.
$$

Hence, the product is $\boldsymbol{0}$ for all columns except the $k^{\text{th}}$. As we sum over all $k \in [K]$ to compute the expectation, the resulting submatrix is rank 1 with $K$ copies of this same column. We therefore, compute the expectation of this column first.

$$
\begin{aligned}
\sum_{i=1}^{n-1}\sum_{j=i+1}^{n}\left(\boldsymbol{x}_i\boldsymbol{x}_i^T - \boldsymbol{x}_j\boldsymbol{x}_j^T\right)\left(\boldsymbol{x}_i - \boldsymbol{x}_j\right) &= \sum_{i=1}^{n-1}\sum_{j=i+1}^{n}\|\boldsymbol{x}_i\|^2\boldsymbol{x}_i + \|\boldsymbol{x}_j\|^2\boldsymbol{x}_j - \boldsymbol{x}_i^T\boldsymbol{x}_j\boldsymbol{x}_i - \boldsymbol{x}_i^T\boldsymbol{x}_j\boldsymbol{x}_j \\
&= \frac{1}{2}\sum_{i}\sum_{j\neq i}\|\boldsymbol{x}_i\|^2\boldsymbol{x}_i + \|\boldsymbol{x}_j\|^2\boldsymbol{x}_j - \boldsymbol{x}_i^T\boldsymbol{x}_j\boldsymbol{x}_i - \boldsymbol{x}_i^T\boldsymbol{x}_j\boldsymbol{x}_j \\
&= (n-1)\sum_{i}\|\boldsymbol{x}_i\|^2\boldsymbol{x}_i - \sum_{i}\sum_{j\neq i}\boldsymbol{x}_i^T\boldsymbol{x}_j\boldsymbol{x}_i \\
&= n\sum_{i}\|\boldsymbol{x}_i\|^2\boldsymbol{x}_i - \sum_{i}\sum_{j}\boldsymbol{x}_i^T\boldsymbol{x}_j\boldsymbol{x}_i \\
&= n\sum_{i}\|\boldsymbol{x}_i\|^2\boldsymbol{x}_i - \sum_{i}\sum_{j}\boldsymbol{x}_i\boldsymbol{x}_i^T\boldsymbol{x}_j
\end{aligned}
$$

$$= n \sum_i \|\boldsymbol{x}_i\|^2 \boldsymbol{x}_i - \sum_i \boldsymbol{x}_i \boldsymbol{x}_i^T \sum_j \boldsymbol{x}_j$$

$$= n \sum_i \|\boldsymbol{x}_i\|^2 \boldsymbol{x}_i - n \sum_i \boldsymbol{x}_i^T \overline{\boldsymbol{x}} \boldsymbol{x}_i$$

$$= n \sum_i \left( \|\boldsymbol{x}_i\|^2 - \boldsymbol{x}_i^T \overline{\boldsymbol{x}} \right) \boldsymbol{x}_i$$

$$= n \sum_i (\boldsymbol{x}_i - \overline{\boldsymbol{x}})^T \boldsymbol{x}_i \cdot \boldsymbol{x}_i.$$

We therefore have

$$\sum_{i=1}^{n-1} \sum_{j=i+1}^{n} \boldsymbol{A}_{ij}^T \boldsymbol{B}_{ij}^{(k)} = \begin{bmatrix} \boldsymbol{0} & \cdots & \underbrace{n \sum_i (\boldsymbol{x}_i - \overline{\boldsymbol{x}})^T \boldsymbol{x}_i \cdot \boldsymbol{x}_i}_{\text{column } k} & \cdots & \boldsymbol{0} \end{bmatrix}.$$

When we then sum over $k$ and normalize by $1/K$, and divide by the $\binom{n}{2}$ unique pairs to finish the expectation computation, we have

$$\mathbb{E}[\boldsymbol{A}_{ij}^T \boldsymbol{B}_{ij}^{(k)}] = \frac{n}{K\binom{n}{2}} \sum_i (\boldsymbol{x}_i - \overline{\boldsymbol{x}})^T \boldsymbol{x}_i \cdot \boldsymbol{x}_i \boldsymbol{1}_K^T = \frac{2n}{K \cdot n \cdot (n-1)} \cdot \sum_{\ell=1}^{n} (\boldsymbol{x}_\ell - \overline{\boldsymbol{x}})^T \boldsymbol{x}_\ell \cdot \boldsymbol{x}_\ell \boldsymbol{1}_K^T,$$

where the factor of $\boldsymbol{1}_K$ simply generates a matrix with $K$ copies of this same column.

**Step 3:** Computing $\mathbb{E}[(\boldsymbol{B}_{ij}^{(k)})^T \boldsymbol{B}_{ij}^{(k)}]$

$$(\boldsymbol{B}_{ij}^{(k)})^T \boldsymbol{B}_{ij}^{(k)} = \begin{bmatrix} \boldsymbol{0}^T \\ \vdots \\ (\boldsymbol{x}_i - \boldsymbol{x}_j)^T \\ \vdots \\ \boldsymbol{0}^T \end{bmatrix} \begin{bmatrix} \boldsymbol{0} & \cdots & \boldsymbol{x}_i - \boldsymbol{x}_j & \cdots & \boldsymbol{0} \end{bmatrix}$$

Hence,

$$[(\boldsymbol{B}_{ij}^{(k)})^T \boldsymbol{B}_{ij}^{(k)}]_{p,q} = \begin{cases} \|\boldsymbol{x}_i - \boldsymbol{x}_j\|^2 & \text{if } p = q = k \\ 0 & \text{otherwise} \end{cases}.$$

Since the non-zero entry in this matrix does not depend on $k$, $\mathbb{E}[(\boldsymbol{B}_{ij}^{(k)})^T \boldsymbol{B}_{ij}^{(k)}]$ is a equal to a constant times the $K$-dimensional identity. To compute this constant, first we evaluate

$$\sum_{i=1}^{n-1} \sum_{j=i+1}^{n} \|\boldsymbol{x}_i - \boldsymbol{x}_j\|^2 = \sum_{i=1}^{n-1} \sum_{j=i+1}^{n} \boldsymbol{G}_{ii} + \boldsymbol{G}_{jj} - 2\boldsymbol{G}_{ij}$$

$$= \frac{1}{2} \sum_i \sum_{j \neq i} \boldsymbol{G}_{ii} + \boldsymbol{G}_{jj} - 2\boldsymbol{G}_{ij}$$

$$= n \cdot \text{Tr}(\boldsymbol{G}) - \sum_i \sum_j \boldsymbol{G}_{ij}$$

$$= n \cdot \text{Tr}(\boldsymbol{G}) - \sum_i \sum_j \boldsymbol{x}_i^T \boldsymbol{x}_j$$

$$= n \cdot \text{Tr}(\boldsymbol{G}) - \left( \sum_i \boldsymbol{x}_i \right)^T \sum_j \boldsymbol{x}_j$$

$$= n \|\boldsymbol{X}\|_F^2 - n^2 \|\overline{\boldsymbol{x}}\|^2$$

The first equality holds since for a Gram matrix $\boldsymbol{G} = \boldsymbol{X}^T \boldsymbol{X}$, we have that $\|\boldsymbol{x}_i - \boldsymbol{x}_j\|^2 = \boldsymbol{G}_{ii} - 2\boldsymbol{G}_{ij} + \boldsymbol{G}_{jj}$. In the final equality, we have used the fact that $\text{Tr}(\boldsymbol{G}) = \|\boldsymbol{X}\|_F^2$.

By summing $(B_{ij}^{(k)})^T B_{ij}^{(k)}$ over all users and unique pairs and then dividing by $K$ and then $\binom{n}{2}$ to compute the expectation, we have:

$$\mathbb{E}[(B_{ij}^{(k)})^T B_{ij}^{(k)}] = \frac{1}{K\binom{n}{2}} \sum_{k=1}^{K} \sum_{i \neq j} (B_{ij}^{(k)})^T B_{ij}^{(k)}$$

$$= \frac{2}{K \cdot n \cdot (n-1)} \cdot \sum_{i=1}^{n-1} \sum_{j=i+1}^{n} \|x_i - x_j\|^2 I_K$$

$$= \frac{2n}{K \cdot n \cdot (n-1)} \cdot \left( \|X\|_F^2 - n\|\bar{x}\|^2 \right) I_K.$$

Steps 1-3 establish the second claim regarding $\mathbb{E}\left[ (Z_{ij}^{(k)})^T Z_{ij}^{(k)} \right]$ by noting that

$$\mathbb{E}\left[ (Z_{ij}^{(k)})^T Z_{ij}^{(k)} \right] = \begin{bmatrix} \mathbb{E}\left[ A_{ij}^T A_{ij} \right] & \mathbb{E}[A_{ij}^T B_{ij}^{(k)}] \\ \mathbb{E}[(B_{ij}^{(k)})^T A_{ij}] & \mathbb{E}[(B_{ij}^{(k)})^T B_{ij}^{(k)}] \end{bmatrix},$$

where we have used both the result of Step 2 and its transpose.

**Step 4:** Computing $\mathbb{E}[B_{ij}^{(k)} (B_{ij}^{(k)})^T]$

$$B_{ij}^{(k)} (B_{ij}^{(k)})^T = [0 \quad \cdots \quad x_i - x_j \quad \cdots \quad 0] \begin{bmatrix} 0^T \\ \vdots \\ (x_i - x_j)^T \\ \vdots \\ 0^T \end{bmatrix} = (x_i - x_j)(x_i - x_j)^T.$$

We compute

$$\sum_{i=1}^{n-1} \sum_{j=i+1}^{n} (x_i - x_j)(x_i - x_j)^T = \frac{1}{2} \sum_{i} \sum_{j \neq i} (x_i - x_j)(x_i - x_j)^T$$

$$= (n-1) \sum_{i} x_i x_i^T - \sum_{i} \sum_{j \neq i} x_i x_j^T$$

$$= n \sum_{i} x_i x_i^T - \sum_{i} \sum_{j} x_i x_j^T$$

$$= nXX^T - \sum_{i} x_i \sum_{j} x_j^T$$

$$= nXX^T - n^2 \bar{x}\bar{x}^T.$$

Note that this expression does not depend on $K$, and so

$$\mathbb{E}[B_{ij}^{(k)} (B_{ij}^{(k)})^T] = \frac{1}{\binom{n}{2}} \sum_{i=1}^{n-1} \sum_{j=i+1}^{n} (x_i - x_j)(x_i - x_j)^T = \frac{2}{n \cdot (n-1)} \left( nXX^T - n^2 \bar{x}\bar{x}^T \right).$$

Steps 1 and 4 establish the claim regarding $\mathbb{E}\left[ Z_{ij}^{(k)} (Z_{ij}^{(k)})^T \right]$ by noting that

$$\mathbb{E}\left[ Z_{ij}^{(k)} (Z_{ij}^{(k)})^T \right] = \mathbb{E}\left[ A_{ij} A_{ij}^T \right] + \mathbb{E}[B_{ij}^{(k)} (B_{ij}^{(k)})^T].$$

To bound $\|\mathbb{E}[Z_{ij}^{(k)} (Z_{ij}^{(k)})^T]\|$, note that

$$\mathbb{E}\left[ Z_{ij}^{(k)} (Z_{ij}^{(k)})^T \right] = \frac{2}{n(n-1)} X \left[ nD - G + nI - 11^T \right] X^T.$$

We can expand the center term as

$$nD - G + nI - 11^T = \begin{cases} (n-1)(\|x_i\|^2 + 1) & i = j \\ -x_i^T x_j - 1 & i \neq j \end{cases}.$$

From the Gershgorin Circle Theorem we then have

$$\|nD - G + nI - 11^T\| \leq \max_i [(n-1)(\|x_i\|^2 + 1) + \sum_{j \neq i} |\langle x_i, x_j \rangle + 1|]$$

$$\leq \max_i [(n-1)(\|x_i\|^2 + 1) + n - 1 + \|x_i\| \sum_{j \neq i} \|x_j\|]$$

$$\leq (n-1)(B^2 + 1) + n - 1 + (n-1)B^2$$

$$= 2(n-1)(B^2 + 1).$$

We then have

$$\left\| \mathbb{E} \left[ Z_{ij}^{(k)} (Z_{ij}^{(k)})^T \right] \right\| \leq \frac{2}{n(n-1)} \|X\|^2 \|nD - G + nI - 11^T\|$$

$$\leq \frac{4(B^2 + 1)}{n} \|X\|^2.$$

We take a slightly different approach in bounding $\left\| \mathbb{E} \left[ (Z_{ij}^{(k)})^T Z_{ij}^{(k)} \right] \right\|$. First, we decompose $\mathbb{E} \left[ (Z_{ij}^{(k)})^T Z_{ij}^{(k)} \right]$ as

$$\mathbb{E} \left[ (Z_{ij}^{(k)})^T Z_{ij}^{(k)} \right] = \frac{2}{n(n-1)} \begin{bmatrix} A & B \\ B^T & C \end{bmatrix},$$

where $A = X(nD - G)X^T$, $B = \frac{n}{K} \cdot \sum_{\ell=1}^n (x_\ell - \overline{x})^T x_\ell \cdot x_\ell 1_K^T$, and $C = \frac{n}{K} \cdot \left( \|X\|_F^2 - n\|\overline{x}\|^2 \right) I_K$. By repeated applications of the triangle inequality,

$$\left\| \begin{bmatrix} A & B \\ B^T & C \end{bmatrix} \right\| \leq \left\| \begin{bmatrix} A & 0 \\ 0 & 0 \end{bmatrix} \right\| + \left\| \begin{bmatrix} 0 & B \\ 0 & 0 \end{bmatrix} \right\| + \left\| \begin{bmatrix} 0 & 0 \\ B^T & 0 \end{bmatrix} \right\| + \left\| \begin{bmatrix} 0 & 0 \\ 0 & C \end{bmatrix} \right\|$$

$$= \|A\| + 2\|B\| + \|C\|.$$

We bound each of these terms in turn. Noting that $A = X(nD - G)X^T$ and that

$$nD - G = \begin{cases} (n-1)\|x_i\|^2 & i = j \\ -x_i^T x_j & i \neq j \end{cases},$$

and so by applying the Gershgorin Circle Theorem as above we have

$$\|nD - G\| \leq \max_i [(n-1)\|x_i\|^2 + \sum_{j \neq i} |\langle x_i, x_j \rangle|]$$

$$\leq \max_i [(n-1)\|x_i\|^2 + \|x_i\| \sum_{j \neq i} \|x_j\|]$$

$$\leq (n-1)B^2 + (n-1)B^2$$

$$= 2(n-1)B^2,$$

and so

$$\|A\| = \|X(nD - G)X^T\| \leq \|X\|^2 \|nD - G\| \leq 2(n-1)B^2 \|X\|^2.$$

Next, we bound $\|B\|$. First note that for any matrix of the form $z1_K^T$ where $z \in \mathbb{R}^K$,

$$z1_K^T = \frac{z}{\|z\|_2} (\|z\|_2 \sqrt{K}) \frac{1_K^T}{\sqrt{K}}$$

and so $\|z1_K^T\| = \|z\|_2 \sqrt{K}$. Applying this result to $B$,

$$\|B\| = \left\| \frac{n}{K} \cdot \sum_{\ell=1}^n (x_\ell - \overline{x})^T x_\ell \cdot x_\ell 1_K^T \right\|$$

$$= \frac{n}{\sqrt{K}} \left\| \sum_{\ell=1}^{n} (\boldsymbol{x}_\ell - \overline{\boldsymbol{x}})^T \boldsymbol{x}_\ell \cdot \boldsymbol{x}_\ell \right\|_2$$

$$\leq \frac{n}{\sqrt{K}} \sum_{\ell=1}^{n} |(\boldsymbol{x}_\ell - \overline{\boldsymbol{x}})^T \boldsymbol{x}_\ell| \, \|\boldsymbol{x}_\ell\|_2$$

$$\leq \frac{2n^2 B^3}{\sqrt{K}}.$$

Finally,

$$\|\boldsymbol{D}\| = \left\| \frac{n}{K} \cdot \left( \|\boldsymbol{X}\|_F^2 - n\|\overline{\boldsymbol{x}}\|^2 \right) \boldsymbol{I}_K \right\|$$

$$= \frac{n}{K} \left| \|\boldsymbol{X}\|_F^2 - n\|\overline{\boldsymbol{x}}\|^2 \right|$$

$$= \frac{n}{K} \left| \|\boldsymbol{X}\|_F^2 - n\overline{\boldsymbol{x}}^T \overline{\boldsymbol{x}} \right|$$

$$= \frac{n}{K} \left| \|\boldsymbol{X}\|_F^2 - \frac{1}{n} \left( \sum_{i=1}^{n} \boldsymbol{x}_i \right)^T \left( \sum_{i=1}^{n} \boldsymbol{x}_i \right) \right|$$

$$= \frac{n}{K} \left| \|\boldsymbol{X}\|_F^2 - \frac{1}{n} \sum_{i=1}^{n} \sum_{j=1}^{n} \boldsymbol{x}_i^T \boldsymbol{x}_j \right|$$

$$= \frac{n}{K} \left| \|\boldsymbol{X}\|_F^2 - \frac{1}{n} \mathbf{1}_n^T \boldsymbol{G} \mathbf{1}_n \right|$$

$$= \frac{n}{K} \left| \mathrm{Tr}(\boldsymbol{G}) - \frac{1}{n} \mathbf{1}_n^T \boldsymbol{G} \mathbf{1}_n \right|$$

$$= \frac{n}{K} \left| \mathrm{Tr}(\boldsymbol{G}) - \frac{\mathbf{1}_n}{\sqrt{n}}^T \boldsymbol{G} \frac{\mathbf{1}_n}{\sqrt{n}} \right|$$

Let $\lambda_i$, $i \in [n]$ denote the eigenvalues of $\boldsymbol{G}$, sorted in decreasing order. Each $\lambda_i \geq 0$ since $\boldsymbol{G}$ is positive semidefinite by construction. We can then rewrite $\mathrm{Tr}(\boldsymbol{G}) = \sum_i \lambda_i$, and $\frac{\mathbf{1}_n}{\sqrt{n}}^T \boldsymbol{G} \frac{\mathbf{1}_n}{\sqrt{n}} \leq \max_{\|\boldsymbol{x}\|_2=1} \boldsymbol{x}^T \boldsymbol{G} \boldsymbol{x} = \lambda_1$, and so $\mathrm{Tr}(\boldsymbol{G}) - \frac{\mathbf{1}_n}{\sqrt{n}}^T \boldsymbol{G} \frac{\mathbf{1}_n}{\sqrt{n}} \geq \mathrm{Tr}(\boldsymbol{G}) - \lambda_1 = \sum_{i>1} \lambda_i \leq 0$, and so

$$\left| \mathrm{Tr}(\boldsymbol{G}) - \frac{\mathbf{1}_n}{\sqrt{n}}^T \boldsymbol{G} \frac{\mathbf{1}_n}{\sqrt{n}} \right| = \mathrm{Tr}(\boldsymbol{G}) - \frac{\mathbf{1}_n}{\sqrt{n}}^T \boldsymbol{G} \frac{\mathbf{1}_n}{\sqrt{n}} \leq \mathrm{Tr}(\boldsymbol{G}) = \|\boldsymbol{X}\|_F^2,$$

where the final inequality follows since $\boldsymbol{G}$ is positive semidefinite. Therefore, $\|\boldsymbol{D}\| \leq (n/K)\|\boldsymbol{X}\|_F^2$. Combining these bounds, we have

$$\left\| \mathbb{E}\left[ (\boldsymbol{Z}_{ij}^{(k)})^T \boldsymbol{Z}_{ij}^{(k)} \right] \right\| \leq \frac{2}{n(n-1)} (\|\boldsymbol{A}\| + 2\|\boldsymbol{B}\| + \|\boldsymbol{C}\|)$$

$$\leq \frac{2}{n(n-1)} \left( 2(n-1)B^2 \|\boldsymbol{X}\|^2 + \frac{4n^2 B^3}{\sqrt{K}} + \frac{n}{K} \|\boldsymbol{X}\|_F^2 \right)$$

$$= \frac{4B^2}{n} \|\boldsymbol{X}\|^2 + \frac{8B^3 n}{(n-1)\sqrt{K}} + \frac{2n}{K(n-1)} \frac{\|\boldsymbol{X}\|_F^2}{n}$$

$$\leq \frac{4B^2}{n} \|\boldsymbol{X}\|^2 + \frac{16B^3}{\sqrt{K}} + \frac{4}{K} \frac{\|\boldsymbol{X}\|_F^2}{n} \quad \text{since } n \geq 2 \implies n/n-1 \leq 2$$

$$\leq \left( 4B^2 + \frac{4\min(d,n)}{K} \right) \frac{\|\boldsymbol{X}\|^2}{n} + \frac{16B^3}{\sqrt{K}} \quad \text{since } \|\boldsymbol{X}\|_F^2 \leq \mathrm{rank}(\boldsymbol{X})\|\boldsymbol{X}\|^2 \leq \min(d,n)\|\boldsymbol{X}\|^2$$

Therefore,

$$\max\{ \| \mathbb{E}[\boldsymbol{Z}_{ij}^{(k)}(\boldsymbol{Z}_{ij}^{(k)})^T]\|, \|\mathbb{E}[(\boldsymbol{Z}_{ij}^{(k)})^T \boldsymbol{Z}_{ij}^{(k)}]\|\} \leq$$

$$\max\left\{\frac{4(B^2+1)}{n}\|\boldsymbol{X}\|^2, \left(4B^2+\frac{4\min(d,n)}{K}\right)\frac{\|\boldsymbol{X}\|^2}{n}+\frac{16B^3}{\sqrt{K}}\right\}$$

$$\leq\left(4(B^2+1)+\frac{4\min(d,n)}{K}\right)\frac{\|\boldsymbol{X}\|^2}{n}+\frac{16B^3}{\sqrt{K}},$$

completing the proof.

As an aside, we believe that this analysis can be tightened. Specifically, we believe that the final $O(1/\sqrt{K})$ term can be sharpened and that the maximum above should scale as $\|\boldsymbol{X}\|^2/n$, which would eliminate the requirement that $K = \Omega(d^2)$ in Corollary 3.3.1.

### D.3  Proof of Corollary 3.3.1

To prove the corollary, we will select values for the constants in Theorem 3.3 that hold with high probability, based on our assumed item, user, and metric distribution. We will derive our result from the extended version of Theorem 3.3 in eq. (42), which holds with probability at least $1 - \delta$ for a constant $B$ such that $\|\boldsymbol{x}_i\|_2 \leq B$ for all $i \in [n]$ and a specification of $0 < \delta < 1$.

To arrive at a setting for $B$, we can rewrite our item vectors as $\boldsymbol{x}_i = \frac{1}{\sqrt{d}}\boldsymbol{\eta}_i$, where $\boldsymbol{\eta}_i$ are i.i.d. $\mathcal{N}(\boldsymbol{0}, \boldsymbol{I})$, and so $\|\boldsymbol{x}_i\|_2^2 = \frac{1}{d}\|\boldsymbol{\eta}_i\|_2^2$. Since $\|\boldsymbol{\eta}_i\|_2^2$ is a chi-squared random variable with $d$ degrees of freedom, from [50] we have that for any $t > 0$, $\mathbb{P}(\|\boldsymbol{\eta}_i\|_2^2 \geq d + 2\sqrt{dt} + 2t) \leq e^{-t}$, and therefore by the union bound we have

$$\mathbb{P}(\max_{i\in[n]}\|\boldsymbol{\eta}_i\|_2^2 \geq d + 2\sqrt{dt} + 2t) \leq \sum_{i\in[n]}\mathbb{P}(\|\boldsymbol{\eta}_i\|_2^2 \geq d + 2\sqrt{dt} + 2t) \leq ne^{-t}.$$

Setting $t = \log\frac{n}{\delta_1}$ for any given $0 < \delta_1 < 1$, we have with probability greater than $1 - \delta_1$ that $\max_{i\in[n]}\|\boldsymbol{\eta}_i\|_2^2 < d + 2\sqrt{d\log\frac{n}{\delta_1}} + 2\log\frac{n}{\delta_1}$, which implies $\max_{i\in[n]}\|\boldsymbol{x}_i\|_2^2 < 1 + 2\sqrt{\frac{1}{d}\log\frac{n}{\delta_1}} + \frac{2}{d}\log\frac{n}{\delta_1}$. To get a more interpretable bound, we note for $n \geq 3$ that $\log\frac{n}{\delta} > 1$ and therefore with probability at least $1 - \delta_1$, $\max_{i\in[n]}\|\boldsymbol{x}_i\|_2^2 < 5\log\frac{n}{\delta_1}$. We can therefore set $B = \sqrt{5\log\frac{n}{\delta_1}}$.

Towards a setting for $\gamma$ as defined in eq. (7), let $\boldsymbol{z}_i := \sqrt{\frac{d}{\sqrt{r}}}\boldsymbol{L}^T\boldsymbol{x}_i$, in which case $\boldsymbol{x}_i^T\boldsymbol{M}\boldsymbol{x}_i = \|\boldsymbol{z}_i\|_2^2$. $\boldsymbol{z}_i$ is normally distributed with $\mathbb{E}[\boldsymbol{z}_i] = \boldsymbol{0}$ and $\text{Cov}(\boldsymbol{z}_i) = \mathbb{E}[\boldsymbol{z}_i\boldsymbol{z}_i^T] = \frac{d}{\sqrt{r}}\boldsymbol{L}^T\mathbb{E}[\boldsymbol{x}_i\boldsymbol{x}_i^T]\boldsymbol{L} = \frac{d}{\sqrt{r}}\boldsymbol{L}^T\left(\frac{1}{d}\boldsymbol{I}\right)\boldsymbol{L} = \frac{1}{\sqrt{r}}\boldsymbol{I}_r$ and therefore $\boldsymbol{z}_i \sim \mathcal{N}(\boldsymbol{0}, \frac{1}{\sqrt{r}}\boldsymbol{I}_r)$ where we notated the identity as $\boldsymbol{I}_r$ as a reminder that $\boldsymbol{z}_i \in \mathbb{R}^r$. Reusing notation, if $\boldsymbol{\eta}_i \sim \mathcal{N}(\boldsymbol{0}, \boldsymbol{I}_r)$, we can write $\boldsymbol{z}_i = r^{-\frac{1}{4}}\boldsymbol{\eta}_i$ and so $\|\boldsymbol{z}_i\|_2^2 = \frac{1}{\sqrt{r}}\|\boldsymbol{\eta}_i\|_2^2$. By the same arguments as above, for a given $0 < \delta_2 < 1$ we have with probability at least $1 - \delta_2$ that $\max_{i\in[n]}\|\boldsymbol{\eta}_i\|_2^2 < 5r\log\frac{n}{\delta_2}$ and therefore $\max_{i\in[n]}\|\boldsymbol{z}_i\|_2^2 < 5\sqrt{r}\log\frac{n}{\delta_2}$. Applying a similar argument to the user points, letting $\boldsymbol{w}_k := \sqrt{\frac{d}{\sqrt{r}}}\boldsymbol{L}^T\boldsymbol{u}_k$ we have for a given $0 < \delta_3 < 1$ that with probability at least $1 - \delta_3$, $\max_{k\in[K]}\|\boldsymbol{w}_k\|_2^2 < 5\sqrt{r}\log\frac{K}{\delta_3}$. We then have:

$$\max_{i,j\in[n],k\in[K]}|\delta_{i,j}^{(k)}| = \max_{i,j\in[n],k\in[K]}|\boldsymbol{x}_i^T\boldsymbol{M}\boldsymbol{x}_i - \boldsymbol{x}_j^T\boldsymbol{M}\boldsymbol{x}_j + (\boldsymbol{x}_i - \boldsymbol{x}_j)^T\boldsymbol{v}_k|$$

$$= \max_{i,j\in[n],k\in[K]}|\|\boldsymbol{z}_i\|_2^2 - \|\boldsymbol{z}_j\|_2^2 - 2(\boldsymbol{x}_i - \boldsymbol{x}_j)^T\frac{d}{\sqrt{r}}\boldsymbol{L}\boldsymbol{L}^T\boldsymbol{u}_k|$$

$$= \max_{i,j\in[n],k\in[K]}|\|\boldsymbol{z}_i\|_2^2 - \|\boldsymbol{z}_j\|_2^2 - 2(\boldsymbol{z}_i - \boldsymbol{z}_j)^T\boldsymbol{w}_k|$$

$$\leq \max_{i,j\in[n],k\in[K]}\|\boldsymbol{z}_i\|_2^2 + \|\boldsymbol{z}_j\|_2^2 + 2|(\boldsymbol{z}_i - \boldsymbol{z}_j)^T\boldsymbol{w}_k|$$

$$\leq 2\max_{i\in[n]}\|\boldsymbol{z}_i\|_2^2 + 2\max_{i,j\in[n],k\in[K]}|(\boldsymbol{z}_i - \boldsymbol{z}_j)^T\boldsymbol{w}_k|$$

$$\leq 2\max_{i\in[n]}\|\boldsymbol{z}_i\|_2^2 + 2\max_{i,j\in[n],k\in[K]}\|\boldsymbol{z}_i - \boldsymbol{z}_j\|_2\|\boldsymbol{w}_k\|_2$$

$$\leq 2\max_{i\in[n]}\|\boldsymbol{z}_i\|_2^2 + 4\max_{i\in[K]}\|\boldsymbol{z}_i\|_2\max_{k\in[K]}\|\boldsymbol{w}_k\|_2.$$

Taking a union bound over the events $\max_{i\in[n]}\|\boldsymbol{z}_i\|_2^2 \geq 5\sqrt{r}\log\frac{n}{\delta_2}$ and $\max_{k\in[K]}\|\boldsymbol{w}_k\|_2^2 \geq 5\sqrt{r}\log\frac{K}{\delta_3}$ along with a failure of $\|\boldsymbol{x}_i\|_2 \leq B$ for all $i \in [n]$, and setting $\delta_1 = \delta_2 = \delta_3 =: \delta$ for convenience, we have with probability at least $1 - 3\delta$ that these failure events do not occur and so

$$\max_{i,j\in[n],k\in[K]}|\delta_{i,j}^{(k)}| \leq 2\max_{i\in[n]}\|\boldsymbol{z}_i\|_2^2 + 4\max_{i\in[K]}\|\boldsymbol{z}_i\|_2\max_{k\in[K]}\|\boldsymbol{w}_k\|_2$$

$$\leq 10\sqrt{r}\log\frac{n}{\delta} + 20\sqrt{r\log\frac{n}{\delta}\log\frac{K}{\delta}}$$

$$\leq 30\sqrt{r}\log\frac{\max\{n,K\}}{\delta},$$

and so we can set $\gamma = 30\sqrt{r}\log\frac{\max\{n,K\}}{\delta}$.

We next bound $\|\boldsymbol{X}\|$. For convenience, let $\widetilde{\boldsymbol{x}}_i := \sqrt{d}\boldsymbol{x}_i$ such that $\widetilde{\boldsymbol{x}}_i \sim \mathcal{N}(\boldsymbol{0}, \boldsymbol{I}_d)$, and $\widetilde{\boldsymbol{X}} := [\widetilde{\boldsymbol{x}}_1, \ldots, \widetilde{\boldsymbol{x}}_n] = \sqrt{d}\boldsymbol{X}$, so that we have $\|\widetilde{\boldsymbol{X}}\| = \sqrt{d}\|\boldsymbol{X}\|$. From [21], we know that for any $t > 0$,

$$\mathbb{P}(\|\widetilde{\boldsymbol{X}}\| \geq \sqrt{n} + \sqrt{d} + t) < e^{-\frac{t^2}{2}}.$$

For any given $0 < \delta_4 < 1$, let $t = \sqrt{2\log\frac{1}{\delta_4}}$ in which case $\mathbb{P}(\|\widetilde{\boldsymbol{X}}\| \geq \sqrt{n} + \sqrt{d} + \sqrt{2\log\frac{1}{\delta_4}}) < \delta_4$. Therefore, with probability at least $1 - \delta_4$, $\|\boldsymbol{X}\| = \frac{1}{\sqrt{d}}\|\widetilde{\boldsymbol{X}}\| \leq \sqrt{\frac{n}{d}} + 1 + \sqrt{\frac{2}{d}\log\frac{1}{\delta_4}} \leq \sqrt{3(\frac{n}{d} + 1 + \frac{2}{d}\log\frac{1}{\delta_4})}$ since from Jensen's inequality, for any $m$ non-negative scalars $a_1, \ldots, a_m$ we have $\sum_i \sqrt{a_i} \leq \sqrt{m\sum_i a_i}$. Therefore, with probability at least $1 - \delta_4$, $\|\boldsymbol{X}\|^2 \leq 3(\frac{n}{d} + 1 + \frac{2}{d}\log\frac{1}{\delta_4})$.

Finally, we select a setting for $\lambda^*$. Note that by definition each $\boldsymbol{v}_k$ lies in the column space of $\boldsymbol{M}$; hence, $[\boldsymbol{M}, \boldsymbol{v}_1, \ldots \boldsymbol{v}_K]$ is a rank $r$ matrix. By norm equivalence, we have that

$$\|[\boldsymbol{M} \quad \boldsymbol{v}_1 \quad \cdots \quad \boldsymbol{v}_K]\|_* \leq \sqrt{r}\|[\boldsymbol{M} \quad \boldsymbol{v}_1 \quad \cdots \quad \boldsymbol{v}_K]\|_F$$

$$= \sqrt{r(\|\boldsymbol{M}\|_F^2 + \sum_k\|\boldsymbol{v}_k\|_2^2)}$$

$$= \sqrt{r(d^2 + \sum_k\|\boldsymbol{v}_k\|_2^2)}$$

$$\leq \sqrt{r(d^2 + K\max_{k\in[K]}\|\boldsymbol{v}_k\|_2^2)}.$$

Note that $\|\boldsymbol{v}_k\|_2^2 = \|-2\frac{d}{\sqrt{r}}\boldsymbol{L}\boldsymbol{L}^T\boldsymbol{u}_k\|_2^2 = 4\frac{d^2}{r}\boldsymbol{u}_k^T\boldsymbol{L}\boldsymbol{L}^T\boldsymbol{L}\boldsymbol{L}^T\boldsymbol{u}_k = 4\frac{d^2}{r}\boldsymbol{u}_k^T\boldsymbol{L}\boldsymbol{L}^T\boldsymbol{u}_k = 4\frac{d}{\sqrt{r}}\|\boldsymbol{w}_k\|_2^2$. Recall that with probability at least $1 - \delta_3$, $\max_{k\in[K]}\|\boldsymbol{w}_k\|_2^2 < 5\sqrt{r}\log\frac{K}{\delta_3}$. Therefore with probability at least $1 - \delta_3$, $\max_{k\in[K]}\|\boldsymbol{v}_k\|_2^2 < 20d\log\frac{K}{\delta_3}$, i.e.,

$$\|[\boldsymbol{M} \quad \boldsymbol{v}_1 \quad \cdots \quad \boldsymbol{v}_K]\|_* \leq \sqrt{r\left(d^2 + 20dK\log\frac{K}{\delta_3}\right)},$$

and so we can set $\lambda_* = \sqrt{r(d^2 + 20dK\log\frac{K}{\delta_3})}$.

Taking a union bound over all of the event failures described above and taking $\delta_1 = \delta_2 = \delta_3 = \delta_4 = \delta$ for simplicity, where $\delta$ is the same as as in Theorem 3.3, we have with probability greater than $1 - 5\delta$,

$$R(\widehat{\boldsymbol{M}}, \{\widehat{\boldsymbol{v}}_k\}_{k=1}^K) - R(\boldsymbol{M}^*, \{\boldsymbol{v}_k^*\}_{k=1}^K)$$

$$\leq 2L\sqrt{\frac{2\lambda_*^2\log(2d+K)}{|\mathcal{S}|}\left[\left(4(B^2+1) + \frac{4\min(d,n)}{K}\right)\frac{\|\boldsymbol{X}\|^2}{n} + \frac{16B^3}{\sqrt{K}}\right]} + \tag{43}$$

$$\frac{4L(B^2+B)\lambda_*}{3|\mathcal{S}|}\log(2d+K) + \sqrt{\frac{8L^2\gamma^2\log(2/\delta)}{|\mathcal{S}|}},$$

with the selection of $\lambda_*$, $B$, $\|\boldsymbol{X}\|$, and $\gamma$ as described above. To arrive at an order of magnitude statement for this expression, we begin with the term under the first square root:

$$\frac{2\lambda_*^2 \log(2d+K)}{|\mathcal{S}|}\left[\left(4(B^2+1)+\frac{4\min(d,n)}{K}\right)\frac{\|\boldsymbol{X}\|^2}{n}+\frac{16B^3}{\sqrt{K}}\right]$$

$$=\frac{2r(d^2+20dK\log\frac{K}{\delta})\log(2d+K)}{|\mathcal{S}|}\left[3\left(20\log\frac{n}{\delta}+4+\frac{4\min(d,n)}{K}\right)\left(\frac{1}{d}+\frac{1}{n}+\frac{2}{dn}\log\frac{1}{\delta}\right)+\frac{16\left(5\log\frac{n}{\delta}\right)^{\frac{3}{2}}}{\sqrt{K}}\right].$$

If $K=\Omega(d^2)$ and $n\geq d$, then

$$3\left(20\log\frac{n}{\delta}+4+\frac{4\min(d,n)}{K}\right)\left(\frac{1}{d}+\frac{1}{n}+\frac{2}{dn}\log\frac{1}{\delta}\right)+\frac{16\left(5\log\frac{n}{\delta}\right)^{\frac{3}{2}}}{\sqrt{K}}$$

$$\leq\frac{6}{d}\left(20\log\frac{n}{\delta}+4+\frac{4d}{K}\right)\left(1+\log\frac{1}{\delta}\right)+\frac{16\left(5\log\frac{n}{\delta}\right)^{\frac{3}{2}}}{\sqrt{K}}$$

$$=O\left(\frac{1}{d}\right)O\left(\log n+\frac{d}{K}+1\right)+O\left(\frac{(\log n)^{\frac{3}{2}}}{\sqrt{K}}\right)$$

$$=O\left(\frac{1}{d}\right)O\left(\log n+\frac{1}{d}+1\right)+O\left(\frac{(\log n)^{\frac{3}{2}}}{d}\right)$$

$$=O\left(\frac{1}{d}\right)\left[O\left(\log n+(\log n)^{\frac{3}{2}}+1\right)\right]$$

where we have treated $\delta$ as a constant, and so in total the first square root term in eq. (43) scales as

$$O\left(\sqrt{\left[\frac{r(d+K\log\frac{K}{\delta})\log(2d+K)}{|\mathcal{S}|}\right]\left[\log n+(\log n)^{\frac{3}{2}}+1\right]}\right).$$

We ignore the second term in eq. (43), since it decays faster than $\sqrt{\frac{1}{|\mathcal{S}|}}$. Plugging in our selection for $\gamma$, the third term in eq. (43) scales as

$$\sqrt{\frac{7200L^2r\left(\log\frac{\max\{n,K\}}{\delta}\right)^2\log(2/\delta)}{|\mathcal{S}|}}=O\left(\sqrt{\frac{r\left(\log\max\{n,K\}\right)^2}{|\mathcal{S}|}}\right),$$

where we have treated $\delta$ as a constant. By slightly loosening each term's scaling and combining terms, we have

$$R(\widehat{\boldsymbol{M}},\{\widehat{\boldsymbol{v}}_k\}_{k=1}^K)-R(\boldsymbol{M}^*,\{\boldsymbol{v}_k^*\}_{k=1}^K)$$

$$=O\left(\sqrt{\left[\frac{r(d+K\log\frac{K}{\delta})\log(2d+K)}{|\mathcal{S}|}\right]\left[(\log(\max\{n,K\}))^2+(\log n)^{\frac{3}{2}}+1\right]}\right).$$

Suppressing log terms, this equals

$$\widetilde{O}\left(\sqrt{\frac{rd+rK}{|\mathcal{S}|}}\right).$$

# E  Proofs and additional results for recovery guarantees

We can also demonstrate a recovery result in the low-rank setting.

**Theorem E.1.** *Assume the data is gathered as in Theorem 4.1, where $\boldsymbol{M}^*,\{\boldsymbol{v}_k^*\}_{k=1}^K$ satisfy the constraints in (7), and that (7) is solved with loss $\ell_f$. With probability at least $1-\delta$*

$$\frac{1}{n}\sigma_{\min}\left(\boldsymbol{J}[\boldsymbol{X}_\otimes^T,\boldsymbol{X}^T]\right)^2\left(\|\widehat{\boldsymbol{M}}-\boldsymbol{M}^*\|_F^2+\frac{1}{K}\sum_{k=1}^K\|\widehat{\boldsymbol{v}}_k-\boldsymbol{v}_k^*\|^2\right)$$

$$\leq \frac{L}{C_f^2} \sqrt{\frac{\lambda_*^2 \log(2d+K)}{2|\mathcal{S}|} \left[\left(8 + \frac{4\min(d,n)}{K}\right) \frac{\|\boldsymbol{X}\|^2}{n} + \frac{16}{\sqrt{K}}\right]} + \tag{44}$$

$$\frac{2L\lambda_*}{3C_f^2|\mathcal{S}|} \log(2d+K) + \frac{L}{C_f^2} \sqrt{\frac{\gamma^2 \log(2/\delta)}{2|\mathcal{S}|}}. \tag{45}$$

To prove both Theorems 4.1 and E.1, we begin with a helpful lemma that in conjunction with Theorems 3.1 and 3.3 establishes the recovery upper bounds in Theorems 4.1 and E.1.

**Lemma E.2.** *In the same settings as Theorems 4.1 and E.1, we have that*

$$\frac{1}{n}\sigma_{\min}\left(\boldsymbol{J}[\boldsymbol{X}_\otimes^T, \boldsymbol{X}^T]\right)^2 \left(\|\widehat{\boldsymbol{M}} - \boldsymbol{M}^*\|_F^2 + \frac{1}{K}\sum_{k=1}^K \|\widehat{\boldsymbol{v}}_k - \boldsymbol{v}_k^*\|^2\right)$$
$$\leq \frac{1}{4C_f^2}(R(\boldsymbol{M}, \{\boldsymbol{v}_k\}_{k=1}^K) - R(\boldsymbol{M}^*, \{\boldsymbol{v}_k^*\}_{k=1}^K)).$$

*Proof of Lemma E.2.* Recall that $\ell_f(y_p, p; \boldsymbol{M}, \boldsymbol{v}) = -\log(f(y_p \delta_p(\boldsymbol{M}, \boldsymbol{v}_k)))$, and we have that $\mathbb{P}(y_p^{(k)} = -1) = f(-\delta_p(\boldsymbol{M}^*, \boldsymbol{v}_k^*))$. Furthermore, recall that we have taken a uniform distribution over pairs $p$ and users $k$. Hence, it is straightforward to show that we may write the excess risk of any metric $\boldsymbol{M}$ and points $\{\boldsymbol{v}_k\}_{k=1}^K$ as

$$R(\boldsymbol{M}, \{\boldsymbol{v}_k\}_{k=1}^K) - R(\boldsymbol{M}^*, \{\boldsymbol{v}_k^*\}_{k=1}^K) = \frac{1}{K\binom{n}{2}} \sum_{i<j} \sum_{k=1}^K KL(f(-\delta_{ij}(\boldsymbol{M}^*, \boldsymbol{v}_k^*)) \| f(-\delta_{ij}(\boldsymbol{M}, \boldsymbol{v}_k))),$$

where $KL(p \| q) = p\log(p/q) + (1-p)\log((1-p)/(1-q))$. Define $\boldsymbol{\Delta}(\boldsymbol{M}, \{\boldsymbol{v}_k\}_{k=1}^K) \in \mathbb{R}^{\binom{n}{2} \times K}$ such that $[\boldsymbol{\Delta}(\boldsymbol{M}, \boldsymbol{v}_k)]_{p,k} = \delta_p(\boldsymbol{M}, \boldsymbol{v}_k)$, where we slightly abuse notation to let $p$ denote the row of $\boldsymbol{\Delta}$ corresponding to pair $p$. We have the following result (proved at the end of the section):

**Proposition E.3.** *Let $C_f := \min_{x:|x|\leq\gamma} f'(x)$. Then,*

$$\frac{2C_f^2}{K\binom{n}{2}} \left\|\boldsymbol{\Delta}\left(\boldsymbol{M}, \{\boldsymbol{v}_k\}_{k=1}^K\right) - \boldsymbol{\Delta}\left(\boldsymbol{M}^*, \{\boldsymbol{v}_k^*\}_{k=1}^K\right)\right\|_F^2 \leq R(\boldsymbol{M}, \{\boldsymbol{v}_k\}_{k=1}^K) - R(\boldsymbol{M}^*, \{\boldsymbol{v}_k^*\}_{k=1}^K).$$

Next, define $\overline{\boldsymbol{S}} \in \{0,1\}^{\binom{n}{2} \times n}$ to be the *complete* selection matrix of all $\binom{n}{2}$ unique pairs of items such that the $i,j^{\text{th}}$ row is 1 in the $i^{\text{th}}$ column, $-1$ in the $j^{\text{th}}$ column, and 0 otherwise. Note that $\Delta(\cdot, \cdot)$ is linear in both terms. Therefore, we may factor

$$\left\|\boldsymbol{\Delta}\left(\boldsymbol{M}, \{\boldsymbol{v}_k\}_{k=1}^K\right) - \boldsymbol{\Delta}\left(\boldsymbol{M}^*, \{\boldsymbol{v}_k^*\}_{k=1}^K\right)\right\|_F^2 = \sum_{k=1}^K \left\|\overline{\boldsymbol{S}}[\boldsymbol{X}_\otimes^T, \boldsymbol{X}^T] \begin{bmatrix} \boldsymbol{\nu}(\boldsymbol{M} - \boldsymbol{M}^*) \\ \boldsymbol{v}_k - \boldsymbol{v}_k^* \end{bmatrix}\right\|^2.$$

Hence, we may lower bound the above as

$$\sum_{k=1}^K \left\|\overline{\boldsymbol{S}}[\boldsymbol{X}_\otimes^T, \boldsymbol{X}^T] \begin{bmatrix} \boldsymbol{\nu}(\boldsymbol{M} - \boldsymbol{M}^*) \\ \boldsymbol{v}_k - \boldsymbol{v}_k* \end{bmatrix}\right\|^2$$
$$\geq \sigma_{\min}\left(\overline{\boldsymbol{S}}[\boldsymbol{X}_\otimes^T, \boldsymbol{X}^T]\right)^2 \sum_{k=1}^K \left\|\begin{bmatrix} \boldsymbol{\nu}(\boldsymbol{M} - \boldsymbol{M}^*) \\ \boldsymbol{v}_k - \boldsymbol{v}_k^* \end{bmatrix}\right\|^2$$
$$= \sigma_{\min}\left(\overline{\boldsymbol{S}}[\boldsymbol{X}_\otimes^T, \boldsymbol{X}^T]\right)^2 \left(K\|\boldsymbol{\nu}(\boldsymbol{M} - \boldsymbol{M}^*)\|^2 + \sum_{k=1}^K \|\boldsymbol{v}_k - \boldsymbol{v}_k^*\|^2\right)$$
$$\geq \sigma_{\min}\left(\overline{\boldsymbol{S}}[\boldsymbol{X}_\otimes^T, \boldsymbol{X}^T]\right)^2 \left(K\|\boldsymbol{M} - \boldsymbol{M}^*\|_F^2 + \sum_{k=1}^K \|\boldsymbol{v}_k - \boldsymbol{v}_k^*\|^2\right).$$

where $\sigma_{\min}(\boldsymbol{A})$ denotes the smallest singular value of a matrix $\boldsymbol{A}$ and the final inequality follows from the fact that for symmetric $d \times d$ matrix $\boldsymbol{A}$,

$$\|\boldsymbol{\nu}(\boldsymbol{A})\|_2^2 = \|\text{vec}^*(2\boldsymbol{A} - \boldsymbol{I} \odot \boldsymbol{A})\|_2^2 = \sum_{j\geq i}((2 - \mathbb{1}_{i=j})\boldsymbol{A}_{i,j})^2$$

$$= 4 \sum_{j>i} \boldsymbol{A}_{i,j}^2 + \sum_i \boldsymbol{A}_{i,i}^2$$

$$\geq 2 \sum_{j>i} \boldsymbol{A}_{i,j}^2 + \sum_i \boldsymbol{A}_{i,i}^2$$

$$= \sum_{i>j} \boldsymbol{A}_{i,j}^2 + \sum_{i<j} \boldsymbol{A}_{i,j}^2 + \sum_i \boldsymbol{A}_{i,i}^2$$

$$= \|\boldsymbol{A}\|_F^2.$$

Since $n \geq D + d + 1$, surely the selection matrix of all possible paired comparisons $\overline{\boldsymbol{S}}$ contains the construction presented in Appendix C.6 which satisfies the conditions of Proposition 2.2, and so $\overline{\boldsymbol{S}}[\boldsymbol{X}_\otimes^T, \boldsymbol{X}^T]$ is a tall matrix that is full column rank if the items $\boldsymbol{x}_i$ are drawn i.i.d. according to a distribution that is absolutely continuous with respect to the Lebesgue measure, in which case $\sigma_{\min}\left(\overline{\boldsymbol{S}}[\boldsymbol{X}_\otimes^T, \boldsymbol{X}^T]\right)^2 > 0$. To simply this expression even further, we have the following result (proved at the end of the section):

**Proposition E.4.** *For* $\overline{\boldsymbol{S}} \in \mathbb{R}^{\binom{n}{2} \times n}$, $\overline{\boldsymbol{S}}^T \overline{\boldsymbol{S}} = n\boldsymbol{J}$ *for* $\boldsymbol{J} := \boldsymbol{I}_n - \frac{1}{n} \mathbf{1}_n \mathbf{1}_n^T$.

Therefore,

$$\sigma_{\min}\left(\overline{\boldsymbol{S}}[\boldsymbol{X}_\otimes^T, \boldsymbol{X}^T]\right)^2 = n\lambda_{\min}\left(\begin{bmatrix} \boldsymbol{X}_\otimes \\ \boldsymbol{X} \end{bmatrix} \boldsymbol{J}^2 [\boldsymbol{X}_\otimes^T, \boldsymbol{X}^T]\right) = n\sigma_{\min}\left(\boldsymbol{J}[\boldsymbol{X}_\otimes^T, \boldsymbol{X}^T]\right)^2.$$

Finally, note that,

$$\frac{2nC_f^2}{K\binom{n}{2}} \geq \frac{4C_f^2}{Kn}.$$

The proof follows by rearranging terms. $\qquad \square$

*Proof of Proposition E.3.* By Lemma 5.2 of [12], for $(y, z) \in (0, 1)$, $KL(y\|z) \geq 2(y-z)^2$. Now, let $y = f(x)$ and $z = f(x')$, for a continuously differentiable function $f$. Then $2(y - z)^2 \geq 2(\min_a f'(a))^2 (x - x')^2$ since $f$ is monotonic. Applying this to the decomposition of the excess risk alongside the definition of $\Delta(\boldsymbol{M}, \boldsymbol{v}_k)$ establishes the result. $\qquad \square$

*Proof of Proposition E.4.* Note that for $\overline{\boldsymbol{S}} \in \mathbb{R}^{\binom{n}{2} \times n}$, by construction the $i^{\text{th}}$ column corresponds to item $i$ and each row maps to a pair (e.g., $(i, j)$) of the possible $\binom{n}{2}$ unique pairs such that exactly 2 elements are non-zero in each row, with a 1 in the column of one item in the pair and a $-1$ in the other. Since $\overline{\boldsymbol{S}}^T \overline{\boldsymbol{S}}$ is a Gram matrix, it is sufficient to characterize the inner products between any two columns of $\overline{\boldsymbol{S}}$. First, for the $i^{\text{th}}$, note that $i$ can be paired with $n - 1$ other items uniquely. Hence, there are exactly $n - 1$ non-zero entries in each column all of which are 1 or $-1$. Hence, every diagonal entry of $\overline{\boldsymbol{S}}^T \overline{\boldsymbol{S}}$ is $n - 1$. For the off diagonal entries, consider a pair $i \neq j$. As each row corresponds to a unique pair, the supports of $i$ and $j$ overlap in a single entry corresponding to the $(i, j)$ pair. By construction one column has a 1 at this entry and the other has a $-1$. Hence the inner product between these two columns is $-1$ and all off diagonal entries of $\overline{\boldsymbol{S}}^T \overline{\boldsymbol{S}}$ are $-1$. Hence, $\overline{\boldsymbol{S}}^T \overline{\boldsymbol{S}} = n\boldsymbol{I} - \mathbf{1}_n \mathbf{1}_n^T = n\boldsymbol{J}$. $\qquad \square$

# F    Additional experimental details

In this section we provide additional experimental details and results.

## F.1    Datasets

The color preference data was originally collected by [24] and we include a .mat file of the dataset in the paper supplement along with a full description in the code README document. Before running our learning algorithms, we centered the $3 \times 37$ item matrix $\boldsymbol{X}$ of CIELAB coordinates, and

normalized the centered coordinates by the magnitude of the largest norm color in CIELAB space such that $\max_{i \in [n]} \|\boldsymbol{x}_i\|_2 = 1$ after centering and normalization.

For both the normally distributed items and color experiments, we performed train and test dataset splits over multiple simulation runs, and averaged results across each run. For each simulation run, we *blocked* the train/test splitting by user, in that all users were queried equally in both training and test data. Specifically, during each run the dataset was randomly shuffled *within* each user's responses, and then a train/test split created. For the normally distributed data, this consisted of 300 training comparisons per user, and 300 test comparisons. For the color dataset, each user provided $37 \times 36 = 1332$ responses, which was partitioned into a training set of 300 pairs and a test set of 1032 pairs. To vary the number of training pairs per user, for both datasets we trained incrementally on the 300 pairs per user in a randomly permuted order (while evaluating on the full test set).

For both normally distributed items and color preference data, we repeat 30 independent trials. In the color preference data, since the dataset is fixed ahead of time the only difference between each trial is the train/test splitting as detailed above. In the normally distributed experiments, we generate responses as $\mathbb{P}(y_{i,j}^{(k)} = -1) = (1 + e^{\beta(\boldsymbol{x}_i^T \boldsymbol{M}^* \boldsymbol{x}_i - \boldsymbol{x}_j^T \boldsymbol{M}^* \boldsymbol{x}_j + (\boldsymbol{v}_k^*)^T (\boldsymbol{x}_i - \boldsymbol{x}_j)))^{-1}$ for a noise scaling parameter $\beta > 0$.

### F.2   Implementation and computation

In out experiments, we did not enforce the $\gamma$ constraints required for our theoretical results (i.e., the $\gamma$ constraints in eqs. (5) and (7)). This constraint is added to the theory to guard against highly coherent $\boldsymbol{x}_i$ vectors. In the simulated instances, this quantity appears to be controlled by the isotropic nature of both the normally distributed and color datasets, along with the fact that we are constraining the norms of the latent parameters to be learned.

For all simulated experiments, we leveraged ground-truth knowledge of $\boldsymbol{M}^*$ and $\boldsymbol{v}_k^*$ to set the hyper-parameter constraints. This was done to compare each method under its best possible hyperparameter tuning — namely, the smallest norm balls that still contained the true solution. Specifically, we set hyperparameters for the normally distributed items experiment as follows:

- **Frobenius metric**: $\|\boldsymbol{M}\|_F \leq \|\boldsymbol{M}^*\|_F$, for all $k \in [K]$, $\|\boldsymbol{v}_k\|_2 \leq 2\max_{k \in [K]} \|\boldsymbol{M}^* \boldsymbol{u}_k^*\|$

- **Nuclear full**: $\|[\boldsymbol{M}, \boldsymbol{v}_1, \cdots, \boldsymbol{v}_K]\|_* \leq \|[\boldsymbol{M}^*, -2\boldsymbol{M}^* \boldsymbol{u}_1^*, \cdots, -2\boldsymbol{M}^* \boldsymbol{u}_K^*]\|_*$

- **Nuclear metric**: $\|\boldsymbol{M}\|_* \leq \|\boldsymbol{M}^*\|_*$, for all $k \in [K]$, $\|\boldsymbol{v}_k\|_2 \leq 2\max_{k \in [K]} \|\boldsymbol{M}^* \boldsymbol{u}_k^*\|$

- **Nuclear split**: $\|\boldsymbol{M}\|_* \leq \|\boldsymbol{M}^*\|_*$, $\|[\boldsymbol{v}_1, \cdots, \boldsymbol{v}_K]\|_* \leq 2\|\boldsymbol{M}^*[\boldsymbol{u}_1^*, \cdots, \boldsymbol{u}_K^*]\|_*$

- **Nuclear full, single**: for each $k \in [K]$, $\|[\boldsymbol{M}, \boldsymbol{v}_k]\|_* \leq \|[\boldsymbol{M}^*, -2\boldsymbol{M}^* \boldsymbol{u}_k^*]\|_*$.

For the color preferences experiment, we set all hyperparameters under an a priori estimate of $\boldsymbol{M}^* = \boldsymbol{I}$ (due to the assumed perceptual uniformity of CIELAB space). Specifically, we constrained $\|\boldsymbol{M}\|_F \leq \|\boldsymbol{I}\|_F = \sqrt{3}$ (since $d = 3$), and constrained $\|\boldsymbol{v}_k\|_2 \leq 2$, since under the heuristic assumption that $\boldsymbol{M}^* = \boldsymbol{I}$ we have $\|\boldsymbol{v}_k\|_2 = \|-2\boldsymbol{M}\boldsymbol{u}_k\|_2 = 2\|\boldsymbol{u}_k\| \lesssim 2\max_{i \in [n]} \|\boldsymbol{x}_i\|_2 = 2$, where in the last inequality we have approximated the distribution of ideal points $\boldsymbol{u}_k$ with the empirical item distribution over centered, scaled CIELAB colors (which have maximal norm of 1 as described above).

To solve these optimizations, we leveraged CVXPY[9] with different solvers. When learning on normally distributed items, we set $\ell(x) = \log(1 + \exp(-\beta x))$ to be the logistic loss, where $\beta > 0$ is the same parameter used to generate the response noise. We used the Splitting Conic Solver with a convergence tolerance set to $1e - 6$ to balance between accuracy and computation time. For the color preference data, we used the hinge loss $\ell(x) = \max\{0, 1 - x\}$, solved using the CVXOPT solver with default parameters, which we found performed more stably than SCS. As an additional safeguard for numerical stability due to the presence of negative eigenvalues near machine precision, we project all learned metrics back onto the positive semidefinite cone after solving with CVXPY. All additional code was written in Python, and all experiments were computed on three Dell 740 servers with 36, 3.1 GHz Xeon Gold 6254 CPUs.

---

[9] https://www.cvxpy.org/

When estimating ideal points $\widehat{\boldsymbol{u}}_k$ from $\widehat{\boldsymbol{M}}$ and $\widehat{\boldsymbol{v}}_k$, rather than using the exact pseudo-inverse we perform a regularized recovery as in [7], since $\widehat{\boldsymbol{M}}$ may have recovery errors. Specifically, for regularization parameter $\alpha > 0$ we estimate $\widehat{\boldsymbol{u}}_k$ as

$$\widehat{\boldsymbol{u}}_k = -2(4\widehat{\boldsymbol{M}}^2 + \alpha\boldsymbol{I})^{-1}\widehat{\boldsymbol{M}}^T\widehat{\boldsymbol{v}}_k. \tag{46}$$

We only perform ideal point recovery in the normally distributed item experiments, since no ground-truth ideal points are available in the real-world color preference data. Since we know a priori that $\boldsymbol{u}_k \sim \mathcal{N}(\boldsymbol{0}, \frac{1}{d}\boldsymbol{I})$, we can leverage the interpretation of the recovery estimate in eq. (46) as the maximum a posteriori estimator under a Gaussian prior over $\boldsymbol{u}_k$ with Gaussian observations in order to set $\alpha = d$.

### F.3 Additional experiments and details

Below we present experimental results in additional simulation settings, as well as supplementary figures for the data presented in the main paper body. We detail specific performance metrics below: in the following, let $\widehat{\boldsymbol{V}} := [\widehat{\boldsymbol{v}}_1, \cdots, \widehat{\boldsymbol{v}}_K]$, $\boldsymbol{U}^* := [\boldsymbol{u}_1^*, \cdots, \boldsymbol{u}_K^*]$ $\boldsymbol{V}^* := -2\boldsymbol{M}^*\boldsymbol{U}^*$, and $(\boldsymbol{M}^*)^\dagger$ denote the pseudoinverse of the ground-truth metric.

- *Test accuracy*: fraction of test data responses predicted correctly from $\mathrm{sign}(\widehat{\delta}_{i,j}^{(k)})$, where $\widehat{\delta}_{i,j}^{(k)}$ is computed as in eq. (2) using $\widehat{\boldsymbol{M}}$ and $\widehat{\boldsymbol{v}}_k$ as parameter estimates.
- *Relative metric error*: $\frac{\|\widehat{\boldsymbol{M}} - \boldsymbol{M}^*\|_F}{\|\boldsymbol{M}^*\|_F}$
- *Relative ideal point error*: $\frac{\|\widehat{\boldsymbol{U}} - (\boldsymbol{M}^*)^\dagger \boldsymbol{M}^* \boldsymbol{U}^*\|_F}{\|(\boldsymbol{M}^*)^\dagger \boldsymbol{M}^* \boldsymbol{U}^*\|_F}$. We compare recovery error against $(\boldsymbol{M}^*)^\dagger \boldsymbol{M}^* \boldsymbol{U}^*$ rather than $\boldsymbol{U}^*$ since if $\boldsymbol{M}^*$ is low-rank, then the components of $\boldsymbol{U}^*$ in the kernel of $\boldsymbol{M}^*$ are not recoverable.
- *Relative pseudo-ideal point error*: $\frac{\|\widehat{\boldsymbol{V}} - \boldsymbol{V}^*\|_F}{\|\boldsymbol{V}^*\|_F}$

In Figure 1e, we compute the heatmap of the crowd's metric as the empirical average of the learned metrics over all independent trials.

We present additional simulation results with normally distributed items in two noise regimes — "high" noise with $\beta = 1$ in the logistic model, and "medium" noise with $\beta = 4$ in the logistic model. We generate the dataset in the same manner as in Section 5. We present results for the low-rank case as in the main paper body ($d = 10$, $r = 1$) as well as the full-rank case ($d = r = 10$). In the full-rank case, to generate a ground-truth metric $\boldsymbol{M}^*$ we generate a $d \times r$ matrix $\boldsymbol{L}$ whose entries are sampled independently according to the standard normal distribution, compute $\boldsymbol{M}^* = \boldsymbol{L}\boldsymbol{L}^T$, and normalize $\boldsymbol{M}^*$ such that it has a Frobenius norm of $d$. Otherwise, if we generated $\boldsymbol{L}$ as in the low-rank experiments, $\boldsymbol{M}^*$ would simply become a scaled identity matrix.

Fig. 2 repeats the main results in the paper body, with an additional subfigure depicting recovery error for pseudo-ideal points. This is a "high" noise, low-rank setting. The remaining figures are: "high" noise full-rank metric (Fig. 3); "medium" noise low-rank metric (Fig. 4); and "medium" noise full-rank metric (Fig. 5). In Fig. 6 we analyze the color prediction results from Fig. 1d in the low query count regime.

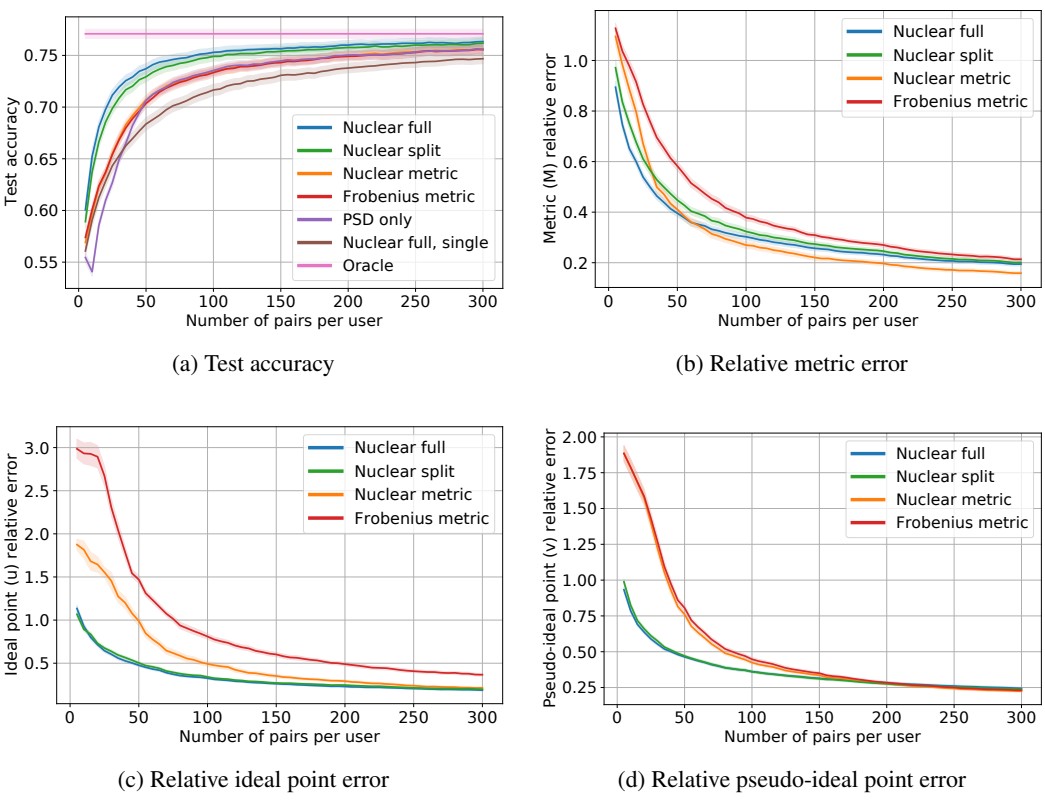

Figure 2: Full prediction and recovery results for a high noise setting ($\beta = 1$) with a low-rank metric ($r = 1$). Error bars indicate $\pm 1$ standard error about the sample mean. (a-c) appear in the main paper body, with (d) added here for completeness. **Nuclear full** gives the best performance on prediction and **Nuclear split** is a close second (subfigure a), reflecting the necessity of modeling the low-rank nature of the $M^*$ and $V^*$. While the **Nuclear metric** method that only places a nuclear norm constraint on $M$ performs well in terms of relative error for recovering $M^*$ (subfigure b), it achieves far worse performance for estimating $U^*$ and $V^*$ as shown in subfigures (c) and (d). This reflects the importance of enforcing that $\widehat{M}$ and $\widehat{V}$ share a column space.

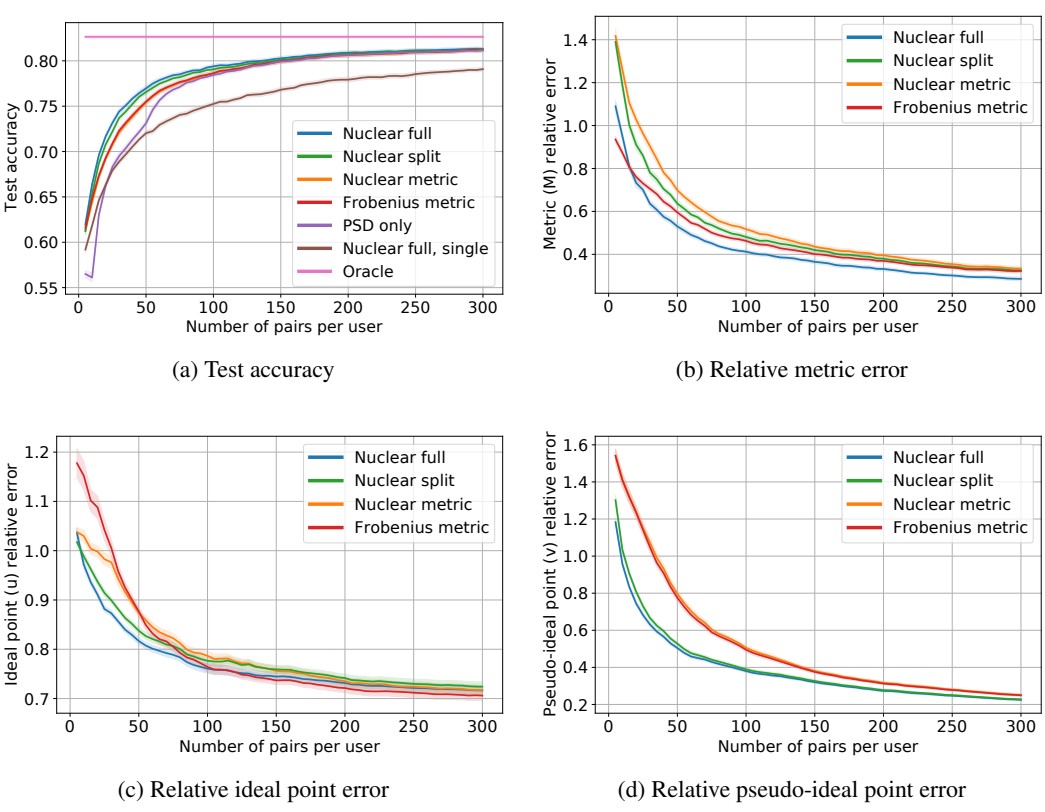

Figure 3: Prediction and recovery results for a high noise setting ($\beta = 1$) with a full-rank metric ($d = r = 10$). Error bars indicate $\pm 1$ standard error about the sample mean. Surprisingly, even in the full-rank scenario, **Nuclear full** demonstrates the highest prediction performance, even in comparison to **Frobenius metric** which is designed for full-rank metrics. That said, the difference is less stark for estimating both $M^*$ and $U^*$.

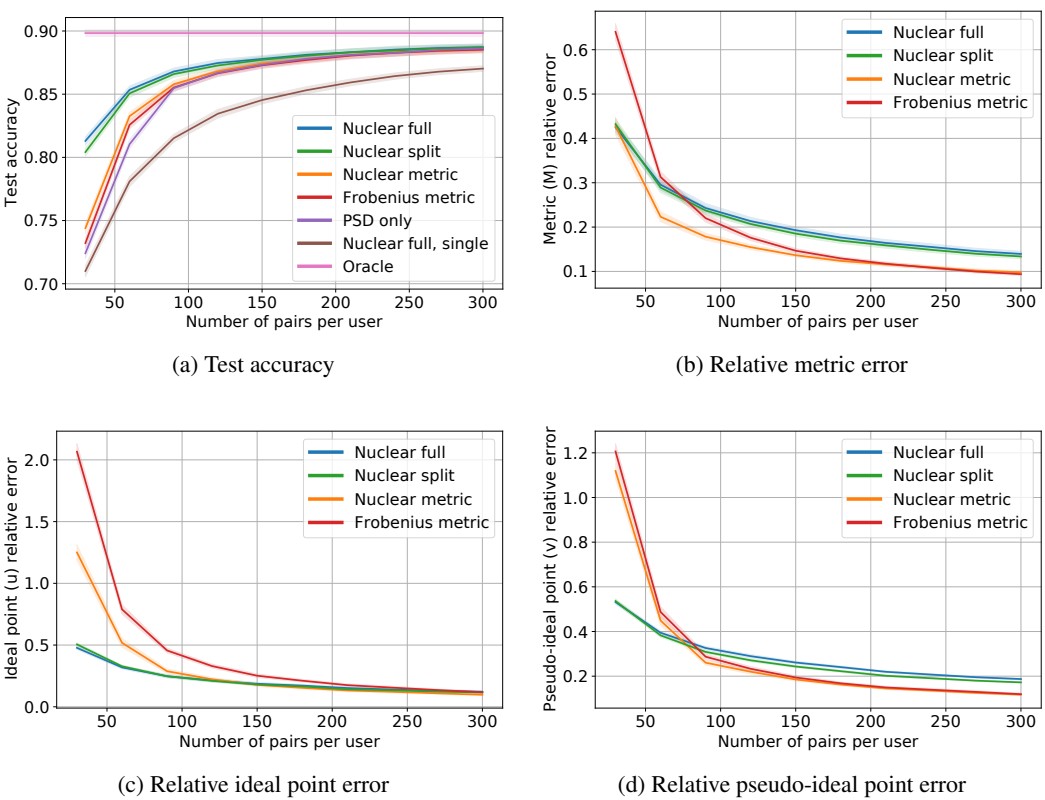

(a) Test accuracy

(b) Relative metric error

(c) Relative ideal point error

(d) Relative pseudo-ideal point error

Figure 4: Prediction and recovery results for a medium noise setting ($\beta = 4$) with a low-rank metric ($r = 1$). Error bars indicate $\pm 1$ standard error about the sample mean. Similar trends as Figure 2 hold except that the differences are less pronounced. Interestingly, for large numbers of samples per user, **Nuclear metric** and **Frobenius metric** appear to achieve better performance on estimating $\boldsymbol{M}^*$ and therefore achieve better performance for estimating $\boldsymbol{V}^*$, though they do not achieve as strong of performance for estimating $\boldsymbol{U}^*$, perhaps because these methods do not enforce that $\widehat{\boldsymbol{M}}$ and $\widehat{\boldsymbol{V}}$ share a column space.

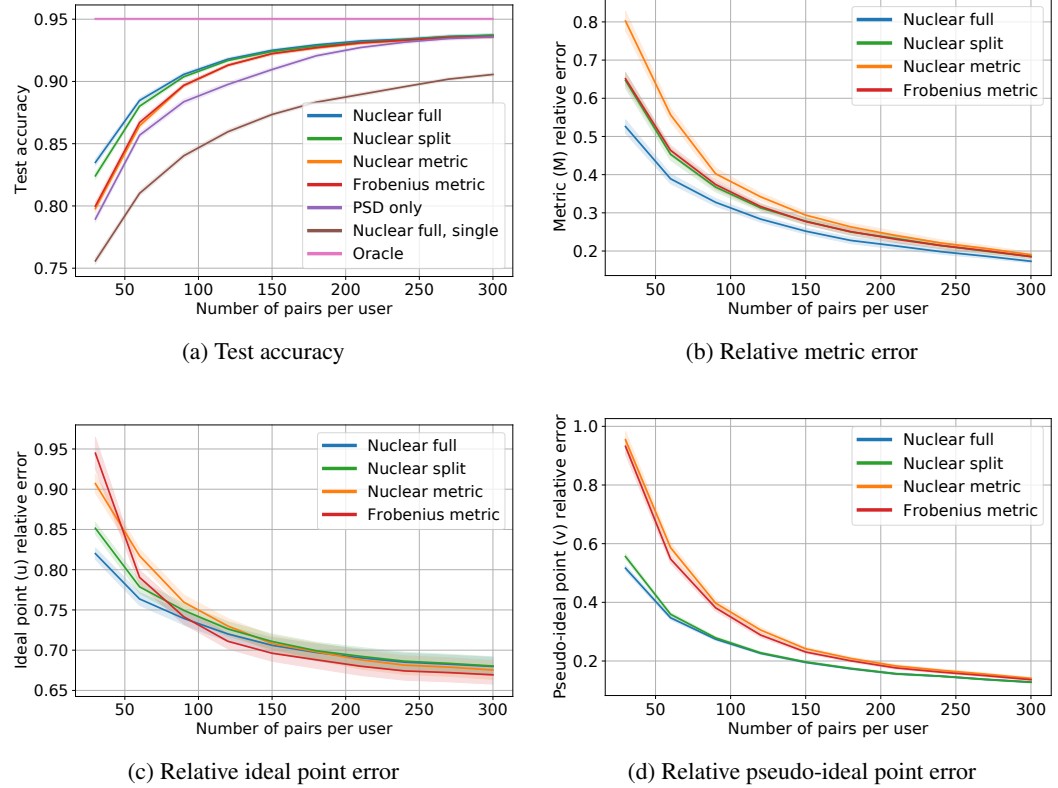

(a) Test accuracy

(b) Relative metric error

(c) Relative ideal point error

(d) Relative pseudo-ideal point error

Figure 5: Prediction and recovery results for a medium noise setting ($\beta = 4$) with a full-rank metric ($d = r = 10$). Error bars indicate $\pm 1$ standard error about the sample mean. Similar trends as Figure 3 hold in this case.

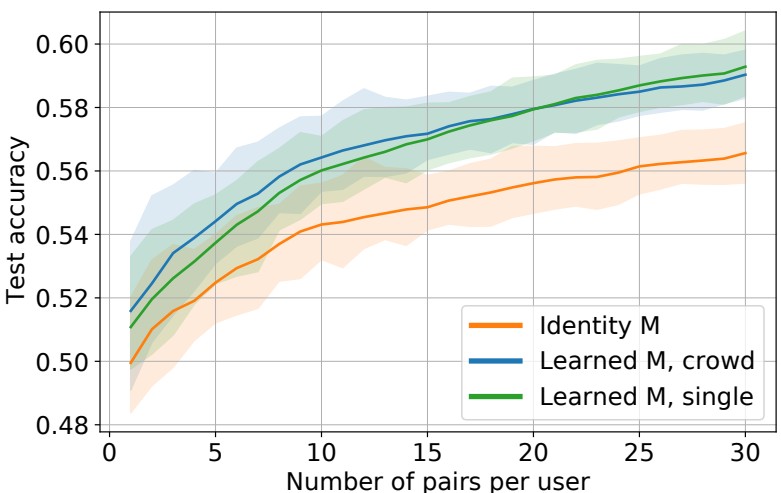

Figure 6: Comparison of methods on color preference data in the low query regime, with error bars representing 2.5% and 97.5% percentiles. The identity metric performs more poorly than the methods that learn a metric tuned to user judgments. The method that learns a single $M$ for the crowd and method that learns a $M$ for each individual perform similarly in most of the range of number of pairs. There is a slight advantage to the method that learns a metric for the crowd when very few pairs have been given to each user. This stems from the fact that the crowd metric can amortize the cost of learning the metric over the responses given by all users.

### F.4 Comparison with [7] in the single user case

Although the focus of this work is developing and analyzing algorithms for simultaneous metric and preference learning over *multiple* users, for completeness we compare the performance of our approach against the algorithms presented in [7], which only study simultaneous metric and preference learning in the *single* user case ($K = 1$, in our notation). Xu and Davenport propose both "single-step" and "alternating" estimation algorithms, referred to here as **XDsingle** and **XDalt** respectively (we direct the reader to [7] for the details of these algorithms). Since [7] does not specifically consider the low-rank metric setting, we evaluate these methods on the full-rank simulated data described in Fig. 3. To run their algorithms, we use the default hyperparameter values described in their paper and found in their code (see [7] for details and code link).

In the table below we evaluate their algorithms' performance using the same procedure as in Fig. 3, at sampled intervals of 50 queries per user. For reference, we also copy the corresponding values from **Nuclear full, single** in Fig. 3. We evaluate **XDsingle** and **XDalt** in an analogous manner to **Nuclear full, single**: the crowd of $K = 10$ simulated users generates responses according to a common metric, but the single user algorithms are solved separately, learning a separate metric for each user. Test accuracy, metric recovery error, and ideal point recovery error are then averaged over all users.

| Number of queries per user | 50 | 100 | 150 | 200 | 250 | 300 |
|---|---|---|---|---|---|---|
| **Nuclear full, single** | 0.720 | 0.752 | 0.768 | 0.779 | 0.785 | 0.791 |
| **XDsingle** | 0.641 | 0.682 | 0.701 | 0.712 | 0.727 | 0.731 |
| **XDalt** | 0.683 | 0.715 | 0.737 | 0.745 | 0.757 | 0.762 |

(a) Test accuracy

| Number of queries per user | 50 | 100 | 150 | 200 | 250 | 300 |
|---|---|---|---|---|---|---|
| **Nuclear full, single** | 1.134 | 0.975 | 0.874 | 0.797 | 0.747 | 0.704 |
| **XDsingle** | 1.003 | 1.889 | 1.816 | 1.513 | 1.233 | 1.085 |
| **XDalt** | 0.791 | 0.757 | 0.733 | 0.715 | 0.680 | 0.674 |

(b) Relative metric error

| Number of queries per user | 50 | 100 | 150 | 200 | 250 | 300 |
|---|---|---|---|---|---|---|
| **Nuclear full, single** | 0.985 | 0.879 | 0.841 | 0.819 | 0.794 | 0.782 |
| **XDsingle** | 1.000 | 0.963 | 0.910 | 0.882 | 0.862 | 0.866 |
| **XDalt** | 0.922 | 0.906 | 0.887 | 0.887 | 0.875 | 0.875 |

(c) Relative ideal point error

Table 1: Comparison of **Nuclear full, single** against the methods proposed in [7] for the *single* user case. We evaluate prediction accuracy and recovery error (averaged over 30 trials) for a high noise setting ($\beta = 1$) with a full-rank metric ($d = r = 10$). Simulation details are otherwise identical to those in Fig. 1a-c.

Overall, among the single user methods **Nuclear full, single** recovers the highest test prediction accuracy and lowest ideal point recovery error, while **XDalt** results in the lowest metric estimation error. While our work and [7] share the same model in the single user case (as a reminder, [7] does not consider the multi-user case), each frames the problem differently and solves a different optimization problem, so it is intuitively unclear how each single user algorithm differs theoretically.