# OpenReview forum: "One for All: Simultaneous Metric and Preference Learning over Multiple Users"
_NeurIPS.cc/2022/Conference — NeurIPS 2022 Accept_

### Official Review · Reviewer_eLgv · 2022-07-07

**Rating:** 8
**Confidence:** 3
**Soundness:** 4 excellent
**Presentation:** 4 excellent
**Contribution:** 4 excellent

**Summary:**

Authors propose a preference learning method which ideal points of multiple users and the underlying Mahalanobis metric are jointly learned. In the noiseless setting which the exact differences between distances from ideal points to items are learned, authors provide sufficient and necessary conditions for identifiability. When only noisy binary labels are available, authors provide sample complexity analysis for both full-rank and low-rank metrics. Recovery guarantees of parameters are also provided. There are two empirical results. In the simulation experiments, authors study the statistical efficiency of the proposed method and its variants. In the real data example, authors experiment on pairwise color preference data, which finding matches prior findings from the area of the application.

**Questions:**

When there is only a single user, the estimation method proposed in this paper is quite different from that of Xu and Davenport (NeurIPS 2020). Would it be of interest to compare the proposed estimation method with Xu and Davenport in the special case of single user? This will allow us to understand the impact of differences between these two estimation strategies.

**Limitations:**

In-depth discussion on limitations and societal impact are provided in Appendix A.1 and A.2.

I also appreciate that in the real data experiments, authors validate their findings with published work from the area the data are gathered from. However, I still wonder what prevents authors from running larger-scale experiments from learning-to-rank datasets or recommender system datasets. The proposed method may not achieve the best NDCG, but such an experiment will give practitioners better idea about how applicable the proposed method is to larger-scale problems? While it's good to know an externally validated finding could be recovered in the real data experiment, the demonstrated practical value of the method still seems limited.

**Strengths And Weaknesses:**

First of all, the theoretical analysis is of high quality and quite comprehensive. Authors go well beyond the prior analysis of Xu and Davenport (NeurIPS 2010), which only provided identifiability analysis in a much simpler single-user setting. Sufficient conditions of identifiability provides a good intuition on how the sample complexity for estimating the metric $M$ is amortized. Actual sample complexity analysis for noisy settings also match this intuition on the amortization of the cost for estimating $M$. This analysis is also built upon the analysis of preference matrix, which could be of separate interest in future studies.

Second, the paper is very clearly written. Almost authors provide intuitive explanation on every mathematical object they introduce. The analysis naturally progresses from simpler setting to more complex settings, building intuition first and leveraging that intuition for the discussion of higher technical complexity.

While it's good to know an externally validated finding could be recovered in the real data experiment, the demonstrated practical value of the method still seems limited.

---

> ### Author Response · Authors · 2022-08-02
> **Thank you for reviewing our paper**
>
> Thank you for taking the time to read and review our work. Please refer to the general response to all reviewers for a summary of the core points of our submission as well as for responses to questions shared by multiple reviewers. Below, we respond to your individual points directly.
>
> Please see the general response for comments about a comparison to Xu and Davenport.
>
> Regarding other experimental results: we appreciate that you have taken the time to recognize the theoretical contributions of this work. Our core focus in these experiments was to show empirically what we have claimed theoretically. In particular, our emphasis was to demonstrate the amortization effect that is observed in large crowds of users, and to provide initial experimental results for our mostly theoretical work. The color dataset is particularly well suited to this aim since all respondents were shown the same set of colors and the feature vectors themselves were constructed to be a perceptually uniform, linear space. This mitigates possible sources of error in the results due to model mismatch. We agree that testing this model on a greater range of datasets to study the flexibility of the model itself is an interesting avenue for future work. Otherwise, please see the general response for additional comments and questions regarding additional real-world datasets.

---

### Official Review · Reviewer_u2T5 · 2022-07-13

**Rating:** 6
**Confidence:** 4
**Soundness:** 2 fair
**Presentation:** 2 fair
**Contribution:** 3 good

**Summary:**

This paper studies a preference learning problem, in which each item and user is associated with a latent position, and when a user is close to an item (under the Mahalanobis-norm), the user likes the item more. The problem assumes that the item’s latent positions are known, the users’ latent positions are unknown and the Mahalanobis-norm (the $M$ matrix in the paper) is unknown. The goal is to estimate the unknowns.

The theoretical result looks quite strong (if they are correct) and the experiments seem to be too weak to be forgivable.


**Questions:**

Please see above.

**Limitations:**

Please see the strength & weakness.

**Strengths And Weaknesses:**

My concerns:

(*) Why this particular model?

Specifically, there are three lines of closely related modeling:

1. If the preference function is in inner product form, e.g., whether a user likes an item is a function of <x, u>, where x is known and u is unknown, then this is a standard linear/logistic regression.
2. If both x and u are known, $M$ is unknown (but assumed to be low rank), then it is also extensively studied (an EC result a few years back; I cannot quite find the paper; sorry; nevertheless, the results probably are implied by recent works by Negahban and Wainwright).
3. If both x and u are unknown, the preference function is distance between latent positions, it is also studied by Katz-Samuels and Scott 2018 and Liu, Wu, Liu, Xia 19.

So the proposed model seems to be a very special case that is not considered in the literature. I dont see strong argument why this specific formulation is needed. I can think of two ways to rebuttal: (i) there are many practical settings, in which this model is more relevant; this probably cannot fly because of the weak experiments, (ii) the new problem is mathematically interesting because it exhibits new structure that 1, 2, and 3 does not have — this may fly.

(**) some technical details
I cannot quite parse line 195 to line 221 (around Theorem 3.1). The authors formulated an optimization problem, which is not convex. Then the author proceeds to introduce a new set of variables $v_k = -2 M u_k$ and translates the problem to Eq. 2. (i) It is not completely clear to me this is a relaxation. It is not obvious that a feasible solution to Eq. 2 implies a feasible solution for the original problem (especially enforcing |u_k| < \lambda_u via |v_k| < \lambda_v and |M|_F < lambda_F seems difficult). This kind of M u_k interaction looks very painful so if it would be quite impressive if it is handled correctly. (ii) this change of variable almost resembles the inner product model mentioned above; so what's the difference in terms of techniques and why not directly inner product model?

The low-rank result appears to be interesting: the standard route would just hope that $M$ is low rank so the regularizer is nuclear norm of $M$. But the authors seem to be saying that for this specific case, doing the nuclear norm for $[M, v_1, …, v_k]$ is more suitable. If Thm 3.1 is correct/meaningful, then Thm 3.3 is believable and probably involves some non-trivial analysis.

(***) Experiments:
Using color dataset seems to be quite strange. There are many benchmark dataset available and I feel unsure why the authors do not use them — maybe minimally try the original netflix for a theory result?

---

> ### Author Response · Authors · 2022-08-02
> **Thank you for reviewing our paper**
>
> Thank you for taking the time to read and review our work. Please refer to the general response to all reviewers for a summary of the core points of our submission as well as for responses to questions shared by multiple reviewers. Below, we respond to your individual points directly.
>
> **Regarding our model of x being known, but M and u being unknown:** This model follows from prior art in the literature and is not a creation of this work. Our work directly extends and provides theoretical analysis to [7] which proposes the model herein for the case of a single user. In general, when x is known but u is unknown this becomes the “ideal point” model as studied by [5, 20] among others. Using distance to an unknown point to model human preferences is a common assumption in the psychometrics literature, eg, [1, 2]. We extend the work of [7] which studies this model for a single user, additionally including M being unknown. The addition of an unknown metric M allows for added flexibility of the model, as we show empirically with the color preference dataset. As an example of this added flexibility, extending the proof of [5] there are $O(n^d)$ possible preference rankings of n items for a known M, but $O(n^{d^2})$ possible rankings when M is learnable. A core result of our work is to show that this added flexibility can be achieved nearly for free by amortization over the crowd.
>
> Conversely, when x and u are both unknown, this problem becomes ordinal embedding which is a special case of metric learning [12]. This is a mathematically and semantically different problem than the one considered herein where we are modeling a set of unknown preferences amongst known items.
>
> **Regarding mathematical novelty and the convex relaxation:** you are correct in identifying some of the core mathematical challenges of this work: 1) showing the model is identifiable at all and 2) showing that the convex relaxation has the same solution as the nonconvex problem. Neither challenge is non-trivial mathematically. To handle the first point in the noiseless, unquantized measurement case, we develop new theory for selection matrices in Section 2 which precisely characterizes when the system of equations admits a unique solution for $M$ and $v$. If $M$ and $v$ are estimated exactly, then $u$ can be recovered from $M$ exactly if $M$ is full rank. Otherwise if $M$ is rank deficient, as we discuss in footnote 2 only the component of $u$ in the row space of $M$ has any impact on the measurement model, and this component can be recovered exactly. We believe this is a key insight that decouples the non-convex nature of this measurement model into two stages: simultaneously estimate $M$ along with the ideal point component in the column space of $M$ (given by $v$), in which case the measurement model becomes linear in $M$ and $v$, and then estimate $u$ from $M$ and $v$.
>
> To handle the second point, note that we may choose $\lambda_v = 2 \lambda_F \lambda_u$, resulting in a larger constraint set than the original problem and so the optimization is a relaxation of the previous (proof: for the “true” $v$, $\lVert v \rVert_2 = 2 \lVert M u \rVert_2 \le 2 \lVert M \rVert \lVert u \rVert_2 \le 2 \lVert M \rVert_F \lVert u \rVert_2 \le 2 \lambda_F \lambda_u$). From there, showing that we can recover $u^*$ is equivalent to showing that as long as the true $v^* = -2 M^* u^*$ lives in the constraint set, our optimization is a consistent estimator. Specifically, our results in Section 4 imply that the convex relaxation in the presence of noisy data (with a known link function) asymptotically recovers the correct values of $M$ and $v$, in which case we can recover $u$ exactly as discussed above, and so the convex relaxation asymptotically recovers the solution to the noiseless, nonconvex problem. In the finite sample regime, it is left to future work to study the propagation of error in estimating $M$ and $v$ and how it impacts the estimate of $u$, which may involve non-trivial analysis. However, this is a very interesting question, and as we demonstrate empirically in Figs. 1b and 1c we believe that the resulting estimate for $u$ is also consistent. We will be sure to enhance the discussion of these points in the next revision.

---

> > ### Author Response · Authors · 2022-08-02
> > **reviewer response, part 2**
> >
> > **Regarding differences with inner product model:** first, the ideal point model (distance based) is characterized by a fundamentally different geometry than a model of preference based on inner products. The core modeling difference comes down to whether or not magnitudes have a monotonic influence on preference: in an inner product model where items with larger values of $\langle u, x \rangle$ are preferred, an item’s preference can be increased simply by increasing its magnitude. Qualitatively, this may be suboptimal in certain applications, such as in flavor mixing in beverages where amplified feature intensity (e.g., more coffee acidity) does not always mean that the item will be preferred over its less amplified counterpart.
> >
> > On the other hand, the ideal point model accounts for item magnitude. Quantitatively, it has been demonstrated in the psychometrics literature that such a distance-based model can more accurately model preference than an inner product (i.e., attribute based) model (see Dubois 1975 - this was cited in [7], and we will include this citation in our next revision). These modeling differences are non-trivial to handle mathematically, and require a reformulation of the problem - rather than searching for an ideal angle on a sphere in the inner product model, the ideal point  model searches for a localized neighborhood in all of $\mathbb{R}^d$ (see for instance [5]). We will be sure to expand on this discussion in our next revision.
> >
> > To address the reviewer’s specific question: recall that (in the noiseless case for simplicity), $x_i$ is preferred to $x_j$ if and only if $\lVert x_i - u \rVert_M^2 - \lVert x_j - u \rVert_M^2< 0$. If we define $v = -2 M u$, this difference of squared distances becomes $\lVert x_i \rVert_M^2 - \lVert x_j \rVert_M^2 + (x_i - x_j)^T v$. If it so happened that all items lie on a sphere with respect to the Mahalanobis norm according to $M$ (i.e., $\lVert x_i \rVert_M^2 = \lVert x_j \rVert_M^2$), then indeed this difference of squared distances is equal to $\langle v, x_i - x_j \rangle$, meaning that item $x_i$ is preferred if and only if $\langle v, x_i \rangle < \langle v, x_j \rangle$, in which case the ideal point and inner product models give equivalent preference scoring up to a sign (with respect to “pseudo-ideal point” $v$). However, it is almost surely not the case that $\lVert x_i \rVert_M^2 = \lVert x_j \rVert_M^2$ for all items: recall that $M$ is an arbitrary PSD matrix to be learned, and in general may consist of a non-trivial reweighting of item features. These item norm terms (with respect to the Mahalanobis norm given by $M$) have a very non-trivial impact on the measurement model beyond the term that simply takes inner products with $v$.
> >
> > **Regarding experiments:** the core contribution of this work is developing theory for simultaneous metric and preference learning, and the experiments are intended as initial, easily interpreted validations of the theoretical claims regarding prediction and recovery. We chose the color preference data as a multi-user, publicly available analog of the graduate admissions dataset originally included in [7]. A strength of this color data is that the features exist in a linear and perceptually uniform color space and the comparisons were collected under laboratory conditions, so we can be confident that there are no external sources of modeling error that could affect the results. Please see the general response for additional discussion and questions around real-world experiments.
> >
> > Bernard Dubois (1975) ,"Ideal Point Versus Attribute Models of Brand Preference: a Comparison of Predictive Validity", in NA - Advances in Consumer Research Volume 02, eds. Mary Jane Schlinger, Ann Abor, MI : Association for Consumer Research, Pages: 321-334.

---

### Official Review · Reviewer_KXvU · 2022-07-19

**Rating:** 6
**Confidence:** 3
**Soundness:** 3 good
**Presentation:** 4 excellent
**Contribution:** 3 good

**Summary:**

This paper investigates a joint method that can perform preference learning and metric learning.


**Questions:**

- Although I wasn't able to fully grasp the detailed derivations and proofs, they seem sufficient to support the model's identifiability, prediction, and recovery claims. However, I hope other reviewers will check their soundness by looking at the full Appendix.
- The synthetic dataset experiment results are not consistently better than the baseline settings. More justifications and interpretations of the synthetic experiments would be helpful.
-  The real dataset experiments were conducted with only one dataset, with constrained setups. Why haven't the authors explored other preference datasets like OpenAI's summarization dataset?
- The lack of empirical support suggests that the title of this work should be framed focusing on the "theoretical analysis" of the identifiability and recovery guarantees of simultaneous preference and metric learning.



**Limitations:**

Yes. I really appreciate seeing that the authors have listed out the appropriateness of the assumption and their potential impracticality in real-world settings.

**Strengths And Weaknesses:**

Strengths:
- The main contribution of this paper is modeling a shared metric between all users rather than learning separate metrics for each user
- This simple idea makes the sample complexity reduced from O(d2) paired comparisons per user to only O(d),

Weakness:
 - The lack of empirical results and justification.

---

> ### Author Response · Authors · 2022-08-02
> **Thank you for reviewing our paper**
>
> Thank you for taking the time to read and review our work. Please refer to the general response to all reviewers for a summary of the core points of our submission as well as for responses to questions shared by multiple reviewers. Below, we respond to your individual points directly.
>
> Regarding contributions and empirical results: the core emphasis of this paper is to provide the first theoretical analysis of simultaneous metric and preference learning from paired comparisons over multiple users. In particular, we are interested in studying the metric cost amortization effect that comes from learning in large crowds. As such, the empirical baselines we evaluate do not stem from prior algorithms in the literature, but rather are ablated sets of constraints for our proposed optimization program. Hence, the differences here are due to the geometry of the constraint sets, not the objective functions or high-level algorithms. As we discuss in the paragraph starting at 325, for the low-rank synthetic experiment featured in our main text we generally find that “Nuclear full” performs the best uniformly out of these constraint sets. This is because it outperforms the other methods in terms of test accuracy, recovers higher quality metrics at a lower number of queries, and performs significantly better ideal point recovery. We discuss a more nuanced comparison between the ablation constraint sets in the text and provide geometric intuition for why “Nuclear full” (and “Nuclear split” following closely behind) may be performing the best. We will expand on these points in the next iteration.
>
> Regarding additional real data experiments: the core theoretical focus of this work is the study of amortization effects and demonstrating recovery results. To our understanding, the OpenAI summarization dataset is not an ideal fit for the model we study in our paper, because there is no overlap in the documents that individual raters are reviewing. In contrast, in the color dataset a crowd of users are all judging the same set of colors and giving their personal preferences (which need not agree with other users). We believe that the color dataset is a strong practical dataset to study the linear metrics model employed by our work, though we agree that extending to more complex application settings is an interesting avenue for future work.

---

### Official Review · Reviewer_BgB1 · 2022-07-24

**Rating:** 6
**Confidence:** 2
**Soundness:** 3 good
**Presentation:** 3 good
**Contribution:** 3 good

**Summary:**

The purpose of this paper is to investigate simultaneous preference and metric learning from a crowd of respondents. For learning the metric and ideal points, the authors ask each user a series of questions in the form “do you prefer item i or item j?” These comparisons can be used to provide a signal from which user's optimal preference latent point uk a common metric M can be estimated. The main contributions of this work are threefold: 1) This paper can provide the sufficient number of items and paired comparisons needed for an exact preference and metric learning over generic items. This is possible when noiseless difference of item distances are known exactly. 2) Instead of exact distance comparisons, if the provided paired comparisons are noisy, this paper provides prediction guarantees when learning from such comparisons. Based on the authors' work, two convex algorithms that learn full-rank and low-rank metrics are presented with prediction error bounds. 3) Authors also provide guaranteed recovery of the ideal point and the metric when learning from noisy binary labels under assumptions about the response distribution.
In the experiment section, this approach is evaluated on synthetic datasets as well as real psychometric data examining individual and collective color preferences.

**Questions:**

line 103: "Note that eq. (1) includes a nonlinear interaction between M and uk; however, by defining vk := −2Muk (which we refer to as user k’s “pseudo-ideal point”) eq. (1) becomes linear in M and vk" --> I don't understand this part. Is it possible to rename something and make it convex? or you are solving another problem which is a relaxation of the main problem? If so, please state that.

When you describe your method, you discuss noisy/noiseless settings. In the experiment section (in the main paper) how do you measure noise? or do you account for noisy setting in the experiments?



**Limitations:**

I appreciate that the paper has discussed its limitations in the supplementary materials throughly. It would be nice to see some of these discussions in the main paper as well, perhaps in a conclusion section.

**Strengths And Weaknesses:**

Strength:

The paper is well-written and easy to read. The paper looks theoretically sound with clearly described assumptions and has provided proofs when needed. In the experiments section, the method has been evaluated on synthetic and real-world data. Authors have provided significant amount of detail about their approach, proofs, additional experiments, broader impact and discussions about the limitations of their approach in the supplementary materials. This makes their paper more clear and easy to understand.

Weaknesses/Concerns:

In the experiment section, it seems like you don't compare your method to [1] as you mentioned this work is closest to your work. In general, it is kind of hard for me to abstract how significant is the contribution of this paper. I can't what has been improved over other state-of-the-art techniques in the experiment section.

Experiments: Color Dataset: "We also compare against a baseline that solves the same optimization as eq. (2) separately for each individual respondent" --> please add the baseline name you used in Fig. 1d in the text as well. It is not clear which baseline you describe in the text.

line 26: add citation for Mahalanobis distance

line 218-222: sometimes you use $M_*$ and sometimes $M^*$ which I think both are the same term.


[1] A. Xu and M. Davenport, “Simultaneous preference and metric learning from paired comparisons,” in Advances in Neural Information Processing Systems, 2020, pp. 454–465.

---

> ### Author Response · Authors · 2022-08-02
> **Thank you for reviewing our paper**
>
> Thank you for taking the time to read and review our work. Please refer to the general response to all reviewers for a summary of the core points of our submission as well as for responses to questions shared by multiple reviewers. Below, we respond to your individual points directly.
>
> Regarding line 103, if $v_k$ and $M$ are recovered exactly, then we can also recover $u_k$ exactly (see footnote 2 for a discussion of the case when $M$ is rank deficient - in this case, only the component of $u_k$ in the row space of $M$ can be measured, and this component is also recoverable exactly if $v_k$ is known). In other words, our insight here is that the non-convex equation (1) can be solved in two stages (with unquantized measurements): solve a convex relaxation in terms of $M$ and $v_k$, and then solve for $u_k$ given the solution to this first stage. To reiterate, in the noiseless, unquantized case, this is an exact solution. For a more general discussion in the one-bit noisy case, please refer to our comments “Regarding mathematical novelty and the convex relaxation” in our response to Reviewer u2T5, since this reviewer asks a similar question in the noisy, one-bit case. We will make sure these points are clear in the next paper revision.
>
> Regarding noise: label noise can exist in the data. When gathering data from human respondents as in the color experiment, this noise can exist due to non-transitivity of responses (e.g., color A is preferred to B, color B is preferred to C, but C is preferred to A), temporal inconsistency in answering preference judgements (preferring A over B at one query, but preferring B over A in another query), and other sources of randomness. In general, we assume that for a pair i,j the label y_ij is a Bernoulli random variable with parameter p_ij (which may be specific to that pair). In the noiseless case, p_ij = 0 or 1. We do not explicitly measure this noise in our experiments, but account for it in our algorithms by utilizing a loss function that encourages the model to agree with the responses, but does so in a “soft” manner (e.g., hinge loss). We also note that the inherent noise level in the synthetic data can be observed by inspecting the performance of the “Oracle” baseline in Figure 1a, which measures test prediction accuracy when the ground-truth parameters (metric and ideal point) are known exactly, and serves as a ceiling on test accuracy. We will clarify these points in the draft.
>
> Regarding a discussion of limitations in the main draft: we agree that moving a discussion of conclusions and limitations to the main draft is a good idea, and will do so in the next revision.
>
> We will also add the individual-user color baseline name to the text, add a citation for Mahalanobis metric learning, and be consistent with our use of $M_*$ and $M^*$ (thank you for pointing this out!).

---

### Author Response · Authors · 2022-08-02
**General Response**

Thank you to all of the reviewers for their careful assessment of our work. We will respond individually to the questions and comments posed by each reviewer, but first summarize the core contributions of our work in the interest of clarity, and respond to general questions posed by multiple reviewers:
1. The main contribution of our paper is the first theoretical analysis of simultaneous metric and preference learning from paired comparisons — for multiple users or a single user — which has previously been established as a model of interest to the NeurIPS community in the single user case [7].
2. In the noiseless setting, we establish necessary and sufficient conditions for recovery. To do so, we develop a novel, linear algebraic analysis of pairwise selection matrices.
3. In the noisy setting, we develop and analyze a convex relaxation in both the full and low rank settings. We establish the first prediction error bounds for simultaneous metric and preference learning.
4. We additionally provide recovery guarantees from noisy data under an assumed response distribution. This analysis serves as a culmination of our theoretical results, combining the identifiability tools involving pair selection matrices from Section 2 with guarantees on prediction error from Section 3.
5. A key thread in our theoretical results is that by having a large crowd of users (large $K$), the cost of learning the metric can be amortized over the crowd. To demonstrate this point and compare algorithm performance against several ablation baselines, we show several empirical results on a simulated and a real dataset.

One question shared among reviewers is the extent of the experiments. Given that our paper is mostly theoretical, our experimental results serve an initial evaluation of our proposed methods rather than a comprehensive empirical study over large-scale datasets. The scale of our experiments is consistent with those in [7]. Both papers have one evaluation in a low-dimensional synthetic dataset (ambient dimension of 10 or fewer), and one evaluation on a low-dimensional real-world dataset. In [7], their real-world data consisted of a graduate admissions dataset with five features per item. As this data was not public, our real-world test instead leverages a dataset of color preferences with three features per item. We draw these parallels to show that the scale of our paper’s results is consistent with previous work published at NeurIPS. Furthermore, the use of color data itself provides a sound scientific baseline to compare our method against: as we discuss in the paper, each color is parameterized in a color space specifically designed to be perceptually uniform. Therefore, it is reasonable to expect that the identity metric ($M = I$, i.e., Euclidean distance) is a reasonable baseline in this setting. We show in fact that this perceptual uniformity misses an element of preference, as our learned metric results in higher preference prediction accuracy and weighs certain features more strongly than others.

---

> ### Author Response · Authors · 2022-08-02
> **Empirical comparison to [7]**
>
> In response to requests by Reviewers BgB1 and eLgv we ran the algorithms in [7] in the full-rank metric setting of Figure 3 in our paper’s appendix (note that the algorithms in [7] are not designed for low-rank recovery), using the default parameters in their code. We refer to the algorithms of Xu and Davenport in [7] as “XD-single” and “XD-alt” below, referring respectively to their “single step estimation” and “alternating estimation” algorithms. We have included these results at specific numbers of paired queried per user, along with the corresponding values from our algorithms in Figure 3. We have separated the multi-user methods (“Nuclear full”, “Frobenius metric”) from the single-user methods. XD-single and XD-alt are evaluated in an analogous manner to “Nuclear full, single” as described in our paper: the crowd of $K = 10$ users generates responses according to the same metric, but each user is solved separately, as if each had their own individual metric. Test accuracy, metric recovery error, and ideal point recovery error are then averaged over all users.
>
> Test accuracy (mean, 30 trials, higher is better)
>
> Number of pairs per user      50          100        150        200        250        300
> Nuclear full                            0.770     0.794     0.803     0.809     0.812     0.813
> Frobenius metric                   0.755     0.786     0.800     0.806     0.809     0.812
>
> Nuclear full, single                0.720     0.752     0.768     0.779     0.785     0.791
> XD-single                              0.641     0.682     0.701     0.712     0.727     0.731
> XD-alt                                   0.683     0.715      0.737     0.745     0.757     0.762
>
>
> Metric (M) relative error (mean, 30 trials, lower is better)
>
> Number of pairs per user      50          100        150        200        250        300
> Nuclear full                            0.530     0.412     0.365     0.332     0.302     0.286
> Frobenius metric                   0.595     0.463     0.401     0.370     0.338     0.323
>
> Nuclear full, single                1.134     0.975     0.874     0.797     0.747     0.704
> XD-single                              1.003     1.889     1.816     1.513     1.233     1.085
> XD-alt                                    0.791     0.757     0.733     0.715     0.680     0.674
>
>
> Ideal point (u) relative error (mean, 30 trials, lower is better)
>
> Number of pairs per user      50           100        150        200        250        300
> Nuclear full                            0.817      0.761     0.745     0.732     0.721     0.717
> Frobenius metric                   0.877      0.764     0.737     0.721     0.712     0.706
>
> Nuclear full, single                 0.985     0.879     0.841     0.819     0.794     0.782
> XD-single                              1.000      0.963     0.910     0.882     0.862     0.866
> XD-alt                                    0.922      0.906     0.887     0.887     0.875     0.875
>
> Overall, among the single-user methods “Nuclear full, single” recovers the highest test prediction accuracy and lowest ideal point recovery error, while XD-alt results in the lowest metric estimation error. While our paper and [7] share the same model in the single-user case (as a reminder, [7] does not consider the multi-user case), each frames the problem differently (our novel formulation is well-suited to analysis via selection matrices and intuitively decouples the ideal point from the metric) and so it is intuitively unclear how each single-user algorithm differs theoretically. Since the focus of our work is on the multi-user case rather than the single-user case, we propose to include the above results in an appendix of our next paper revision.

---

### Meta-Review · Area_Chair_JnnC · 2022-08-26

**Recommendation:** Accept
**Confidence:** Certain

**Metareview:**

All reviewers felt this paper deserved to be accepted given its strong theoretical contributions and clear presentation. The authors may wish to consider additional experiments given that a number of reviewers found the empirical evaluation to be relatively weak, and are encouraged to follow through on incorporating conclusions and limitations into the main paper body.

**Award:**

No

---

### Decision · Program_Chairs · 2022-09-14

Accept